# BA-LoRA: Bias-Alleviating Low-Rank Adaptation to Mitigate Catastrophic Inheritance in Large Language Models

**Yupeng Chang[1], Yi Chang[1,2,3], Yuan Wu[1,2]**[*]
[1]School of Artificial Intelligence, Jilin University
[2]Key Laboratory of Symbolic Computation and Knowledge Engineering, Jilin University
[3]International Center of Future Science, Jilin University
`yupeng.chang.ai@gmail.com, yichang@jlu.edu.cn, yuanwu@jlu.edu.cn`

## Abstract

Parameter-efficient fine-tuning (PEFT) has become a *de facto* standard for adapting large language models (LLMs). However, we identify a critical vulnerability within popular low-rank adaptation methods such as LoRA: they can exacerbate "Catastrophic Inheritance"—the unchecked propagation of biases, noise, and data imbalances from pre-training. This phenomenon can degrade model robustness and fairness, undermining the benefits of efficient adaptation. To address this, we introduce Bias-Alleviating Low-Rank Adaptation (BA-LoRA). Our approach is founded on a principled decomposition of Catastrophic Inheritance into three core challenges: Knowledge Drift, Representation Collapse, and Overfitting to Noise. BA-LoRA systematically mitigates these issues by incorporating a trio of targeted regularizers: consistency, diversity, and an SVD-based term, designed to preserve core knowledge, promote representational richness, and encourage robust, low-rank output representations, respectively. We conduct comprehensive evaluations on a suite of Natural Language Generation (NLG) and Natural Language Understanding (NLU) tasks using diverse, prominent open-source language models (e.g., LLaMA-2-7B and DeBERTa-v3-base). Our results show that BA-LoRA not only outperforms state-of-the-art LoRA variants in terms of performance and stability, but also demonstrates superior robustness and bias mitigation on targeted evaluations. These results provide evidence that BA-LoRA can counteract the adverse effects of Catastrophic Inheritance.

## 1 Introduction

Large language models (LLMs) like GPT-4 (OpenAI, 2023) and LLaMA (Touvron et al., 2023) have redefined the state-of-the-art in natural language processing (NLP), largely due to their training on vast, web-scale corpora (Zhao et al., 2023; Chang et al., 2024). This strategy, while enabling unprecedented generalization (Gao et al., 2020; Penedo et al., 2023), comes at a cost: models can inherit and internalize the biases, noise, and imbalances latent within these large-scale datasets (Parashar et al., 2024; Liu & He, 2024; Chen et al., 2024).

Recent research shows that these inherited flaws can degrade model performance and persist even after fine-tuning, posing significant risks to fairness and safety (Qi et al., 2023; Bommasani et al., 2021; Mallen et al., 2022; Carlini et al., 2023). For example, noise within the training data can degrade model generalization (Chen et al., 2024), while the long-tailed distribution of concepts can cause LLMs to overemphasize overrepresented topics (Zhu et al., 2024; Dong et al., 2023).

This phenomenon, termed "Catastrophic Inheritance" (Chen et al., 2024), arises when models inherit such biases, noise, and imbalances from pre-training. In this work, we focus on how these inherited artifacts can be further amplified during downstream fine-tuning, a challenge that motivates our investigation into targeted mitigation strategies. While constructing less biased datasets and

---

[*]Corresponding author

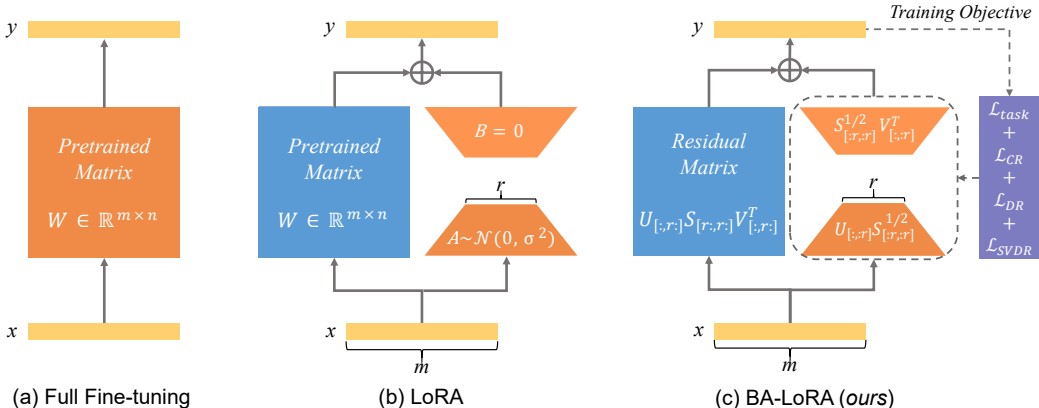

Figure 1: Comparison of three fine-tuning frameworks: (a) Full Fine-tuning updates the entire matrix $W$; (b) LoRA trains a low-rank adapter while keeping $W$ frozen; and (c) our proposed BA-LoRA. Blue and orange modules denote frozen and trainable parameters, respectively. BA-LoRA initializes its adapter and residual matrix ($W^{\mathrm{res}}$) using the SVD of $W$ following PiSSA (Meng et al., 2024). It augments the task loss ($\mathcal{L}_{\mathrm{task}}$) with three regularizers, shown in purple, to mitigate Catastrophic Inheritance by preserving knowledge, promoting diversity, and emphasizing dominant data patterns.

developing more robust model architectures are prominent approaches (Liu & He, 2024), this study explores an alternative: innovations in fine-tuning. Fine-tuning is a powerful method for enhancing task-specific performance and aligning models with user intent (Han et al., 2024; Ouyang et al., 2022). However, its computational demands are substantial; for instance, 16-bit fine-tuning of an LLaMA-65B model requires over 780 GB of GPU memory (Dettmers et al., 2024). To address these limitations, parameter-efficient fine-tuning (PEFT) techniques, such as Low-Rank Adaptation (LoRA) (Hu et al., 2021), have gained prominence.

LoRA enables efficient fine-tuning by approximating parameter updates using low-rank matrices. As illustrated in Figure 1(a), Full Fine-tuning directly updates the entire weight matrix $W$. In contrast, LoRA (Figure 1(b)) introduces a learnable low-rank adapter $\Delta W = AB$, where $A \in \mathbb{R}^{m \times r}$ and $B \in \mathbb{R}^{r \times n}$ are trainable matrices with a rank $r \ll \min(m, n)$. Only $A$ and $B$ are updated, while the original weights $W$ remain frozen. By initializing $A$ with scaled random values and $B$ to zero, LoRA ensures the adapter has no effect at the start of training. For a linear layer, the forward pass is then computed as $Y = X(W + AB)$, reducing the trainable parameter count and improving fine-tuning efficiency (Hu et al., 2021).

While PEFT methods like LoRA offer remarkable efficiency, their constrained, low-rank updates introduce a potential vulnerability: they can exacerbate Catastrophic Inheritance when fine-tuning on noisy or imbalanced data without explicit regularization. By restricting model adjustments to a low-dimensional bottleneck, these methods may lack sufficient degrees of freedom to correct for inherited biases and may instead amplify spurious correlations from pre-training data. To bridge this gap, we argue that a more principled approach is needed. We first deconstruct Catastrophic Inheritance into three primary failure modes: **Knowledge Drift**, where the model unintentionally forgets robust pre-trained knowledge while learning new tasks (Kirkpatrick et al., 2017); **Representation Collapse**, where fine-tuning on imbalanced data causes output diversity to decrease (Bardes et al., 2021); and **Overfitting to Noise**, where the model learns spurious correlations from the training data that hinder generalization (Chen et al., 2019). This paper introduces Bias-Alleviating Low-Rank Adaptation (BA-LoRA), a method that systematically mitigates these issues. As depicted in Figure 1(c), BA-LoRA builds upon the efficient PiSSA (Meng et al., 2024) initialization and incorporates a trio of regularizers: a consistency regularizer to combat Knowledge Drift, a diversity regularizer to prevent Representation Collapse, and an SVD-based regularizer to mitigate Overfitting to Noise. Recognizing the differences between NLU and NLG tasks, we tailor these strategies accordingly.

Our comprehensive evaluation demonstrates BA-LoRA's strong performance and analyzes the sources of its effectiveness. BA-LoRA consistently outperforms leading LoRA variants across diverse benchmarks, including mathematical reasoning, coding, and conversational AI for NLG, as well as the

GLUE benchmark (Wang et al., 2018) for NLU, using models such as LLaMA-2-7B (Touvron et al., 2023) and DeBERTa-v3-base (He et al., 2021). Moreover, we move beyond standard leaderboards to test our central hypothesis. An empirical comparison of models pre-trained with high-quality curated (RoBERTa (Liu et al., 2019)) versus noisier web-scale (T5 (Raffel et al., 2020)) data shows that BA-LoRA achieves larger gains on the latter, which is consistent with our hypothesis that BA-LoRA is particularly beneficial when mitigating the effects of inherited noise. This primary finding is supported by comprehensive ablation studies and qualitative visualizations that support the importance of our three-pronged strategy. Together, these results not only demonstrate BA-LoRA's effectiveness but also support our framework for understanding and mitigating this phenomenon.

## 2 METHOD

### 2.1 PRINCIPAL SINGULAR VALUES AND SINGULAR VECTORS ADAPTATION (PiSSA)

As a variant of LoRA, PiSSA (Meng et al., 2024) accelerates convergence by retaining the core LoRA architecture while modifying its initialization. It leverages the principal singular components of the original weight matrix $W$ to initialize the adapter matrices $A$ and $B$, and places the remaining components in a residual matrix $W^{\text{res}} \in \mathbb{R}^{m \times n}$. We write the SVD of $W \in \mathbb{R}^{m \times n}$ as $W = USV^{\top}$, where $U$ and $V$ contain the left and right singular vectors, and $S$ is a diagonal matrix of singular values sorted in descending order. PiSSA partitions the singular components into principal $\{U_{[:,:r]}, S_{[:r,:r]}, V_{[:,:r]}\}$ and residual $\{U_{[:,r:]}, S_{[r:,r:]}, V_{[:,r:]}\}$ parts, where $r$ is the user-specified adapter rank. The principal components are then used to initialize the low-rank adapter with $A \in \mathbb{R}^{m \times r}$ and $B \in \mathbb{R}^{r \times n}$:

$$A = U_{[:,:r]} \, S_{[:r,:r]}^{1/2} \in \mathbb{R}^{m \times r} \tag{1}$$

$$B = S_{[:r,:r]}^{1/2} \, V_{[:,:r]}^{\top} \in \mathbb{R}^{r \times n} \tag{2}$$

The residual matrix $W^{\text{res}}$ remains frozen during fine-tuning:

$$W^{\text{res}} = U_{[:,r:]} \, S_{[r:,r:]} \, V_{[:,r:]}^{\top} \in \mathbb{R}^{m \times n} \tag{3}$$

PiSSA preserves the pre-trained model's full capacity at the start of fine-tuning by using $W = W^{\text{res}} + AB$. This approach prioritizes adaptation along the leading singular directions, thereby accelerating convergence from the start. Inheriting LoRA's benefits of reduced trainable parameter count and deployment simplicity, PiSSA further uses SVD-based initialization to improve fine-tuning efficiency. This concentration of updates also motivates the output-space regularizers introduced in BA-LoRA.

### 2.2 BIAS-ALLEVIATING LOW-RANK ADAPTATION (BA-LoRA)

Catastrophic Inheritance refers to vulnerabilities arising from biases, noise, and imbalances inherent in large-scale training data, such as attribute bias and class imbalance, that can degrade downstream performance, introduce unfair biases, and raise safety concerns. These effects manifest during fine-tuning as three distinct subproblems: **Knowledge Drift**, where the model unintentionally forgets or distorts robust pre-trained knowledge (Kirkpatrick et al., 2017); **Representation Collapse** (Bardes et al., 2021); and **Overfitting to Noise** (Chen et al., 2019). To address them, we propose BA-LoRA, a unified framework with three regularizers—consistency, diversity, and SVD-based—each aligned with one subproblem. Instead of constraining low-rank adapter weights, BA-LoRA regularizes the output space to shape functional behavior and mitigate inherited biases, with tailored variants for NLU and NLG tasks.

#### 2.2.1 REGULARIZATIONS FOR NLU TASKS

**Consistency Regularization.** To combat **Knowledge Drift**, we adopt a knowledge distillation approach based on standard practices (Hinton et al., 2015), using the Kullback-Leibler (KL) divergence

between the temperature-scaled probability distributions. Let $\mathbf{Z}_P, \mathbf{Z}_F \in \mathbb{R}^{N \times D}$ be the batch output logits from the pre-trained and fine-tuned models, respectively, where $N$ is the batch size and $D$ is the number of classes. The loss is defined as:

$$\mathcal{L}_{\text{CR\_NLU}} = T^2 \cdot \text{KL}\big(\text{softmax}(\mathbf{Z}_P/T) \,\|\, \text{softmax}(\mathbf{Z}_F/T)\big) \tag{4}$$

where $T$ is a temperature parameter that softens the distributions. This objective encourages the fine-tuned model to mimic the decision behavior of the pre-trained model on examples where the teacher signal is reliable, thereby preserving foundational knowledge. The $T^2$ scaling factor keeps gradient magnitudes commensurate with standard cross-entropy loss.

**Diversity Regularization.** To counteract **Representation Collapse**, particularly on imbalanced datasets, we promote diversity in the model's predictions across a batch. Inspired by (Bardes et al., 2021), we regularize the batch-wise output logits to decorrelate the predictions for different classes. Let $\mathbf{Z}_F \in \mathbb{R}^{N \times D}$ be the logit matrix for a batch. We first center the logits and then compute the $D \times D$ covariance matrix $C(\mathbf{Z}_F)$. The regularizer penalizes the off-diagonal elements of this matrix:

$$\mathcal{L}_{\text{DR\_NLU}} = \frac{1}{D} \sum_{i \neq j} [C(\mathbf{Z}_F)]_{i,j}^2 \tag{5}$$

where the covariance matrix is computed as:

$$C(\mathbf{Z}_F) = \frac{1}{N-1} \mathbf{Z}_{\text{centered}}^\top \mathbf{Z}_{\text{centered}}, \quad \text{where } \mathbf{Z}_{\text{centered}} = \mathbf{Z}_F - \bar{\mathbf{Z}}_F \tag{6}$$

Here, $\bar{\mathbf{Z}}_F$ is the matrix where each row is the mean logit vector computed over the batch. This loss discourages excessive correlation between the model's predictions for any two distinct classes across the batch, thus preventing the model from collapsing toward a few dominant classes.

**Singular Value Decomposition Regularization.** To mitigate **Overfitting to Noise** and encourage the model to learn robust features, we introduce a regularizer that encourages the spectral energy of the batch-wise output logit matrix to concentrate in its leading singular components. Inspired by the principle that dominant singular values capture the most salient data patterns (Chen et al., 2019), this regularizer incentivizes the model to form simpler, more coherent decision boundaries for samples within a batch, rather than fitting to high-frequency logit fluctuations that are poorly aligned with the task labels. On the fine-tuned logit matrix $\mathbf{Z}_F \in \mathbb{R}^{N \times D}$, we perform SVD and maximize the ratio of spectral energy concentrated in the top-$k$ singular values:

$$\mathcal{L}_{\text{SVDR\_NLU}} = -\frac{\sum_{i=1}^{k} \sigma_i}{\sum_{j=1}^{\min(N,D)} \sigma_j} \tag{7}$$

where $\sigma_i$ is the $i$-th largest singular value. The hyperparameter $k$ controls the rank preference. In the NLU experiments, where the number of classes $D$ is typically moderate, the computational cost of performing an exact SVD is minimal and does not substantially affect training efficiency.

**Overall Objective Function for NLU.** The overall NLU objective is formulated as follows:

$$\mathcal{L}_{\text{NLU}} = \mathcal{L}_{\text{task\_NLU}} + \lambda_1 \mathcal{L}_{\text{CR\_NLU}} + \lambda_2 \mathcal{L}_{\text{DR\_NLU}} + \lambda_3 \mathcal{L}_{\text{SVDR\_NLU}} \tag{8}$$

where $\mathcal{L}_{\text{task\_NLU}}$ represents the standard cross-entropy loss for the downstream task, and $\lambda_1$, $\lambda_2$, and $\lambda_3$ are weighting parameters that balance the contribution of each regularization term.

### 2.2.2 REGULARIZATIONS FOR NLG TASKS

**Consistency Regularization.** To combat **Knowledge Drift**, we employ temperature-controlled knowledge distillation (Hinton et al., 2015), using the Kullback-Leibler (KL) divergence between the output distributions of the fine-tuned (student) model, $\mathcal{P}_F$, and the pre-trained (teacher) model, $\mathcal{P}_P$. A temperature parameter $T$ softens these distributions, encouraging the student to learn the teacher's softened output distribution rather than only its top prediction, especially on tokens where the teacher distribution provides a meaningful and well-calibrated soft target. The loss is defined as:

$$\mathcal{L}_{\text{CR\_NLG}} = T^2 \cdot \frac{1}{M} \sum_{i=1}^{M} \text{KL}\big(\mathcal{P}_P(\cdot \mid \mathbf{x}, i; T) \,\|\, \mathcal{P}_F(\cdot \mid \mathbf{x}, i; T)\big) \tag{9}$$

where for an input sequence $\mathbf{x}$, $\mathbf{z}_{P,i}$ and $\mathbf{z}_{F,i}$ denote the teacher and student logit vectors at position $i$, respectively. The temperature-scaled conditional probability distributions over the vocabulary are formally defined as $\mathcal{P}_P(\cdot \mid \mathbf{x}, i; T) = \text{softmax}(\mathbf{z}_{P,i}/T)$ and $\mathcal{P}_F(\cdot \mid \mathbf{x}, i; T) = \text{softmax}(\mathbf{z}_{F,i}/T)$. The loss is averaged over all $M$ valid (non-padded) tokens in the batch. The $T^2$ scaling factor maintains gradient magnitude consistency with standard distillation.

**Diversity Regularization.** To counteract **Representation Collapse** in generation, we address a fundamental challenge: naively maximizing the entropy of the entire vocabulary distribution can conflict with the task objective of producing coherent text (Gat et al., 2020). We address this with a *focused* entropy regularizer. Inspired by Top-$K$ sampling, our method promotes diversity within the set of most plausible candidate tokens, denoted as $\mathcal{V}_{\text{top-}K}$. For each token, we minimize the negative entropy, equivalently maximizing entropy, computed solely within this restricted set:

$$\mathcal{L}_{\text{DR\_NLG}} = \frac{1}{M} \sum_{i=1}^{M} \sum_{v \in \mathcal{V}_{\text{top-}K}^{(i)}} \mathcal{P}_F'(v \mid \mathbf{h}_i) \log \mathcal{P}_F'(v \mid \mathbf{h}_i) \tag{10}$$

where $\mathcal{P}_F'(v \mid \mathbf{h}_i)$ is the re-normalized probability from the fine-tuned model for candidate token $v$ within the set $\mathcal{V}_{\text{top-}K}^{(i)}$ for the $i$-th valid token, given the corresponding final hidden state $\mathbf{h}_i$. It is computed as:

$$\mathcal{P}_F'(v \mid \mathbf{h}_i) = \frac{\mathcal{P}_F(v \mid \mathbf{h}_i)}{\sum_{v' \in \mathcal{V}_{\text{top-}K}^{(i)}} \mathcal{P}_F(v' \mid \mathbf{h}_i)} \tag{11}$$

**Singular Value Decomposition Regularization.** To mitigate **Overfitting to Noise**, we regularize the structure of the batch-wise output logit matrix. Building on the principle that dominant singular values capture salient data patterns (Chen et al., 2019), we encourage a low-rank structure. For tractability with large vocabularies, we use randomized SVD (Halko et al., 2011) and normalize by the Frobenius norm to avoid expensive full-spectrum computation. We thus define the loss as the negative ratio of the sum of the top-$k$ singular values to the Frobenius norm:

$$\mathcal{L}_{\text{SVDR\_NLG}} = -\frac{\sum_{i=1}^{k} \tilde{\sigma}_i}{\|\mathbf{Z}_{\text{valid}}\|_F} \tag{12}$$

Here, $\tilde{\sigma}_i$ is the $i$-th largest approximated singular value of the valid logit matrix $\mathbf{Z}_{\text{valid}} \in \mathbb{R}^{M \times |\mathcal{V}|}$, where $|\mathcal{V}|$ is the vocabulary size, and $\|\cdot\|_F$ denotes the Frobenius norm.

**Overall Objective Function for NLG.** Integrating these components, the final objective function for NLG tasks is a weighted sum of the task loss and the three regularization terms:

$$\mathcal{L}_{\text{NLG}} = \mathcal{L}_{\text{task\_NLG}} + \lambda_1 \mathcal{L}_{\text{CR\_NLG}} + \lambda_2 \mathcal{L}_{\text{DR\_NLG}} + \lambda_3 \mathcal{L}_{\text{SVDR\_NLG}} \tag{13}$$

where $\mathcal{L}_{\text{task\_NLG}}$ is the standard causal language modeling loss. A detailed sensitivity analysis is provided in Appendix C.2.

## 3 EXPERIMENTS

### 3.1 IMPLEMENTATION DETAILS

Our experimental setup is broadly aligned with recent PEFT studies (Meng et al., 2024). For NLG tasks on LLaMA-2-7B, we use the AdamW optimizer (Loshchilov & Hutter, 2017) with a learning rate of $2 \times 10^{-5}$, a cosine schedule with a 0.03 warmup ratio, and no weight decay. We set `lora_dropout` to 0, use BFloat16 precision, set both the LoRA rank ($r$) and alpha ($\alpha$) to 128, and use an effective batch size of 32. The regularization weights for our method are set to $\lambda_1 = 0.025, \lambda_2 = 0.005, \lambda_3 = 0.005$, with an SVD rank of $k = 10$. For NLU tasks on the GLUE benchmark, learning rates, batch sizes, and other core hyperparameters are task-specific to align with the corresponding baselines, as detailed in the appendix. For our method in the NLU setting, we use no weight decay and set the regularization hyperparameters to $\lambda_1 = 0.15, \lambda_2 = 0.03, \lambda_3 = 0.03$, with an SVD rank of $k = 5$. For each backbone, regularization weights are selected via coarse grid search on a held-out validation split and then kept fixed for that setting. Unless otherwise specified, our runs were conducted on NVIDIA A40 GPUs and averaged over three random seeds (42, 1024, 2024). Detailed hyperparameter configurations for all models and tasks are available in Appendix B.

Table 1: Performance comparison on NLG tasks. We compare our method (BA-LoRA) against popular fine-tuning baselines, including Full Fine-tuning and various state-of-the-art parameter-efficient techniques. The best results in each column are highlighted in **bold**. Avg is the arithmetic mean of the five reported metrics.

| Methods | GSM8K | MATH | HumanEval | MBPP | MT-Bench | Avg |
|---|---|---|---|---|---|---|
| Full FT | $48.9_{\pm0.49}$ | $7.48_{\pm0.22}$ | $20.52_{\pm0.29}$ | $23.64_{\pm0.38}$ | $4.85_{\pm0.09}$ | 21.08 |
| LoRA | $42.68_{\pm0.54}$ | $5.92_{\pm0.15}$ | $16.80_{\pm0.38}$ | $21.51_{\pm0.43}$ | $4.60_{\pm0.14}$ | 18.30 |
| AdaLoRA | $41.95_{\pm0.90}$ | $6.24_{\pm0.38}$ | $18.10_{\pm0.46}$ | $20.19_{\pm0.71}$ | $4.79_{\pm0.18}$ | 18.25 |
| DoRA | $41.77_{\pm0.74}$ | $6.20_{\pm0.48}$ | $16.86_{\pm0.54}$ | $21.60_{\pm0.49}$ | $4.48_{\pm0.14}$ | 18.18 |
| MiLoRA | $43.09_{\pm1.16}$ | $6.31_{\pm0.39}$ | $17.55_{\pm0.24}$ | $20.22_{\pm0.37}$ | $4.50_{\pm0.17}$ | 18.33 |
| LoRA+ | $47.84_{\pm0.39}$ | $7.21_{\pm0.49}$ | $20.07_{\pm0.38}$ | $23.69_{\pm0.29}$ | $5.11_{\pm0.06}$ | 20.78 |
| LoRA-FA | $40.25_{\pm0.46}$ | $5.66_{\pm0.47}$ | $15.91_{\pm0.41}$ | $20.01_{\pm0.32}$ | $4.67_{\pm0.12}$ | 17.30 |
| LoRA-GA | $50.47_{\pm0.98}$ | $7.13_{\pm0.44}$ | $19.44_{\pm0.45}$ | $23.05_{\pm0.40}$ | $5.04_{\pm0.10}$ | 21.03 |
| PiSSA | $51.48_{\pm0.34}$ | $7.60_{\pm0.18}$ | $19.48_{\pm0.45}$ | $23.84_{\pm0.46}$ | $4.92_{\pm0.07}$ | 21.46 |
| CorDA | $53.90_{\pm0.56}$ | $8.52_{\pm0.27}$ | $21.03_{\pm0.37}$ | $24.15_{\pm0.44}$ | $5.15_{\pm0.09}$ | 22.55 |
| CorDA++ | $55.03_{\pm0.52}$ | $8.95_{\pm0.37}$ | $21.76_{\pm0.39}$ | $24.74_{\pm0.47}$ | $\mathbf{5.64}_{\pm0.12}$ | 23.22 |
| **BA-LoRA** | $\mathbf{55.86}_{\pm0.35}$ | $\mathbf{9.47}_{\pm0.52}$ | $\mathbf{23.58}_{\pm0.25}$ | $\mathbf{36.86}_{\pm0.31}$ | $5.11_{\pm0.05}$ | **26.18** |

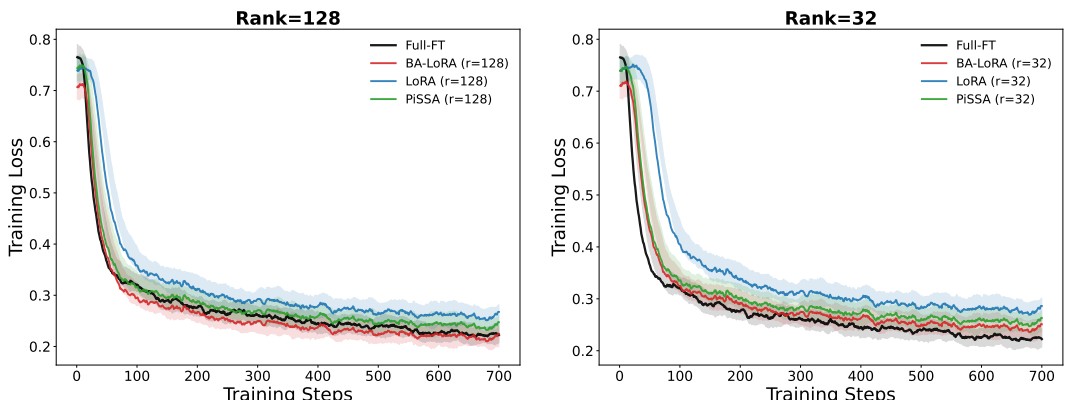

Figure 2: Training task loss of Full Fine-tuning (Full FT), LoRA, PiSSA, and BA-LoRA on Meta-MathQA: (left) rank 128 and (right) rank 32. All curves are smoothed for visual clarity.

## 3.2 RESULTS AND ANALYSIS

### 3.2.1 PERFORMANCE ON NLG AND NLU TASKS

To evaluate BA-LoRA on NLG tasks, we conduct a fair comparison against strong baselines (Table 1). To ensure a matched setup, we report baseline scores directly from their original publications (see Appendix B.3) and evaluate BA-LoRA under the same recipe. These baselines include Full Fine-tuning, LoRA (Hu et al., 2021), AdaLoRA (Zhang et al., 2023b), DoRA (Liu et al., 2024), MiLoRA (Wang et al., 2024a), LoRA+ (Hayou et al., 2024), LoRA-FA (Zhang et al., 2023a), LoRA-GA (Wang et al., 2024b), PiSSA (Meng et al., 2024), CorDA (Yang et al., 2024), and CorDA++ (Yang et al., 2025). Following the standard literature setting of using 100K data points and a single epoch for efficiency, we fine-tune LLaMA-2-7B (Touvron et al., 2023) equipped with BA-LoRA on MetaMathQA (Yu et al., 2023) and assess mathematical problem-solving capabilities (Luo et al., 2025) using the GSM8K (Cobbe et al., 2021) and MATH (Yu et al., 2023) validation sets, reporting accuracy. Similarly, for coding tasks, our model is fine-tuned on CodeFeedback (Zheng et al., 2024) and evaluated via HumanEval (Chen et al., 2021) and MBPP (Austin et al., 2021), reporting Pass@1 metrics. To assess conversational abilities, BA-LoRA is trained on WizardLM-Evol-Instruct (Xu et al., 2024) and evaluated on MT-Bench (Zheng et al., 2024), with response quality judged by GPT-4 and first-turn scores reported.

As shown in Table 1, BA-LoRA achieves the best average performance among the compared methods on LLaMA-2-7B. Specifically, compared to the highly competitive CorDA++, BA-LoRA further

Table 2: Performance comparison on NLU benchmarks. We compare BA-LoRA with various PEFT baselines on the DeBERTa-v3-base model. The best result in each column is highlighted in **bold**.

| Methods | #Params | MNLI | SST-2 | MRPC | CoLA | QNLI | QQP | RTE | STS-B | Avg |
|---|---|---|---|---|---|---|---|---|---|---|
| Full FT | 184M | $90.34_{\pm0.18}$ | $\mathbf{96.33}_{\pm0.11}$ | $89.95_{\pm1.07}$ | $71.43_{\pm0.72}$ | $94.24_{\pm0.10}$ | $92.11_{\pm0.28}$ | $83.75_{\pm1.81}$ | $91.04_{\pm0.48}$ | 88.65 |
| BitFit | 0.1M | $89.54_{\pm0.29}$ | $94.68_{\pm0.11}$ | $87.95_{\pm1.33}$ | $67.31_{\pm0.49}$ | $92.45_{\pm0.17}$ | $88.72_{\pm0.45}$ | $79.12_{\pm0.39}$ | $91.63_{\pm0.37}$ | 86.43 |
| HAdapter | 1.22M | $90.23_{\pm0.07}$ | $95.38_{\pm0.06}$ | $89.97_{\pm0.27}$ | $68.73_{\pm0.27}$ | $94.31_{\pm0.29}$ | $91.99_{\pm0.28}$ | $84.76_{\pm0.39}$ | $91.58_{\pm0.13}$ | 88.37 |
| PAdapter | 1.18M | $90.42_{\pm0.36}$ | $95.49_{\pm0.10}$ | $89.71_{\pm0.35}$ | $69.04_{\pm0.10}$ | $94.38_{\pm0.26}$ | $92.15_{\pm0.43}$ | $85.53_{\pm0.18}$ | $91.69_{\pm0.13}$ | 88.55 |
| LoRA | 1.33M | $90.71_{\pm0.16}$ | $94.79_{\pm0.16}$ | $89.85_{\pm0.21}$ | $70.05_{\pm0.34}$ | $93.94_{\pm0.09}$ | $92.07_{\pm0.48}$ | $85.43_{\pm0.09}$ | $91.67_{\pm0.29}$ | 88.56 |
| DoRA | 1.27M | $90.48_{\pm0.10}$ | $95.85_{\pm0.08}$ | $91.04_{\pm0.15}$ | $71.03_{\pm0.18}$ | $94.21_{\pm0.37}$ | $92.34_{\pm0.16}$ | $86.19_{\pm0.25}$ | $91.92_{\pm0.38}$ | 89.13 |
| AdaLoRA | 1.27M | $90.87_{\pm0.08}$ | $96.18_{\pm0.43}$ | $90.81_{\pm0.40}$ | $71.64_{\pm0.12}$ | $94.68_{\pm0.46}$ | $92.37_{\pm0.35}$ | $87.78_{\pm0.36}$ | $91.97_{\pm0.43}$ | 89.54 |
| PiSSA | 1.33M | $90.47_{\pm0.44}$ | $95.81_{\pm0.45}$ | $91.48_{\pm0.49}$ | $72.27_{\pm0.29}$ | $94.41_{\pm0.41}$ | $92.21_{\pm0.26}$ | $87.14_{\pm0.08}$ | $91.93_{\pm0.25}$ | 89.47 |
| **BA-LoRA** | 1.33M | $\mathbf{91.26}_{\pm0.49}$ | $96.25_{\pm0.09}$ | $\mathbf{92.11}_{\pm0.55}$ | $\mathbf{75.46}_{\pm0.62}$ | $\mathbf{95.35}_{\pm0.14}$ | $\mathbf{93.63}_{\pm0.52}$ | $\mathbf{88.58}_{\pm0.73}$ | $\mathbf{92.71}_{\pm0.38}$ | **90.67** |

improves performance on the reasoning task GSM8K by 0.83 points and the coding task HumanEval by 1.82 points. While CorDA++ maintains an edge on MT-Bench, BA-LoRA's gains on other benchmarks lead to a higher average score, achieving a 2.96-point uplift over CorDA++. The largest margin appears on MBPP, suggesting that BA-LoRA's output-space regularization can be helpful for code-generation settings with noisy or limited supervision. This performance improvement is further supported by the model's optimization dynamics. As illustrated in Figure 2, BA-LoRA exhibits favorable training behavior on the MetaMathQA dataset. Across both high ($r = 128$) and low ($r = 32$) ranks, our method achieves a lower final training task loss than LoRA and PiSSA, reaching levels comparable to Full Fine-tuning, which suggests that the proposed regularization scheme helps guide optimization toward a favorable solution space. An extensive comparison across diverse model families and scales, including dense and MoE models from 7B to LLaMA-3-70B, is provided in Appendix C.3 (Figure 5).

To assess BA-LoRA on NLU tasks, we experiment on the GLUE benchmark (Wang et al., 2018), which includes two single-sentence classification tasks (CoLA, SST-2), five paired-text classification tasks (MNLI, RTE, QQP, MRPC, QNLI), and one text similarity prediction task (STS-B). The evaluation metrics include accuracy for MNLI, following the evaluation protocol of our baselines; the Matthews correlation coefficient for CoLA; the Pearson correlation coefficient for STS-B; and accuracy for the remaining tasks. We use the DeBERTa-v3-base model (He et al., 2021) and compare BA-LoRA against eight strong baseline methods, including Full Fine-tuning (Full FT), BitFit (Zaken et al., 2021), HAdapter (Houlsby et al., 2019), PAdapter (Pfeiffer et al., 2020), LoRA (Hu et al., 2021), DoRA (Liu et al., 2024), AdaLoRA (Zhang et al., 2023b), and PiSSA (Meng et al., 2024).

Table 2 presents the results of DeBERTa-v3-base across eight NLU tasks, demonstrating the strong overall performance of BA-LoRA. It surpasses all parameter-efficient fine-tuning (PEFT) baselines across all tasks and achieves the highest average score. On average, BA-LoRA outperforms PiSSA and LoRA by 1.20 and 2.11 points, respectively. The consistent, broad-based improvements across this diverse suite of both NLG and NLU tasks provide evidence that our three-pronged strategy mitigates the failure modes associated with Catastrophic Inheritance and offers a practical route to strong and competitive parameter-efficient adaptation.

### 3.2.2 MITIGATING THE EFFECTS OF NOISY PRE-TRAINING DATA

Given that large-scale pre-training corpora from web crawls often contain noise (Gao et al., 2020; Dodge et al., 2021), an important challenge is ensuring that fine-tuning enhances the core signal rather than inherited noise. To investigate BA-LoRA's ability to address this issue, we conduct a comparative study on models pre-trained on corpora with different levels of curation. Our testbeds are RoBERTa-base (Liu et al., 2019), pre-trained on a relatively curated mixed corpus, and T5-base (Raffel et al., 2020), pre-trained on the large-scale C4 web corpus.[1] While these models differ in architecture, their distinct pre-training corpora provide a useful but not fully controlled testbed for evaluating robustness against inherited noise. We evaluate on a representative subset of the GLUE benchmark.

As detailed in Table 3, BA-LoRA achieves the best average performance among strong PEFT baselines. The central finding is that the advantage of BA-LoRA is more pronounced on the model pre-trained on the noisier web-scale corpus. While BA-LoRA achieves a 1.11-point average improvement over

---

[1]Specifically, RoBERTa is trained on a ~160GB mixed corpus including BooksCorpus and English Wikipedia alongside web text. In contrast, the C4 corpus (~750GB) used for T5 is derived from the Common Crawl web scrape via heuristic filtering, making it a useful testbed for web-scale noise.

Table 3: Performance comparison of our method (BA-LoRA) against PEFT baselines (LoRA, PiSSA) on RoBERTa-base and T5-base. Models are evaluated on a subset of the GLUE benchmark. The best result for each model is highlighted in **bold**.

| Model | Methods | MNLI | SST-2 | CoLA | QNLI | MRPC | Avg |
|---|---|---|---|---|---|---|---|
| RoBERTa-base | LoRA | $85.63_{\pm0.01}$ | $94.03_{\pm0.02}$ | $62.40_{\pm0.71}$ | $91.37_{\pm0.97}$ | $87.98_{\pm0.23}$ | 84.28 |
| | PiSSA | $85.72_{\pm0.40}$ | $93.64_{\pm0.13}$ | $67.28_{\pm0.59}$ | $91.40_{\pm0.54}$ | $88.11_{\pm0.24}$ | 85.23 |
| | **BA-LoRA** | $\mathbf{86.59}_{\pm0.58}$ | $\mathbf{94.83}_{\pm0.45}$ | $\mathbf{67.91}_{\pm0.21}$ | $\mathbf{92.28}_{\pm0.37}$ | $\mathbf{90.07}_{\pm0.32}$ | **86.34** |
| T5-base | LoRA | $85.30_{\pm0.04}$ | $94.04_{\pm0.11}$ | $69.35_{\pm0.05}$ | $92.96_{\pm0.09}$ | $68.38_{\pm0.01}$ | 82.01 |
| | PiSSA | $85.75_{\pm0.07}$ | $94.07_{\pm0.06}$ | $74.27_{\pm0.39}$ | $93.15_{\pm0.14}$ | $76.31_{\pm0.51}$ | 84.71 |
| | **BA-LoRA** | $\mathbf{86.91}_{\pm0.48}$ | $\mathbf{95.20}_{\pm0.29}$ | $\mathbf{80.19}_{\pm1.03}$ | $\mathbf{94.12}_{\pm0.32}$ | $\mathbf{83.43}_{\pm0.71}$ | **87.97** |

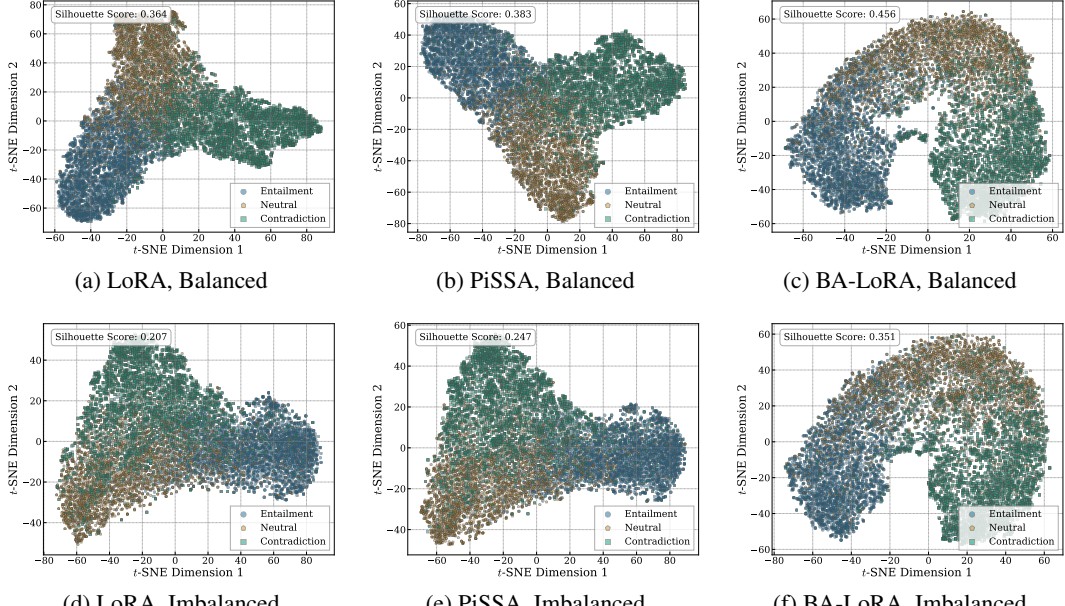

Figure 3: t-SNE visualizations of final hidden-layer representations from RoBERTa-base fine-tuned with LoRA, PiSSA, and BA-LoRA on the MNLI task under balanced (top row) and imbalanced (bottom row) settings.

the strongest baseline (PiSSA) on RoBERTa-base (86.34 vs. 85.23), this improvement increases to 3.26 points on T5-base (87.97 vs. 84.71). The disparity in improvement margin ($\Delta_{\text{T5}} = 3.26$ vs. $\Delta_{\text{RoBERTa}} = 1.11$) is consistent with our hypothesis that BA-LoRA is particularly beneficial for models pre-trained on noisier web corpora, though it does not isolate architectural factors.

### 3.2.3 MITIGATING REPRESENTATIONAL BIAS FROM DATA IMBALANCE

This experiment qualitatively investigates BA-LoRA's capacity to counteract the representational degradation caused by data imbalance, which is one way Catastrophic Inheritance can manifest during downstream fine-tuning. We visualize the final hidden-layer representations from RoBERTa-base fine-tuned on the MNLI task using t-SNE (Van der Maaten & Hinton, 2008). As shown in Figure 3, we compare feature manifolds learned on the standard balanced dataset with those learned from a deliberately imbalanced version, constructed by subsampling the training data to a 100:10:1 ratio for the "Entailment", "Neutral", and "Contradiction" classes. This controlled comparison is designed to simulate the challenge of learning from highly skewed data distributions on top of a pre-trained model, where Catastrophic Inheritance-style failures can arise.

The visualization highlights the methods' resilience to data imbalance. While the representations learned by LoRA and PiSSA show degradation and increased class overlap (Figure 3(d,e)), BA-LoRA exhibits a better-separated manifold (Figure 3(f); see Table 12 in Appendix C.6 for quantitative

Table 4: Ablation study of BA-LoRA regularizations on GSM8K, MATH, and NLU tasks. Results on GSM8K and MATH are from LLaMA-2-7B, while the NLU column reports the average GLUE score from DeBERTa-v3-base. "Baseline" (PiSSA) is fine-tuned without our proposed regularizations. $+\mathcal{L}_{CR}$, $+\mathcal{L}_{DR}$, and $+\mathcal{L}_{SVDR}$ denote adding only the corresponding regularization term to the baseline. "BA-LoRA (Full)" is the full model using all regularizations.

| Configuration | GSM8K | MATH | GLUE Avg |
|---|---|---|---|
| Baseline (PiSSA) | $51.48 \pm 0.34$ | $7.60 \pm 0.18$ | 89.47 |
| $+ \mathcal{L}_{CR}$ | $54.25 \pm 0.59$ | $9.15 \pm 0.25$ | 90.18 |
| $+ \mathcal{L}_{DR}$ | $53.60 \pm 0.46$ | $8.95 \pm 0.18$ | 89.85 |
| $+ \mathcal{L}_{SVDR}$ | $52.95 \pm 0.55$ | $8.70 \pm 0.22$ | 89.71 |
| **BA-LoRA (Full)** | $55.86 \pm 0.35$ | $9.47 \pm 0.52$ | 90.67 |

verification). This visualization is consistent with the role of our diversity regularizer ($\mathcal{L}_{DR}$) in mitigating feature degradation under skewed data distributions. This effect is further supported by the consistency ($\mathcal{L}_{CR}$) and SVD ($\mathcal{L}_{SVDR}$) regularizers, which together help preserve distinct and robust representations.

### 3.2.4 ABLATION STUDY

Our ablation study (Table 4) empirically supports our principled deconstruction of Catastrophic Inheritance. On NLG tasks with the LLaMA-2-7B model, we observe a consistent pattern: each regularizer yields a positive contribution over the baseline ("Baseline"). Specifically, gains from the consistency regularizer ($\mathcal{L}_{CR}$) support its role in combating Knowledge Drift by preserving foundational knowledge. Similarly, improvements from the diversity regularizer ($\mathcal{L}_{DR}$) highlight the importance of mitigating Representation Collapse, and the contribution from the SVD regularizer ($\mathcal{L}_{SVDR}$) supports its benefit in mitigating Overfitting to Noise.

This trend is mirrored in NLU tasks, where the DeBERTa-v3-base model also shows a clear uplift with each regularizer over the baseline. The full BA-LoRA model, which combines all three components, consistently achieves the highest performance across all evaluated settings. In summary, these results provide evidence that Knowledge Drift, Representation Collapse, and Overfitting to Noise are important and complementary failure modes in fine-tuning. Consequently, our integrated, three-pronged solution yields strong generalization and robustness across both NLU and NLG domains. The selection of our regularization coefficients ($\lambda_1$, $\lambda_2$, $\lambda_3$) is further supported by a detailed sensitivity analysis in Figure 4, and Table 10 further shows that the same regularization framework consistently improves multiple LoRA-style methods (LoRA, DoRA, PiSSA).

### 3.2.5 COMPUTATIONAL COST ANALYSIS

To quantitatively evaluate the computational efficiency and performance of our method, we conduct a comparative experiment on two A40 GPUs (48 GB each) using DeepSpeed (Rasley et al., 2020) ZeRO-2 optimization. Full FT is OOM under the $2 \times$A40 ZeRO-2 setup, so we report lower bounds for its memory cost and training time. We fine-tune the LLaMA-2-7B model on the first 100,000 entries of the MetaMathQA dataset. This experiment benchmarks four methods: full fine-tuning (Full FT), LoRA, PiSSA, and our proposed BA-LoRA.

Table 5: Computational cost and performance comparison. Costs are measured on two A40 GPUs for fine-tuning LLaMA-2-7B. Memory cost reports the aggregated peak GPU memory across the two A40 GPUs. Full FT exceeds the available memory under this setup (OOM); thus, we report lower bounds ($> 96$ GB, $> 24$ h).

| Method | Memory Cost | Training Time | GSM8K |
|---|---|---|---|
| Full FT | $> 96$ GB | $> 24$ h | $48.9 \pm 0.49$ |
| LoRA | 66.32 GB | 4 h 31 min | $42.68 \pm 0.54$ |
| PiSSA | 66.59 GB | 4 h 17 min | $51.48 \pm 0.34$ |
| **BA-LoRA** | 77.34 GB | 4 h 48 min | $55.86 \pm 0.35$ |

For each method, we measure peak GPU memory consumption and total training time to assess computational cost. Model performance is subsequently evaluated on the GSM8K benchmark.

The results in Table 5 quantify the performance-cost trade-offs of these methods. BA-LoRA achieves the highest GSM8K score among the compared methods (55.86), outperforming all baselines. This performance gain is achieved with a moderate overhead compared to PiSSA (+10.75 GB memory, +31 min training), highlighting a favorable performance-cost balance.

## 4 RELATED WORK

Our work bridges two closely related research areas.

**Parameter-Efficient Fine-Tuning (PEFT).** Our method builds upon Low-Rank Adaptation (LoRA) (Hu et al., 2021). Numerous LoRA variants enhance performance and efficiency, such as QLoRA (Dettmers et al., 2024) and PiSSA (Meng et al., 2024), or address activation memory bottlenecks, such as LoRA-FA (Zhang et al., 2023a). However, empirical analyses reveal that constrained low-rank updates can affect the forgetting-plasticity trade-off and interfere with pre-trained knowledge (Magistri et al., 2024). Systematic mitigation of inherited biases within this restricted parameter space remains underexplored. Beyond classical PEFT, several recent works explicitly study forgetting and knowledge drift under parameter-efficient adaptation. For instance, Smith et al. introduce C-LoRA for continual customization of text-to-image diffusion models, mitigating drift by constraining adapter updates (Smith et al., 2023). Similarly, Chen and Garner propose Bayesian PEFT to reduce catastrophic forgetting via priors on adapters (Chen & Garner, 2024). However, these approaches mainly operate in parameter space and are evaluated on diffusion or continual-learning settings. In contrast, BA-LoRA targets Catastrophic Inheritance in language models (Chen et al., 2023; 2025) and uses three output-space regularizers, namely consistency, diversity, and SVD-based regularization, which are instantiated in a unified way for both NLU and NLG tasks.

**Bias Mitigation.** Biases from web-scale corpora are a foundational concern (Bender et al., 2021). While a rich literature exists on data filtering (Dodge et al., 2021) and algorithmic adjustments for full fine-tuning—such as representation debiasing (Ravfogel et al., 2020) and decoding strategies (Sheng et al., 2019) (see (Gallegos et al., 2024) for a survey)—these methods are not specifically designed for PEFT. Our design also relates to work analyzing and mitigating label noise in large-scale pre-training. Chen et al. propose a feature-space framework combining consistency, covariance, and dominant-singular-value regularization to improve robustness to noisy labels (Chen et al., 2023; 2025). Their regularizers act on intermediate representations, whereas BA-LoRA applies analogous ideas directly to the output logits of PEFT-adapted models. More broadly, our diversity regularizer is conceptually aligned with redundancy-reduction objectives such as Barlow Twins (Zbontar et al., 2021) and VICReg (Bardes et al., 2021), which encourage high variance and low cross-covariance to avoid representation collapse. Although recent analyses explore LoRA optimization dynamics, such as mitigating double descent (Chang et al., 2025), and fairness issues within PEFT (Ding et al., 2024), BA-LoRA provides a concrete, multi-component algorithmic framework for mitigating Catastrophic Inheritance in LoRA-based adaptation. It moves beyond analysis by integrating a principled output-space regularization scheme directly into the LoRA-based fine-tuning process.

## 5 CONCLUSION

This paper introduces BA-LoRA, a parameter-efficient fine-tuning framework aimed at mitigating Catastrophic Inheritance. Our core contribution is a principled approach that decomposes this challenge into three subproblems—Knowledge Drift, Representation Collapse, and Overfitting to Noise—and addresses them with three targeted regularizers. Extensive experiments across NLG and NLU benchmarks support our integrated strategy, which achieves strong performance and improves robustness to inherited data biases relative to standard LoRA variants. By explicitly addressing Catastrophic Inheritance within the LoRA framework, BA-LoRA provides a practical pathway for adapting pre-trained models to downstream tasks where robustness and fairness are important.

## ACKNOWLEDGMENTS

This work is supported by the National Key Research and Development Program of China (No.2023YFF0905400), the National Natural Science Foundation of China (No.U2341229) and the Reform Commission Foundation of Jilin Province (No.2024C003).

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

# Appendix

## CONTENTS

## A   BACKGROUND

### A.1   CHALLENGES OF BIAS AND NOISE IN PRE-TRAINING DATA

Bias and noise within pre-training datasets present major challenges for constructing dependable machine learning models. Mislabeled data and imbalanced distributions can cause models to un-

derperform on downstream tasks and reinforce existing biases (Barocas & Selbst, 2016; Gallegos et al., 2024). This issue is especially problematic in large-scale datasets where manual curation is impractical, and reliance on automated data collection may introduce various inaccuracies (Northcutt et al., 2021; Birhane & Prabhu, 2021). Consequently, models trained on such data risk poor generalization and the inheritance of data-induced flaws, which can be amplified during adaptation to downstream tasks (Frénay & Verleysen, 2013; Song et al., 2022). Recent work further extends Catastrophic Inheritance analysis to medical models, linking pre-training noise to structural collapse in feature and logit distributions (Sun et al., 2025). A key goal of fine-tuning is therefore to learn new capabilities while mitigating the effects of this "Catastrophic Inheritance".

## A.2 MITIGATING BIAS THROUGH PARAMETER-EFFICIENT FINE-TUNING

Parameter-efficient fine-tuning (PEFT) methods offer a promising foundation for mitigating Catastrophic Inheritance. By design, adapting models with minimal parameter updates may help limit overfitting to inherited noise and preserve foundational knowledge (Houlsby et al., 2019; Zaken et al., 2021; Lester et al., 2021). However, as we argue in the main paper, this promise is not fully realized by standard PEFT methods. Techniques like Low-Rank Adaptation (LoRA) (Hu et al., 2021) introduce their own inductive biases, such as the low-rank bottleneck, which can inadvertently exacerbate the very issues they are meant to solve by amplifying spurious correlations. This gap motivates the development of more principled, explicit regularization techniques, such as those proposed in our work, that are tailored to the unique challenges of the PEFT paradigm.

## A.3 TYPOLOGIES OF NOISE IN PRE-TRAINING DATA

The vast web-scale corpora used to train modern language models, such as LLaMA-2 (Touvron et al., 2023) and GPT-4 (OpenAI, 2023), often contain noise and distributional biases. The sheer scale of these datasets makes comprehensive manual curation impractical, meaning models are often exposed to duplicated, corrupted, or irrelevant information during pre-training (Elazar et al., 2023; Birhane & Prabhu, 2021). When fine-tuned, these models can struggle to distinguish signal from noise, which in turn degrades downstream performance. Understanding the specific typologies of this data-induced noise is therefore important for developing more robust models. We categorize the primary challenges as follows.

**Low-Quality Data**  This category stems from the uncurated nature of web data. A key issue is **data duplication**, where near-identical content can lead to model overfitting and privacy leakage risks (Carlini et al., 2022; Hernandez et al., 2022). Another challenge is **data corruption**, where inconsistent or erroneous inputs degrade model robustness and performance (Fan et al., 2024; Caswell et al., 2021). Furthermore, **test set contamination**, the leakage of evaluation data into the training corpus, can lead to inflated performance metrics and undermine the reliability of model evaluation (Roberts et al., 2023; Schaeffer, 2023).

**Distributional Skew**  This form of bias arises from non-uniform data distributions. A common form is **category imbalance**, where underrepresentation of certain topics or classes causes the model to perform poorly on those categories, leading to biased or unreliable outputs (Xu et al., 2023; Zhu et al., 2024; Parashar et al., 2024).

**Unsafe and Unethical Content**  Finally, web corpora often contain undesirable content. The presence of **toxic and harmful text**, including offensive, biased, or malicious content, can cause the model to generate inappropriate or harmful outputs, posing safety and ethical risks (Zou et al., 2023; Sun et al., 2024).

## B  EXPERIMENTAL SETUP

To evaluate our proposed method, we conduct experiments on a suite of Natural Language Generation (NLG) and Natural Language Understanding (NLU) tasks. Our experimental design, including models, datasets, and training configurations, is detailed below.

Table 6: Comparison of pre-training data and objectives for different language models.

| Model | Pre-training Data | Pre-training Objective |
|---|---|---|
| DeBERTa-v3-base (He et al., 2021) | Wikipedia, BooksCorpus, OpenWebText, CC-News, Stories | Replaced Token Detection with GDES |
| RoBERTa-base (Liu et al., 2019) | BooksCorpus, English Wikipedia, CC-News, OpenWebText, Stories | Masked Language Modeling |
| T5-base (Raffel et al., 2020) | Colossal Clean Crawled Corpus (C4) | Text-to-Text Denoising Objective |

Table 7: Evaluation metrics for the NLG datasets.

| Datasets | GSM8K | MATH | HumanEval | MBPP | MT-Bench |
|---|---|---|---|---|---|
| Metric | Accuracy | Accuracy | Pass@1 | Pass@1 | GPT-4 Evaluation |

## B.1 MODELS

Our evaluation uses a wide array of pre-trained language models. For NLG tasks, we primarily use large language models with strong generative capabilities, including LLaMA-2 (7B, 13B) (Touvron et al., 2023), LLaMA-3 (8B, 70B) (AI@Meta, 2024), Mistral-7B-v0.1 (Jiang et al., 2023), Mixtral-8x7B-v0.1 (Jiang et al., 2024), Gemma-7B (Team et al., 2024), Qwen-1.5-7B (Bai et al., 2023), Yi-1.5-34B (Young et al., 2024), and the Mixture-of-Experts model DeepSeek-MoE-16B (Dai et al., 2024).

For NLU tasks, our experiments employ several models to investigate different aspects of performance. Our main fine-tuning experiments on the GLUE benchmark use DeBERTa-v3-base (He et al., 2021). For the controlled study on pre-training data noise, we select RoBERTa-base (Liu et al., 2019) and T5-base (Raffel et al., 2020) due to their distinct corpus characteristics.

A detailed overview of the primary NLU models is presented in Table 6. These models provide a useful foundation for our study due to their diverse pre-training methodologies. For instance, RoBERTa-base was pre-trained on a relatively curated mixed corpus, whereas T5-base was pre-trained on the large-scale C4 web corpus. DeBERTa-v3-base uses a diverse pre-training corpus with a replaced token detection objective. This architectural and methodological diversity supports a broader evaluation of our approach.

## B.2 TASKS, DATASETS, AND METRICS

**Natural Language Generation (NLG)** For NLG, we assess model capabilities across mathematical reasoning, code generation, and instruction following. The benchmarks include GSM8K (Cobbe et al., 2021), MATH (Yu et al., 2023), HumanEval (Chen et al., 2021), MBPP (Austin et al., 2021), and MT-Bench (Zheng et al., 2024). As summarized in Table 7, evaluation metrics are task-specific: accuracy for GSM8K and MATH, Pass@1 for HumanEval and MBPP, and GPT-4-based evaluation for MT-Bench.

**Natural Language Understanding (NLU)** For our NLU evaluation, we use the GLUE benchmark (Wang et al., 2018), which comprises a diverse set of tasks. These tasks can be categorized into three groups: two single-sentence classification tasks (CoLA, SST-2), five pairwise text classification tasks (MNLI, RTE, QQP, MRPC, and QNLI), and one text similarity prediction task (STS-B). Following the standard evaluation protocol, we report the Matthews correlation coefficient for CoLA, Pearson correlation for STS-B, and accuracy for the remaining tasks. For MNLI, we follow the evaluation protocol used by our baselines.

## B.3 IMPLEMENTATION AND TRAINING DETAILS

**Baseline Comparison** For a fair and direct comparison, the baseline results presented in our main experiments are obtained from their original publications. Specifically, the NLG baseline results in Table 1 are sourced from the comprehensive study by Yang et al. (2025). For the NLU benchmarks, the results for DeBERTa-v3-base in Table 2 are taken from Kang & Yin (2025). For our BA-LoRA runs, we follow the same core fine-tuning configuration, including model, dataset splits, optimizer, learning-rate schedule, batch size, and LoRA rank and placements, as in these works. We only introduce our additional regularization terms, with coefficients chosen as described in Section 3.1 and Appendix B.4, to ensure a controlled evaluation.

For all GLUE experiments, the consistency and diversity regularizers are computed on the fine-tuning data by passing inputs through the pretrained backbone and the BA-LoRA-adapted backbone, using the same shared classification head, to generate teacher and student logits, respectively.

**MiLoRA vs. BA-LoRA.** MiLoRA proposes a spectral variant of LoRA that explicitly exploits *minor* singular components of pretrained weight matrices: instead of parameterizing adapters along the top singular directions, as in PiSSA, MiLoRA allocates capacity to lower-energy directions in weight space, arguing that these under-utilized components can be effective for adaptation while potentially reducing interference with core pretrained knowledge. In contrast, BA-LoRA keeps the underlying LoRA-style parameterization, such as standard LoRA, PiSSA, or DoRA, and operates entirely in *output space*: it introduces three regularizers on the logits—a consistency term to control knowledge drift, a diversity term to avoid representation collapse, and a spectral SVD term to suppress noisy high-frequency components. Thus, MiLoRA primarily changes *which spectral directions in weight space* are used to represent the adapters, whereas BA-LoRA constrains *how the adapted model behaves* through output-space regularization. These two perspectives are complementary and could, in principle, be combined.

**Data Preprocessing for Visualization** To analyze the model's feature space under data imbalance, we construct a custom imbalanced version of the MNLI training dataset. This process begins by separating the full training set into three subsets based on their labels. We then retain all samples from the `entailment` class (100%), while randomly downsampling the `neutral` class to 10% and the `contradiction` class to 1% of their original sizes. Finally, these three subsets are concatenated and shuffled to form the training set for the visualization model, thereby simulating a scenario with a highly skewed label distribution.

**t-SNE Visualization** For the t-SNE visualization, we fine-tune a RoBERTa-base model for 3 epochs on the imbalanced MNLI training set described above. We then extract the `[CLS]` token representations from the final hidden layer for all samples in the original balanced MNLI validation set. These high-dimensional features are projected into two dimensions using t-SNE with a perplexity of 30, 1000 iterations, and a fixed random seed (42) for reproducibility. The quality of the resulting clusters is also quantitatively assessed using the silhouette score with a cosine distance metric.

**Evaluation Frameworks** For evaluation, we use publicly available frameworks. Code-generation performance is assessed on HumanEval and MBPP using the BigCode Evaluation Harness[2]. Instruction-following performance is evaluated using MT-Bench[3].

### B.4 HYPERPARAMETER SETTINGS

**NLG (LLaMA-2-7B)** Our Natural Language Generation (NLG) experiments involve fine-tuning the LLaMA-2-7B model on a 100,000-sample subset of the MetaMathQA dataset. The model is trained for a single epoch using BFloat16 (bf16) precision, a maximum sequence length of 512, and an effective batch size of 32, achieved with a per-device batch size of 4 and 4 gradient accumulation steps. For optimization, we use the AdamW optimizer with a learning rate of $2 \times 10^{-5}$, no weight decay, and a cosine learning rate schedule with a 3% warm-up phase. The base LoRA configuration uses a rank ($r$) of 128, an alpha ($\alpha$) of 128, and no dropout, with adapters applied to the `q_proj`, `k_proj`, `v_proj`, `o_proj`, `gate_proj`, `up_proj`, and `down_proj` layers. For our proposed BA-LoRA method, we set the regularization coefficients to $\lambda_1 = 0.025$, $\lambda_2 = 0.005$, and $\lambda_3 = 0.005$. The primary coefficient $\lambda_1$ also follows a cosine schedule, while the lambda focus schedule is set to `two_phase` with a 0.2 warm-up and 0.05 ramp-up ratio. Additional parameters for the SVD-based components include an SVD rank (`svd_k`) of 10, an entropy top-k of 20, a distillation temperature of 2.0, and the use of the Frobenius norm for SVD normalization.

**NLU (GLUE Benchmark)** Our Natural Language Understanding (NLU) experiments on the GLUE benchmark involve three models, each with specific hyperparameter configurations as detailed below.

---

[2]`https://github.com/bigcode-project/bigcode-evaluation-harness`
[3]`https://github.com/lm-sys/FastChat`

Table 8: Fine-tuning hyperparameters for DeBERTa-v3-base on each GLUE task. The settings are aligned with the PiSSA baseline.

| Dataset | Epochs | Batch Size | Learning Rate | LoRA Alpha |
|---------|--------|------------|---------------|------------|
| MNLI | 5 | 16 | $5 \times 10^{-4}$ | 8 |
| SST-2 | 20 | 16 | $3 \times 10^{-5}$ | 8 |
| MRPC | 20 | 32 | $2 \times 10^{-4}$ | 8 |
| CoLA | 20 | 16 | $1 \times 10^{-4}$ | 8 |
| QNLI | 10 | 32 | $1 \times 10^{-4}$ | 16 |
| QQP | 10 | 16 | $1 \times 10^{-4}$ | 8 |
| RTE | 50 | 16 | $1 \times 10^{-4}$ | 8 |
| STS-B | 20 | 8 | $3 \times 10^{-4}$ | 8 |

Table 9: Task-specific hyperparameters for fine-tuning RoBERTa-base with LoRA on the GLUE benchmark.

| Hyperparameter | MNLI | SST-2 | MRPC | CoLA | QNLI | QQP | RTE | STS-B |
|----------------|------|-------|------|------|------|-----|-----|-------|
| Batch Size | 16 | 16 | 16 | 32 | 32 | 16 | 32 | 16 |
| # Epochs | 30 | 60 | 30 | 80 | 25 | 25 | 80 | 40 |
| Learning Rate | $5 \times 10^{-4}$ | $5 \times 10^{-4}$ | $4 \times 10^{-4}$ | $4 \times 10^{-4}$ | $4 \times 10^{-4}$ | $5 \times 10^{-4}$ | $4 \times 10^{-4}$ | $4 \times 10^{-4}$ |

DeBERTa-v3-base   We fine-tune DeBERTa-v3-base on the GLUE tasks using the AdamW optimizer with a linear learning rate schedule. To align with the PiSSA baseline, we adopt a set of task-specific hyperparameters. The LoRA rank ($r$) is consistently set to 8 across all tasks. Other key hyperparameters, including the number of epochs, batch size, learning rate, and LoRA alpha, are individually configured for each dataset. The precise configurations are detailed in Table 8. For encoder-only NLU models, such as DeBERTa-v3-base on GLUE, we attach a task-specific linear classification head and compute $\mathbf{Z}_P$ and $\mathbf{Z}_F$ by passing the same fine-tuning inputs through the pretrained teacher encoder and the BA-LoRA-adapted student model, respectively, using the shared classification head. This matches the notation used in Section 2.2.1.

T5-base   In our experiments with T5-base, we fine-tune all models for a single epoch using FP32 precision, a maximum sequence length of 128, and a batch size of 32. Optimization is performed with the AdamW optimizer (Loshchilov & Hutter, 2019) ($\beta_1 = 0.9$, $\beta_2 = 0.999$, $\epsilon = 1 \times 10^{-8}$, and no weight decay), coupled with a learning rate of $1 \times 10^{-4}$. The learning rate schedule incorporates a 3% warm-up phase followed by cosine decay. For the LoRA configuration, we set the rank ($r$) to 8 and alpha ($\alpha$) to 16, and apply LoRA to all linear modules except the embedding and language modeling head, while excluding layer-normalization layers.

RoBERTa-base   For fine-tuning RoBERTa-base on the GLUE benchmark, our setup follows standard practices for LoRA-based methods. We use the AdamW optimizer with a linear learning rate schedule, preceded by a warm-up phase over the first 6% of the total training steps. The LoRA configuration is kept consistent across all tasks: the rank ($r$) is set to 8 for the query ($q$) and value ($v$) projection matrices, and alpha ($\alpha$) is set to 8. The maximum sequence length is fixed at 512 tokens. Other key hyperparameters, including the number of epochs, batch size, and peak learning rate, are configured for each GLUE task. The precise per-task configurations are detailed in Table 9.

## C   More Experiments

### C.1   Generality of the Regularization Framework

To examine whether our regularization framework extends beyond PiSSA, we integrate it with standard LoRA and DoRA. The results, presented in Table 10, show that the framework can improve multiple LoRA-style methods. While our regularizers provide performance gains across all tested methods, their effect on standard LoRA is particularly notable: augmenting standard LoRA with our regularizers is sufficient to match and surpass the PiSSA baseline. This finding suggests that our regularization framework can serve as a broadly applicable enhancement for PEFT methods.

Table 10: Impact of the proposed regularization framework on various LoRA-style methods, evaluated on LLaMA-2-7B. "Reg" denotes the application of our three regularization terms. All results are averaged over 3 runs.

| Method | GSM8K | MATH | HumanEval | MBPP | MT-Bench | Avg |
|---|---|---|---|---|---|---|
| LoRA | $42.68 \pm 0.54$ | $5.92 \pm 0.15$ | $16.80 \pm 0.38$ | $21.51 \pm 0.43$ | $4.60 \pm 0.14$ | 18.30 |
| **LoRA + Reg** | $51.82 \pm 0.36$ | $8.69 \pm 0.39$ | $21.03 \pm 0.58$ | $33.81 \pm 0.51$ | $4.73 \pm 0.24$ | 24.02 |
| DoRA | $41.77 \pm 0.74$ | $6.20 \pm 0.48$ | $16.86 \pm 0.54$ | $21.60 \pm 0.49$ | $4.48 \pm 0.14$ | 18.18 |
| **DoRA + Reg** | $52.71 \pm 0.42$ | $8.23 \pm 0.27$ | $21.05 \pm 0.31$ | $34.78 \pm 0.28$ | $4.96 \pm 0.22$ | 24.35 |
| PiSSA | $51.48 \pm 0.34$ | $7.60 \pm 0.18$ | $19.48 \pm 0.45$ | $23.84 \pm 0.46$ | $4.92 \pm 0.07$ | 21.46 |
| **BA-LoRA (PiSSA + Reg)** | $55.86 \pm 0.35$ | $9.47 \pm 0.52$ | $23.58 \pm 0.25$ | $36.86 \pm 0.31$ | $5.11 \pm 0.05$ | 26.18 |

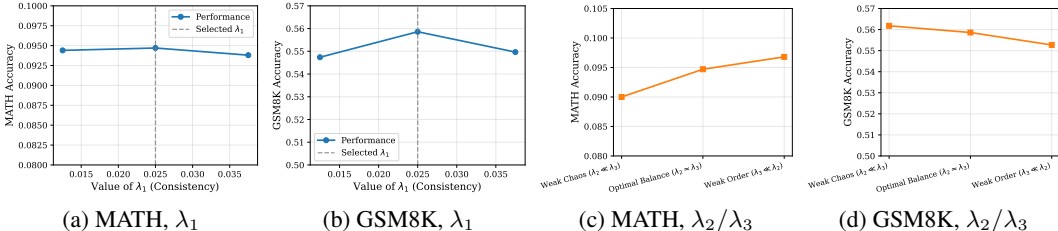

(a) MATH, $\lambda_1$     (b) GSM8K, $\lambda_1$     (c) MATH, $\lambda_2/\lambda_3$     (d) GSM8K, $\lambda_2/\lambda_3$

Figure 4: Sensitivity analysis of the core BA-LoRA regularization coefficients. Panels (a,b) show the effect of varying the consistency weight $\lambda_1$ on MATH and GSM8K, and panels (c,d) show the effect of changing the balance between $\lambda_2$ and $\lambda_3$.

The best overall results are achieved by our full BA-LoRA model. This suggests that PiSSA's principled initialization provides a strong foundation upon which our regularization framework can build, leading to the highest average performance in this comparison. These results support our integrated design for mitigating Catastrophic Inheritance and improving parameter-efficient adaptation.

## C.2 HYPERPARAMETER SENSITIVITY ANALYSIS

To examine the selection of our framework's hyperparameters, we conduct a sensitivity analysis. Centered around our final BA-LoRA configuration on LLaMA-2-7B, this study investigates the influence of the core regularization coefficients ($\lambda_1, \lambda_2, \lambda_3$) by perturbing them from their default values. The results, visualized in Figure 4, reveal a broad region of stable performance, supporting the robustness of our chosen configuration.

**Sensitivity to the Consistency Anchor ($\lambda_1$)** As illustrated in Figures 4(a,b), we vary $\lambda_1$ across $\{0.0125, 0.025, 0.0375\}$ while keeping $\lambda_2 = \lambda_3 = 0.005$. Performance on both MATH and GSM8K remains stable across this range, with only minor fluctuations, indicating a broad region of insensitivity. Our chosen value of $\lambda_1 = 0.025$, highlighted as the default point in the figure, provides a robust setting that balances preserving pre-trained knowledge with acquiring new task-specific capabilities.

**Sensitivity to the Balance between $\lambda_2$ and $\lambda_3$** Next, we investigate the balance between the other two regularizers, which govern a trade-off between performance on the two reasoning benchmarks (MATH and GSM8K). As shown in Figures 4(c,d), we compare our final configuration's balanced setting ($\lambda_2 \approx \lambda_3$) against two asymmetric conditions: a "Weak Chaos" setting ($\lambda_2 \ll \lambda_3$), where the structural regularizer ($\lambda_3$) dominates, and a "Weak Order" setting ($\lambda_3 \ll \lambda_2$), where the diversity regularizer ($\lambda_2$) is dominant. The results show a trade-off: disrupting the balance leads to improvements on one benchmark at the cost of the other, while the balanced configuration provides strong performance on both.

## C.3 ANALYSIS ACROSS DIVERSE MODEL ARCHITECTURES AND SCALES

To assess the generalizability and robustness of BA-LoRA, we compare it against LoRA and PiSSA across ten pre-trained models. This set includes models of varying scales, from LLaMA-2-7B

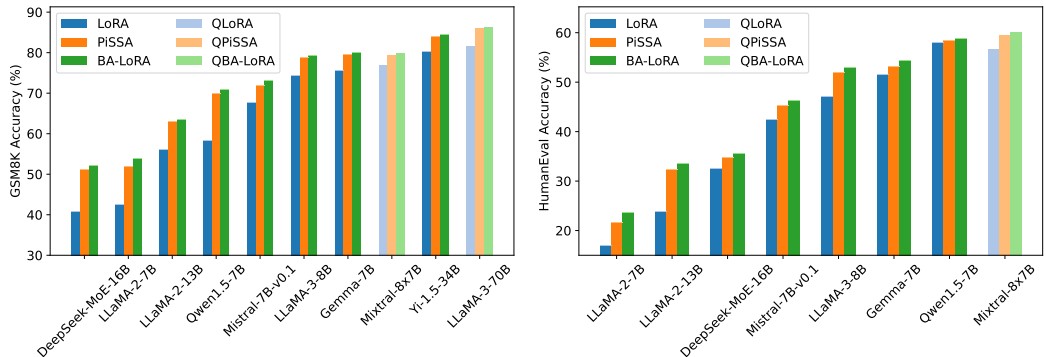

Figure 5: Performance comparison of different models on the GSM8K and HumanEval benchmarks.

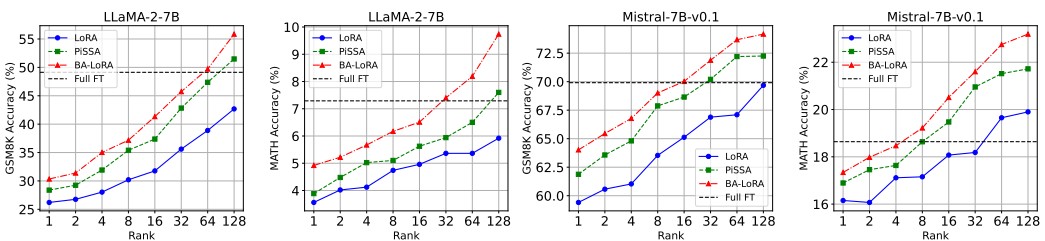

Figure 6: Performance comparison of full fine-tuning, LoRA, PiSSA, and BA-LoRA across different ranks.

to LLaMA-3-70B, and architectures, including both dense models and Mixture-of-Experts (MoE) models such as Mixtral-8x7B. All methods are fine-tuned on a blend of reasoning and code datasets (MetaMathQA-100K and CodeFeedback-100K) and evaluated on GSM8K and HumanEval.

As visualized in Figure 5, BA-LoRA typically achieves the best or near-best performance among LoRA-style methods on both benchmarks across most model families and scales. The gains over LoRA and PiSSA are especially pronounced on several mid- and large-scale models, suggesting that our regularization framework remains effective beyond the LLaMA-2-7B setting.

Furthermore, this performance advantage largely carries over to computation-constrained settings. The figure also plots the performance of 4-bit quantized versions of each method (QLoRA, QPiSSA, and QBA-LoRA). The overall trend is similar: QBA-LoRA generally matches or exceeds the other quantized baselines, indicating that the benefits of our framework are robust to quantization and remain useful for resource-efficient deployment.

### C.4 Performance Analysis Across Different Ranks

We analyze the performance of BA-LoRA, PiSSA, and LoRA across a range of ranks (from 1 to 128) on the LLaMA-2-7B and Mistral-7B-v0.1 models. Each method is fine-tuned for one epoch on the MetaMathQA-100K dataset and evaluated on GSM8K and MATH. The results, presented in Figure 6, show that BA-LoRA consistently performs strongly across ranks, models, and tasks, often outperforming both LoRA and PiSSA. Furthermore, both BA-LoRA and PiSSA can surpass the performance of full fine-tuning at higher ranks, with BA-LoRA often achieving this at relatively low ranks (e.g., rank 16–32). This suggests a strong regularization effect, as standard LoRA generally lags behind the full fine-tuning baseline. Moreover, the performance advantage of BA-LoRA over its counterparts is more pronounced on the Mistral-7B-v0.1 model, suggesting that its benefits can generalize across different foundation model architectures. These results support BA-LoRA as an efficient and effective fine-tuning method.

Table 11: Comparison of diversity regularization objectives on MNLI (DeBERTa-v3-base).

| Method | MNLI Accuracy |
|---|---|
| Standard LoRA | 90.71 |
| BA-LoRA (w/ Entropy Reg.) | 90.41 |
| **BA-LoRA (w/ Covariance Reg.)** | **91.26** |

Table 12: Quantitative analysis of feature separation and classification performance on the minority class under the MNLI imbalanced setting. Metrics are computed in the high-dimensional logit space.

| Method | Global Silhouette (All Classes) | Minority-Class Silhouette (Cluster Quality) | Minority-Class Recall (Accuracy (%)) |
|---|---|---|---|
| Standard LoRA | 0.207 | 0.015 | 5.8 |
| PiSSA | 0.247 | 0.128 | 26.4 |
| **BA-LoRA (Ours)** | **0.351** | **0.425** | **61.7** |

## C.5 COMPARISON OF REGULARIZATION OBJECTIVES FOR NLU

To evaluate our design choice of covariance regularization ($\mathcal{L}_{DR\_NLU}$) over entropy regularization for discriminative tasks, we conduct a comparative study on the MNLI benchmark. While entropy regularization is often used to encourage diversity in generation, applying entropy-based regularization to noisy or imbalanced NLU data may inadvertently encourage overconfident predictions on minority classes rather than effectively preventing representation collapse. The results presented in Table 11 support this hypothesis. Substituting our covariance-based regularizer with an entropy-based one results in performance (90.41%) that is lower than both our method (91.26%) and the standard LoRA baseline (90.71%). This finding suggests that batch-wise feature decorrelation is better suited for mitigating representation collapse in discriminative fine-tuning.

## C.6 QUANTITATIVE ANALYSIS OF CLUSTER QUALITY AND MINORITY REPRESENTATION

To complement the visual analysis provided in Figure 3 in the main text, we conduct a quantitative evaluation of feature-space separation under the imbalanced fine-tuning setting on MNLI. We compute metrics in the original high-dimensional logit space before t-SNE projection to avoid dimensionality-reduction artifacts. We report the **Global Silhouette Score** over all classes, the **Minority-Class Silhouette Score** measuring the isolation of the minority cluster, and the **Minority-Class Recall** measuring classification accuracy on the minority class.

Table 12 presents the results. Standard LoRA exhibits a near-zero minority silhouette score (0.015) and a low minority recall (5.8), quantitatively confirming the representation collapse observed visually. In contrast, BA-LoRA achieves a higher minority silhouette score (0.425) and restores the minority recall to 61.7. These results show that BA-LoRA mitigates the inheritance of pre-trained priors under imbalanced fine-tuning, enabling the model to establish a more distinct decision boundary for the minority class.

## C.7 DIRECT EVALUATION OF PRE-TRAINED CAPABILITIES

To directly assess how different adaptation schemes affect the underlying pre-trained model, we measure perplexity on the *Wikitext-2* (Merity et al., 2016) language modeling benchmark. We evaluate three variants of LLaMA-2-7B—the original pre-trained model, a PiSSA-tuned model, and our BA-LoRA model—using the same backbone as in the main math-reasoning experiments and standard left-to-right evaluation on the Wikitext-2 test set. As shown in Table 13, the pre-trained LLaMA-2-7B achieves a perplexity of 5.42, while BA-LoRA attains 5.68, indicating only a small degradation relative to the pre-trained model. In contrast, the PiSSA baseline reaches 7.12, suggesting a more pronounced shift away from the original pre-training distribution. Together with the downstream results, this supports the view that BA-LoRA preserves the pre-trained model's behavior on natural text better than a pure PiSSA-based adaptation.

Table 13: Wikitext-2 test perplexity (lower is better) for LLaMA-2-7B under different adaptation schemes.

| Method | Wikitext-2 PPL $\downarrow$ |
|---|---|
| Pre-trained LLaMA-2-7B | 5.42 |
| **BA-LoRA (Ours)** | 5.68 |
| PiSSA | 7.12 |

## C.8 Asymptotic Complexity of Regularizers

To complement the empirical wall-clock measurements in Table 5, we summarize the asymptotic cost of each regularizer in BA-LoRA for both NLU and NLG settings. Table 14 reports the additional computational complexity on top of the standard forward pass and task loss, expressed in terms of the batch size $N$, number of classes $D$ for NLU, number of valid tokens $M$ for NLG, vocabulary size $|\mathcal{V}|$, Top-$K$ candidate set size $K$, and SVD rank $k$ used in the randomized approximation.

Table 14: Additional asymptotic complexity of BA-LoRA regularizers on top of the standard forward pass and task loss. Here $N$ is the NLU batch size, $D$ is the number of classes, $M$ is the number of valid tokens in NLG, $|\mathcal{V}|$ is the vocabulary size, $K$ is the Top-$K$ subset size, and $k$ is the rank used in randomized SVD.

| Regularizer | NLU (logits $Z \in \mathbb{R}^{N \times D}$) | NLG (logits $Z \in \mathbb{R}^{M \times |\mathcal{V}|}$) |
|---|---|---|
| Consistency | $O(ND)$ (batch KL over $D$ classes) | $O(M|\mathcal{V}|)$ (token-wise KL over the full vocabulary) |
| Diversity | $O(ND^2)$ (batch covariance on $D$-dimensional logits) | $O(M|\mathcal{V}| + MK)$ (Top-$K$ selection and entropy over $\mathcal{V}_{\text{top-}K}$, $K \ll |\mathcal{V}|$) |
| SVD-based | $O(\min(N,D)^2 \max(N,D))$ (exact SVD of $Z$) | $O(M|\mathcal{V}|k)$ (randomized SVD of $Z$ with rank $k$) |

# D More Discussions

Here, we provide further discussion of our method and experimental findings.

## D.1 Practical Hyperparameter Guidelines

For a new model–task configuration, BA-LoRA can be tuned with a simple recipe without fragile fine-grained search. First, choose initial values for $\lambda_{\text{consistency}}$, $\lambda_{\text{diversity}}$, and $\lambda_{\text{svd}}$ such that, on a small calibration batch, each regularization term contributes a meaningful fraction to the total loss, with gradient magnitudes roughly commensurate with the task loss. This prevents any component from being inactive or overwhelmingly dominant. Second, keep the ratios between the three coefficients fixed and tune a single global scale $\lambda_{\text{global}}$ that multiplies all of them, i.e., $\lambda_i \leftarrow \lambda_{\text{global}} \lambda_i$ for $i \in \{\text{consistency}, \text{diversity}, \text{svd}\}$; in our experience, a coarse search over a few values (e.g., $\{0.5, 1.0, 2.0\}$) on a held-out validation split is sufficient. Third, if needed, individual coefficients can be adjusted by coarse factors, such as $\times 0.5$ or $\times 2$, to emphasize preserving pretrained knowledge (larger $\lambda_{\text{consistency}}$), avoiding collapse (larger $\lambda_{\text{diversity}}$), or suppressing noisy high-frequency components (larger $\lambda_{\text{svd}}$).

As shown by the ablation study in Section 3.2.4 and the sensitivity analysis in Appendix C.2, BA-LoRA achieves stable performance around these configurations and consistently outperforms vanilla LoRA and PiSSA in this region, suggesting that this lightweight procedure is sufficient in practice. Moreover, for each backbone, we fix a single configuration of the auxiliary hyperparameters $(s, T, K, k)$ based on a small validation experiment and reuse it across all datasets, avoiding per-benchmark tuning. Given the stability observed in our preliminary experiments and to maintain efficiency, we use these as reasonable per-backbone defaults rather than performing an exhaustive, computationally expensive sensitivity sweep.

## D.2 Additional Discussion on RoBERTa vs. T5

Section 3.2.2 uses RoBERTa-base and T5-base as a natural comparison for probing robustness to inherited noise, since RoBERTa is trained on a relatively curated mixture of corpora while T5 is pretrained primarily on the C4 web corpus. However, these models also differ in architecture

(encoder-only vs. encoder–decoder) and pre-training objective (masked LM vs. text-to-text denoising). As a result, the larger BA-LoRA gains we observe on T5 should be interpreted as evidence compatible with our noisy-pretraining hypothesis, rather than as a fully controlled causal test. A more definitive analysis would require models with closely matched architectures and objectives but systematically varied pre-training corpora, which we leave for future work.

## D.3 DISCUSSION ON THE CHOICE OF REGULARIZATION TARGETS

A key design choice in BA-LoRA is the application of regularization terms in the model's output space, i.e., on logits and their derived distributions, rather than directly on the trainable adapter parameters ($A$ and $B$). This section provides further justification for this design choice.

Regularizing the low-rank adapter weights directly, for instance by penalizing the norm of $A$ or $B$, is a viable alternative. However, this approach presents a challenge: the mapping from the low-dimensional parameter space of the adapters to the high-dimensional functional space of the model's final output is highly complex and non-linear. Consequently, a simple constraint on the adapter weights, such as a small norm, does not guarantee the desired functional behavior, such as output diversity or consistency with the pre-trained model. The effect of such parameter-space regularization on the final model output can therefore be difficult to control.

In contrast, applying regularization directly in the output space offers a more direct and interpretable path to achieving our goals. By penalizing undesirable properties in the output logits or probability distributions—such as their deviation from the pre-trained model (Knowledge Drift), their lack of diversity (Representation Collapse), or their over-reliance on non-robust features (Overfitting to Noise)—we explicitly constrain the model's final behavior. This aligns the optimization objective with the goal of mitigating the functional consequences of Catastrophic Inheritance. The consistent performance of our framework across diverse models, tasks, and ranks provides empirical support for this output-space regularization strategy.

## D.4 CONCEPTUAL FOUNDATIONS AND SYNERGY OF REGULARIZERS

The three regularization terms proposed in BA-LoRA—consistency, diversity, and SVD-based regularization—are motivated by established principles in the machine learning literature for improving model robustness and generalization, and they are designed to work together.

**Origins.** The **Consistency Regularizer**, implemented as a KLD-based distillation loss in our experiments, is a form of knowledge distillation (Hinton et al., 2015), specifically self-distillation, where the pre-trained model acts as the teacher. The **Diversity Regularizer** is rooted in principles from representation learning and information theory. The covariance-based term for NLU is inspired by methods that combat representation collapse in self-supervised learning (Bardes et al., 2021), while the entropy-based term for NLG is a classic technique to prevent mode collapse and improve diversity in generative models (Cover, 1999). Finally, the **SVD Regularizer** builds upon the principle of spectral regularization, where the singular value spectrum of a weight or feature matrix is constrained to improve generalization. The insight that dominant singular values capture robust data patterns is a recurring theme in robust machine learning and transfer learning (Chen et al., 2019).

Throughout our experiments, we follow the standard temperature-scaled distillation convention (Hinton et al., 2015) and multiply the KL-based distillation loss by $T^2$. Dividing logits by $T$ scales the gradients of the KL divergence approximately by $1/T^2$; this factor therefore keeps the effective gradient norm of the consistency loss roughly invariant to $T$, so that changing $T$ primarily controls the softness of the teacher distribution rather than unintentionally reweighting the regularizer.

**Synergy.** While each regularizer addresses a distinct failure mode, their combination creates a complementary effect. For instance, solely enforcing consistency ($\mathcal{L}_{CR}$) might excessively constrain the model, preventing it from fully adapting to the downstream task. However, when combined with the diversity regularizer ($\mathcal{L}_{DR}$), the model is encouraged to explore new, diverse representations within the bounds of the pre-trained knowledge. Similarly, the SVD regularizer ($\mathcal{L}_{SVDR}$) helps ensure that the diverse representations learned are also more robust, reducing the risk of learning spurious correlations encouraged by a simple diversity objective. Our ablation study (Section 3.2.4)

empirically supports this complementary design, showing that the full BA-LoRA model achieves the best performance among the evaluated configurations.

## D.5   ON APPLYING REPRESENTATION LEARNING PRINCIPLES DURING FINE-TUNING

A key consideration for our work is whether incorporating principles from self-supervised learning (SSL), such as our diversity regularizer, during fine-tuning could disrupt the model's pre-trained representations. We argue that our framework mitigates this risk through two mechanisms.

First, the PEFT paradigm, specifically LoRA, limits the scope of parameter updates. With the vast majority of parameters frozen, the model's core representational geometry remains anchored. Our regularizers guide only the small perturbations introduced by the low-rank adapters, encouraging these updates to refine rather than overwrite foundational knowledge.

Second, our regularization scheme is complementary. The consistency regularizer ($\mathcal{L}_{\mathrm{CR}}$) acts as a counterweight to the diversity regularizer ($\mathcal{L}_{\mathrm{DR}}$). While $\mathcal{L}_{\mathrm{DR}}$ encourages adaptation and prevents representation collapse on the downstream task, $\mathcal{L}_{\mathrm{CR}}$ encourages this adaptation to remain close to the pre-trained model's knowledge manifold. This calibrated balance allows BA-LoRA to enhance task-specific performance while reducing the risk of catastrophic forgetting.

## D.6   LIMITATIONS AND FUTURE WORK

While this study demonstrates the effectiveness of BA-LoRA, there are areas for future research. Our empirical evaluation primarily focuses on English-language benchmarks; future work should extend this evaluation to multilingual settings and specialized domains to assess the broader applicability of the method. Moreover, our analysis of noisy pre-training in Section 3.2.2 relies on RoBERTa-base and T5-base checkpoints, which differ in both architecture and **pre-training objective**. These results should be interpreted as suggestive rather than controlled evidence about pre-training noise, and a more definitive study with a fixed architecture pre-trained under systematically varied noise levels is an important direction for future work. In addition, while BA-LoRA's regularization components have shown strong promise, task-specific adaptations could further improve their performance across a wider range of applications, and exploring such adjustments may be valuable for enhancing robustness and adaptability in diverse use cases.

## D.7   ETHICS STATEMENT

This study aims to develop and evaluate BA-LoRA, a parameter-efficient fine-tuning method designed to mitigate bias and enhance the performance of LLMs. By aiming to create more robust and less biased models, a primary ethical motivation of this work is to contribute to safer and more reliable AI systems. Our research uses existing open-source public datasets for both fine-tuning and evaluation, including MetaMathQA, CodeFeedback, WizardLM-Evol-Instruct, and the GLUE benchmark.

While these datasets are widely recognized and adopted within the research community, we acknowledge that large-scale corpora and LLM-generated instructions may contain inherent societal biases or noisy artifacts. Rather than ignoring these issues, BA-LoRA is designed to reduce the exacerbation of such underlying biases during model adaptation. Furthermore, we recognize that while parameter-efficient methods democratize the fine-tuning of LLMs, they also lower the barrier for potential malicious adaptation (dual-use risk). By constraining the output space to prevent representation collapse and Catastrophic Inheritance, our proposed regularizers represent a step toward keeping adapted models closer to their robust, pre-trained behaviors. We remain committed to the responsible development and transparent reporting of AI technologies.

## D.8   REPRODUCIBILITY

To support the reproducibility of our results, we provide a detailed description of our experimental setup in Section 3.1 and Appendix B, including model descriptions, dataset descriptions, hyperparameter configurations, and evaluation procedures. All models and datasets used are publicly available. In addition, we have refined the implementation scripts and fine-tuning strategies to facilitate independent verification. To further facilitate reproducibility, our source code, including scripts to replicate the main experiments, is publicly available at `https://github.com/llm172/BA-LoRA`.

USE OF LARGE LANGUAGE MODELS

In the preparation of this manuscript, a large language model (LLM) was used as a writing assistant. The LLM's role was limited to improving the clarity, conciseness, and grammatical correctness of the text. Specifically, it was used for tasks such as rephrasing sentences, suggesting alternative vocabulary, and checking for stylistic consistency. All core scientific ideas, experimental designs, data analyses, and conclusions were conceived and formulated by the human authors.

