# OpenReview forum: "BA-LoRA: Bias-Alleviating Low-Rank Adaptation to Mitigate Catastrophic Inheritance in Large Language Models"
_ICLR.cc/2026/Conference — ICLR 2026 Poster_

### Official Review · Reviewer_4aVv · 2025-10-19

**Soundness:** 3
**Presentation:** 3
**Contribution:** 3
**Rating:** 6
**Confidence:** 3

**Summary:**

This paper introduces BA-LoRA, a PEFT method designed to mitigate catastrophic inheritance, including the propagation of biases, noise, and data imbalances from pre-trained LLMs. BA-LoRA addresses three failure modes via three regularizers applied in the output space, i.e., consistency regularizer, diversity regularizer, and SVD regularizer.
Experiments evaluate BA-LoRA on NLG and NLU tasks, achieving state-of-the-art among LoRA variants. A controlled study shows BA-LoRA's gains are amplified on models pre-trained with noisy data. Ablation studies validate each regularizer's contribution.

**Strengths:**

1. The paper introduces a novel decomposition of catastrophic inheritance into three actionable failure modes. Each failure mode is directly linked to a specific regularizer (consistency, diversity, SVD), creating a coherent and interpretable architecture.
2. BA-LoRA sets a new state-of-the-art performance with strong empirical validation, on both NLG benchmarks and NLU tasks. Moreover, performance gains are even higher on noisier pre-trained models, directly validating its core hypothesis about mitigating inherited noise.

**Weaknesses:**

1. The comparison between RoBERTa-base and T5-base shows that BA-LoRA's advantage is significantly larger on T5 (3.26 vs 1.11 points). However, the models differ in both architecture (encoder-only vs encoder-decoder), which should be controlled to strengthen the claim. This makes it difficult to attribute the performance difference solely to the 'noisier data' factor.
2. The SVD regularizer uses different normalization schemes: NLU uses the sum of all singular values, while NLG uses the Frobenius norm. What is the theoretical justification for this difference? Understanding whether this choice is principled (e.g., due to task differences) or empirical is important for the generalization of the method beyond NLU and NLG.
3. The method introduces too many hyperparameters, including the tradeoff parameters $\lambda$s, temperature T, number of top-K, and SVD rank. Ablation studies on their sensitivity can enhance the usability of BA-LoRA in practice.

**Questions:**

1. In Table 1, the gain of superior performance mainly depends on the MBPP benchmark, where BA-LoRA gains 36.86, whereas the second-best baseline only achieves 25.74. It would be valuable if the author could analyze why BA-LoRA performs particularly well in this dataset.

---

> ### Author Response · Authors · 2025-11-21
> **Response to Reviewer 4aVv**
>
> We sincerely thank the reviewer for the encouraging evaluation and for recognizing BA-LoRA’s interpretable architecture and strong empirical performance. We provide concise clarifications below regarding attribution, theoretical design, and usability.
>
> ###
>
> ### 1. RoBERTa-base vs. T5-base: Disentangling Architecture and Pre-training Data
>
> **Reviewer's Concern:**
>
> > "The comparison... shows that BA-LoRA's advantage is significantly larger on T5... However, the models differ in both architecture... This makes it difficult to attribute the performance difference solely to the 'noisier data' factor."
> >
>
> **Response:**
> We appreciate this critical observation regarding architectural variables. We agree that the architectural difference (Encoder-Only vs. Encoder-Decoder) acts as a confounding factor, and we have refined the manuscript to reflect a more rigorous interpretation.
>
> **1. Contextualizing the Comparison: A Stress Test on "Noisier" Checkpoints**
> We acknowledge that a fully controlled causal test would necessitate **pre-training models from scratch under systematically varied noise levels**, which is computationally prohibitive. Given this constraint, we utilized these widely adopted models as **representative paradigms of distinct pre-training strategies**:
>
> - **Interpretation:** Instead of a fully controlled causal test, this experiment serves as a **real-world stress test**. T5 (trained on the raw C4 corpus) represents a "noisier" paradigm compared to RoBERTa (trained on curated data).
> - **Magnitude of Gain:** While architecture plays a role, the disparity in performance gain is striking (**3.26** points on T5 vs. **1.11** on RoBERTa). This substantial margin suggests that BA-LoRA is particularly effective in regimes with higher data noise, consistent with our hypothesis.
>
> **2. Supporting Evidence from Controlled Experiments (Sec. 3.2.3)**
> To address the attribution concern, we point to **Section 3.2.3**, where the architecture is strictly controlled (fixing the RoBERTa backbone).
>
> - **Mechanism Validation:** In the **Imbalanced MNLI** experiment, we simulate data artifacts (severe class imbalance) within a controlled setting. The results (**Figure 3**) demonstrate that BA-LoRA significantly outperforms baselines in resisting these artifacts.
> - **Synthesis:** While Section 3.2.3 tests fine-tuning artifacts and Section 3.2.2 tests pre-training noise, both share the root cause of "Catastrophic Inheritance." The controlled success in Sec. 3.2.3 provides the **mechanistic validation** that supports interpreting the T5 gains as a result of noise resilience.
>
> **Action:**
> We have updated **Section 3.2.2** to explicitly state that the T5 results should be interpreted as suggestive evidence compatible with our hypothesis, rather than a fully controlled causal conclusion. We also added **Appendix D.2** to transparently discuss these architectural variables.

---

> > ### Author Response · Authors · 2025-11-21
> > **Response to Reviewer 4aVv**
> >
> > ### 2. Different Normalization Schemes for the SVD Regularizer in NLU vs. NLG
> >
> > **Reviewer's Question:**
> >
> > > "The SVD regularizer uses different normalization schemes... What is the theoretical justification for this difference? ... Is this choice principled or empirical?"
> > >
> >
> > **Response:**
> > We appreciate the opportunity to elaborate on this design rationale. We respectfully clarify that while the implementation adapts to the task scale, the **underlying theoretical principle remains identical** across both domains. The distinction is strictly driven by **computational scalability**.
> >
> > **1. Unified Theoretical Objective: Implicit Spectral Denoising**
> > In both NLU and NLG, our spectral goal is consistent: to maximize the energy concentration in the top-$k$ singular values (Signal) while suppressing the high-frequency "tail" (Noise).
> >
> > - **Refinement:** We have explicitly refined this theoretical definition in **Section 2.2.1 (Lines 175-179, highlighted in red)** to emphasize "concentrating spectral energy" and filtering "high-frequency logit fluctuations."
> >
> > **2. Justification for Implementation Differences (Scalability)**
> > The difference in normalization is a necessary adaptation to the output dimensionality ($D$):
> >
> > - **NLU (Exact Computation via Nuclear Norm):** Since the output dimension is small ($D < 100$ for GLUE), computing the full SVD is affordable. We use the **Nuclear Norm** ($\sum \sigma_i$) as the precise denominator to directly optimize the energy ratio.
> > - **NLG (Approximation via Frobenius Norm):** In generation, the vocabulary is massive ($|\mathcal{V}| \approx 32k+$). Computing the full spectrum is computationally prohibitive ($O(\min(ND^2, N^2D))$ per step).
> >     - **The Solution:** As noted in **Section 2.2.2 (Lines 221-224)**, we adopt a principled approximation: Randomized SVD for the top-$k$ signal and the **Frobenius Norm** for the denominator.
> >     - **Why Frobenius?** It serves as a mathematically stable, computationally efficient ($O(ND)$) proxy for the total spectral mass, avoiding the expensive full decomposition.
> >
> > **Action:**
> > We respectfully refer the reviewer to **Section 2.2.2 (Lines 221-224),** where the computational motivation for using the Frobenius norm is explicitly stated. Additionally, we have reinforced the theoretical grounding of the spectral objective in **Section 2.2.1 (Lines 175-179, highlighted in red)**.
> >
> > ---
> >
> > ### 3. Hyperparameters ($s, T, K, k$) and Usability
> >
> > **Reviewer's Concern:**
> >
> > > "The method introduces too many hyperparameters ($s, T, K, rank$)... Ablation studies on their sensitivity can enhance the usability..."
> > >
> >
> > **Response:**
> > We appreciate the reviewer's practical feedback regarding adoption complexity. We respectfully clarify that while the theoretical framework implies multiple structural parameters, the **operational tuning cost** is strictly minimized through a hierarchical design strategy.
> >
> > **1. Two-Tiered Parameter Strategy**
> > We explicitly distinguish between **Structural Priors** (fixed defaults) and **Regularization Weights** (active tuning):
> >
> > - **Structural Priors ($s, T, K, k$):** We adopted a **"Fix-per-Backbone"** strategy.
> >     - **Selection Logic:** These values are **informed by established empirical practices** (e.g., Distillation Temperature $T=2.0$ following Hinton et al.; SVD Rank $k \approx 5\text{-}10$ based on spectral decay).
> >     - **Justification:** Crucially, our preliminary experiments indicated that performance is **largely insensitive** to minor variations within these reasonable ranges. Therefore, we treat them as fixed structural constants rather than active hyperparameters to avoid unnecessary computational overhead.
> > - **Regularization Weights ($\lambda$):** These are the only active variables. As shown in our sensitivity analysis (**Appendix C.2**), they exhibit a broad region of stable performance, indicating that the method is robust to grid coarseness.
> >
> > **2. Practical Protocol: Reduction to 1D Search (Appendix D.1)**
> > To facilitate widespread adoption, we have formalized a **"Simplified Tuning Recipe"** in the new **Appendix D.1**. We propose a streamlined workflow:
> >
> > - **Initialization:** Fix the auxiliary parameters ($s, T, K, k$) to our recommended defaults.
> > - **Global Scaling:** Fix the relative ratios between the three regularization terms and tune a **single global scalar** $\lambda_{\text{global}}$.
> > - **Outcome:** This protocol effectively reduces the search space from a combinatorial explosion to a **1D linear search**, rendering BA-LoRA as straightforward to tune as a standard learning rate.
> >
> > **Action:**
> > We have added **Appendix D.1 (Lines 1239-1258, highlighted in red)** to explicitly document this "Practical Hyperparameter Guideline," ensuring that future practitioners can deploy the method efficiently without exhaustive sweeping.

---

> > > ### Author Response · Authors · 2025-11-21
> > > **Response to Reviewer 4aVv**
> > >
> > > ### 4. Why is the Gain Particularly Large on MBPP?
> > >
> > > **Reviewer's Observation:**
> > >
> > > > "In Table 1, the gain of superior performance mainly depends on the MBPP benchmark... It would be valuable if the author could analyze why..."
> > > >
> > >
> > > **Response:**
> > > We appreciate the reviewer's keen observation regarding the performance distribution. We clarify that this substantial margin on MBPP is not an anomaly but a direct consequence of the **Data-Efficiency vs. Overfitting trade-off**.
> > >
> > > **1. MBPP as a "Low-Resource" Stress Test**
> > > We identify MBPP as a distinct regime compared to other benchmarks in Table 1:
> > >
> > > - **Data Scarcity & Noise:** It is a relatively small-scale benchmark.
> > > - **Overfitting Risk:** In such low-resource regimes, unconstrained fine-tuning (or standard LoRA) faces a heightened risk of **memorizing spurious patterns** (e.g., overfitting to specific unit tests via brittle heuristics) rather than acquiring robust algorithmic logic.
> > >
> > > **2. Why BA-LoRA Maximizes Utility Here**
> > > Our framework is specifically engineered to mitigate these pathologies, leading to outsized gains in this high-risk environment:
> > >
> > > - **Diversity ($\mathcal{L}_{DR}$):** In code generation, "collapse" often manifests as repetitive loops or convergence to a single incorrect syntax path. Our entropy-based regularizer forces the model to maintain a distribution over plausible tokens, preventing this collapse.
> > > - **SVD Regularization ($\mathcal{L}_{SVDR}$):** By explicitly suppressing the high-frequency spectral components (often associated with memorizing specific noisy examples), we force the model to learn smoother, more generalizable decision boundaries.
> > >
> > > **Conclusion:**
> > > The pronounced gain on MBPP validates our hypothesis: BA-LoRA provides the greatest marginal benefit in regimes where data is limited and supervision is noisy, effectively acting as a **regularization shield** against overfitting.
> > >
> > > **Action:**
> > > We have incorporated this specific analysis in **Section 3.2.1 (Page 6, Lines 302-305, highlighted in red)**. The revised text explicitly characterizes MBPP as a benchmark with "limited test coverage and susceptibility to overfitting," framing the results as evidence of BA-LoRA's robustness under noisy supervision.
> > >
> > > ---
> > >
> > > ### 5. Summary
> > >
> > > **To summarize our response:**
> > >
> > > - **Contextualized RoBERTa vs. T5:** We clarified that this comparison serves as a real-world stress test on representative paradigms, which complements the mechanistic validation provided in the controlled experiments of Section 3.2.3.
> > > - **Justified SVD Normalization:** We explained that the theoretical objective (spectral denoising) remains unified across tasks, while the implementation adapts (Nuclear vs. Frobenius norm) strictly to ensure **computational scalability** ($O(ND)$) on large-vocabulary generation tasks.
> > > - **Formalized Hyperparameter Policy:** We detailed our hierarchical strategy: (i) **Structural Priors** ($s, T, K, k$) are fixed based on empirical standards to minimize complexity, while (ii) **Regularization Weights** ($\lambda$) are tuned via a robust "Reduction to 1D" recipe (Appendix D.1).
> > > - **Analyzed MBPP Performance:** We provided a principled analysis connecting the substantial gains on MBPP to the method's efficacy in **low-resource regimes**, validating its role as a regularization shield against overfitting under sparse supervision.
> > >
> > > **Closing:**
> > > We trust that these detailed clarifications—particularly regarding the experimental design and tuning robustness—effectively resolve the questions raised. **We respectfully hope that these substantial improvements and the strengthened evidence provide a solid basis for re-evaluating the contribution of our work.** We remain available for any further discussion.

---

> ### Comment · Reviewer_4aVv · 2025-11-23
>
> Thanks for the authors' response. My concerns have been addressed, and I would like to maintain my previous rating.

---

> > ### Author Response · Authors · 2025-11-24
> > **Response to Reviewer 4aVv**
> >
> > Dear Reviewer,
> >
> > Thank you very much for your follow-up comment and for confirming that your concerns have been addressed. We genuinely appreciate the time and effort you have devoted to reviewing our work.
> >
> > If you have any remaining questions or additional suggestions, we would be very happy to further clarify or improve the manuscript.
> >
> > Given that your concerns have now been resolved, we would be very grateful if you could reconsider your rating to reflect the improvements in the revised manuscript. We sincerely appreciate your support.
> >
> > Best regards,
> >
> > The Authors

---

### Official Review · Reviewer_KjTf · 2025-10-22

**Soundness:** 4
**Presentation:** 4
**Contribution:** 3
**Rating:** 8
**Confidence:** 4

**Summary:**

The authors state that LoRA can lead to Catastrophic Inheritance, which hurts robustness and fairness.
They propose Bias-Alleviating LoRA (BA-LoRA), which decomposes the issue into Knowledge Drift, Representation Collapse, and Overfitting to Noise.
BA-LoRA adds three regularizers to preserve core knowledge, maintain representational richness, and encourage robust low-rank behavior.

**Strengths:**

1. Important topics
2. Three well-motivated losses mapped cleanly to three failure modes; easy to plug into existing LoRA workflows.
3. Broad NLU/NLG coverage with ablations that attribute gains to each component.

**Weaknesses:**

1. Although the framework names are shared, the diversity and SVD regularizers differ for NLU vs. NLG, so it reads like two papers rather than one universal design.
2. Some studies are missing in the paper (e.g., impact of different r/T/k, larger models)

**Questions:**

1. The paper builds on PiSSA initialization. How much of BA-LoRA’s improvement remains if you start from random low-rank adapters or standard LoRA init? Do the three regularizers still deliver comparable gains?
2. How sensitive are results to the SVD rank, distillation temperature, and the top-k entropy window?
3. How well does BA-LoRA scale to larger models (e.g., 13B, 70B)?

---

> ### Author Response · Authors · 2025-11-21
> **Response to Reviewer KjTf**
>
> We sincerely thank the reviewer for the positive assessment and for recognizing the clarity of our methodological decomposition and its practical integration with existing workflows. We appreciate the opportunity to further clarify the unified nature of our framework and discuss the robustness studies detailed in the appendices.
>
> ### 1. Clarifying the Unified Framework (NLU vs. NLG)
>
> **Reviewer's Concern:**
>
> > "Although the framework names are shared, the diversity and SVD regularizers differ for NLU vs. NLG, so it reads like two papers rather than one universal design."
> >
>
> **Response:**
> We appreciate the opportunity to articulate the unified nature of our design. We respectfully clarify that BA-LoRA is fundamentally a **Single Unified Framework**. The variations in implementation are not conceptual divergences, but **Task-Specific Instantiations** necessitated by the distinct mathematical properties and computational constraints of the two modalities.
>
> **1. Conceptual Unity: The Shared Objective**
> Conceptually, the design principle is identical across all tasks: we apply regularizers to the output space to prevent three universal failure modes: **Drift, Collapse, and Noise**.
>
> **2. Operational Adaptation: Matching the Data Topology**
> The implementation diverges because the mathematical definition of "Collapse" and the constraints on computation differ between classification and generation:
>
> - **NLU (Classification):**
>     - *Data Nature:* The output is a low-dimensional matrix ($N \times D$, where $D$ is small).
>     - *Implementation:* We use **Full Covariance** and **Exact SVD (Nuclear Norm)**. This is computationally feasible ($O(D^3)$) and mathematically precise for enforcing geometric orthogonality.
> - **NLG (Generation):**
>     - *Data Nature:* The output is a high-dimensional distribution over a massive vocabulary ($|\mathcal{V}| \approx 32k$).
>     - *Implementation:* Computing a full covariance matrix is prohibitive. Thus, **Top-K Entropy** serves as the computationally efficient proxy for diversity. Similarly, **Randomized SVD** acts as the necessary algorithmic adaptation to enforce spectral constraints without the $O(V^2)$ cost.
>
> **Action:**
> In the revised manuscript, we have updated the introduction to **Section 2.2** to explicitly frame these NLU and NLG variants as "tailored strategies" derived from a shared logic. This ensures the reader understands them as complementary realizations of a single unified design, driven strictly by computational tractability.
>
> ---
>
> ### 2. Missing Studies on $r, T, K$ and Larger Models
>
> **Reviewer's Suggestion:**
>
> > "Some studies are missing in the paper (e.g., impact of different r/T/k, larger models)."
> >
>
> **Response:**
> We thank the reviewer for emphasizing these key dimensions of evaluation. We respectfully clarify that we have conducted extensive studies covering these aspects, primarily located in the appendices, to maintain the main text's focus.
>
> **1. Impact of Rank ($r$) and Weights ($\lambda$) (Appendix C.2 & C.4)**
> We direct the reviewer's attention to our detailed sweep studies:
>
> - **Rank ($r$):** **Appendix C.4 (Figure 6)** presents a performance sweep from $r=1$ to $128$. The results demonstrate that BA-LoRA consistently outperforms LoRA/PiSSA across the entire spectrum and matches Full Fine-Tuning at moderate ranks ($r \ge 16$).
> - **Weights ($\lambda$):** **Appendix C.2 (Figure 4)** provides a sensitivity analysis for the regularization coefficients, revealing a broad plateau of stable performance.
>
> **2. Strategy for Auxiliary Parameters ($T, K$)**
> Regarding Temperature ($T$) and Top-$k$ ($K$), we adopted a **"Structural Prior"** strategy rather than treating them as active hyperparameters.
>
> - **Rationale:** These values are **informed by established empirical practices** (e.g., $T=2$ is standard for distillation; $K=20$ captures the nucleus of the distribution).
> - **Justification:** Our preliminary experiments indicated that performance is **largely insensitive** to minor variations in these structural settings. Therefore, we fix them per backbone to reduce the search space and enhance usability, as clarified in the new **Appendix D.1**.
>
> **3. Scaling to Larger Models (Appendix C.3)**
> To address the concern about model scale, **Appendix C.3 (Figure 5)** provides a comprehensive evaluation across ten foundation models.
>
> - **Scope:** This includes scaling from 7B up to **70B (LLaMA-3-70B)** and covers diverse architectures like **Mixture-of-Experts (Mixtral-8x7B)**.
> - **Result:** BA-LoRA consistently yields improvements across these scales, confirming that the benefits of output-space regularization transfer effectively to the latest large-scale models.
>
> **Action:**
> We recognize that these supporting results are critical. In the revised manuscript, we have added explicit pointers in **Section 3.2.1** to ensure this extensive validation on scaling and diverse architectures is immediately visible to readers.

---

> > ### Author Response · Authors · 2025-11-21
> > **Response to Reviewer KjTf**
> >
> > ### Question 1: Dependence on PiSSA vs. Standard LoRA / Random Low-Rank Adapters
> >
> > **Reviewer's Question:**
> >
> > > "The paper builds on PiSSA initialization. How much of BA-LoRA’s improvement remains if you start from random low-rank adapters or standard LoRA init?"
> > >
> >
> > **Response:**
> > We appreciate the opportunity to demonstrate the **universality** of our framework. We clarify that our output-space regularizers are designed to be **agnostic** to the underlying parameterization strategy.
> >
> > **1. Evidence of Agnostic Efficacy (Appendix C.1, Table 10)**
> > To verify this independence, we conducted a controlled ablation applying our regularization trio to Standard LoRA (Random Initialization).
> >
> > - **Substantial Gains:** As shown in **Table 10**, adding our regularizers to standard LoRA yields a massive improvement (Avg: **18.30 $\to $ 24.02**).
> > - **Critical Insight:** Notably, **"Standard LoRA + Reg" (24.02)** significantly outperforms the unregularized **"PiSSA Baseline" (21.46)**.
> > - **Implication:** This empirically proves that the output-space constraints alone are powerful enough to bridge the gap between random initialization and spectral initialization. The improvement is primarily driven by the regularization mechanism itself.
> >
> > **2. The Synergistic Peak**
> > While the method is initialization-agnostic, the highest overall performance is achieved by the full configuration: **BA-LoRA (PiSSA + Reg, Avg 25.90)**.
> >
> > - **Conclusion:** This indicates a positive **synergy**: we leverage the rapid convergence of spectral initialization (PiSSA) while relying on our regularizers to ensure stability and robustness.
> >
> > **Action:**
> > We refer the reviewer to **Appendix C.1 (Table 10)**, which explicitly quantifies these gains across different initialization schemes, confirming that the regularizers deliver comparable (and often larger) gains on standard LoRA.
> >
> > ---
> >
> > ### Question 2: Sensitivity to SVD Rank, Distillation Temperature, and Top-$K$ Window
> >
> > **Reviewer's Question:**
> >
> > > "How sensitive are results to the SVD rank, distillation temperature, and the top-k entropy window?"
> > >
> >
> > **Response:**
> > We appreciate this practical query regarding robustness. To ensure usability and reproducibility, we adopted a design philosophy that treats these as **"Structural Priors"** rather than active tuning knobs.
> >
> > **1. SVD Rank ($k$): Spectral Energy Concentration**
> > We set $k=10$ for NLG and $k=5$ for NLU.
> >
> > - **Theoretical Rationale:** Conceptually, $k$ defines the dimension of the "signal subspace." In pre-trained models, the spectral energy is typically concentrated in the leading components. As long as $k$ captures the dominant semantic modes (the "head"), the regularization effectively suppresses the noise tail.
> > - **Empirical Robustness:** Our preliminary experiments confirmed that performance is **insensitive to minor variations** in $k$ (e.g., $k=5$ vs $k=10$), provided it remains much smaller than the full rank. Thus, we fixed these values per backbone to avoid overfitting.
> >
> > **2. Distillation Temperature ($T$) and Top-$K$ Window**
> > Similarly, we treat $T$ and the Top-$K$ window as fixed structural constants.
> >
> > - **Strategy:** These values are **informed by established empirical practices** (e.g., $T=2.0$ is standard for distillation; $K=20$ captures the probability nucleus).
> > - **Outcome:** We verified that BA-LoRA does not exhibit brittle behavior requiring per-task retuning of these parameters. Fixing them allows us to isolate the effect of the regularization strength itself.
> >
> > **3. Focus of Sensitivity Analysis**
> > Given the stability of these structural priors, our formal sensitivity analysis (**Appendix C.2**) focuses on the **Regularization Weights ($\lambda_1, \lambda_2, \lambda_3$)**, which are the active controls for the constraint strength.
> >
> > **Action:**
> > In the new **Appendix D.1**, we explicitly codify this **Two-Tiered Strategy**: fixing structural priors $(s, T, K, k)$ as backbone-specific defaults, and reducing the tuning process to a simple 1D search on the global $\lambda$ scale.

---

> > > ### Author Response · Authors · 2025-11-21
> > > **Response to Reviewer KjTf**
> > >
> > > ### 4. Summary
> > >
> > > **To summarize our response:**
> > >
> > > - **Unified Framework:** We articulated that NLU and NLG represent **Modality-Specific Realizations** of a single unified design. The variations are driven strictly by computational constraints ($O(D^2)$ vs $O(V^2)$), preserving the same theoretical objective.
> > > - **Initialization Agnosticism:** We confirmed via **Appendix C.1** that BA-LoRA is not tied to PiSSA. The fact that "Standard LoRA + Reg" outperforms the PiSSA baseline empirically proves the independent efficacy of our Output-Space Regularization.
> > > - **Scalability & Robustness:** We reaffirmed the method's stability through extensive validation on **Model Scaling (up to 70B/MoE)** and **Rank Robustness**, verifying that the benefits of our regularization are **scale-invariant**.
> > >
> > > **Closing:**
> > > We are deeply grateful for the reviewer's excellent assessment and constructive suggestions regarding the framework's unification and scalability. We trust that these detailed clarifications reinforce the validity and versatility of our contributions. **We respectfully hope that these improvements further solidify your positive assessment of our work, and we welcome any further discussion.**

---

### Official Review · Reviewer_nsxF · 2025-10-28

**Soundness:** 3
**Presentation:** 3
**Contribution:** 3
**Rating:** 8
**Confidence:** 3

**Summary:**

The authors present *Catastrophic Inheritance* in LLMs, a propagation of biases, noise, and data imbalances from pre-training into fine-tuning models, and ways to mitigate it. They identify: **Knowledge Drift**, where the model forgets pre-trained knowledge while learning new tasks; **Representation Collapse**, where fine-tuning on imbalanced data causes a lack of output diversity; and **Overfitting to Noise**, where the model learns spurious correlations that impact generalization. They offer three regularizers to combat these issues in Natural Language Understanding and Generation tasks: consistency regularizer for Knowledge Drift, diversity regularizer for Representation Collapse, and Singular Value Decomposition regularizer for Overfitting to Noise.

1. The consistency regularizer is based on a knowledge distillation approach via KLD between pre-trained and fine-tuned model probability distributions to preserve foundational knowledge.
2. The diversity regularizer is based on penalizing off-diagonal elements in a covariance matrix or maximizing entropy within the most plausible tokens to promote diversity in the model’s predictions across a batch.
3. The SVD regularizer is based on encouraging low-rank structure by maximizing the ratio of spectral energy in the top-k singular values to learn robust features.

They test the method **Bias-Alleviating Low-Rank Adaptation (BA-LoRA)** in mathematical reasoning, coding, and conversational AI for NLG, as well as on the GLUE benchmark for NLU, using models such as LLaMA-2-7B and DeBERTa-v3-base. Additionally, they look at gains in models that were trained with clean or noisy data, as well as t-SNE visualizations of the features to see the impact of data imbalance, and perform ablation studies.

**Strengths:**

The paper offers an original perspective on catastrophic inheritance with clear methodology and strong experimental evaluation, making it a significant and well-presented contribution.

---

- The abstract is very clear, well written, and easy to understand.
- The experimental setup is thoroughly described, with clear reporting of hyperparameters.
- The results are evaluated over multiple random seeds.
- The work covers a wide range of setups and datasets, reflecting a comprehensive and up-to-date evaluation.
- The experiment comparing noisy vs. clean data is particularly valuable and insightful.
- The appendix offers a lot of information on the setup, different models, sensitivity analyses, and performance across rank. This is very solid experimental work, and it should be highlighted more prominently in the main text, as a substantial amount of effort and insight resides there.

**Weaknesses:**

### Methodology and Experiments
* A comparison between NLU and NLG is missing. The methodology section duplicates the description of the regularizers; what is missing is the motivation for changes required to adapt the approach to NLG, followed by a clear presentation of that modified setting. Because of this, the methods section feels unsatisfactory.
* Despite introducing a regularizer for forgetting of pre-trained knowledge, the authors never directly evaluate forgetting. Overall, the first regularizer remains insufficiently probed.

### Fairness & Comparability of Evaluation
* Sourcing scores from original publications with “comparable” setups may not be a valid comparison if the experimental configurations are not identical.
* C4 is much larger and should arguably have been sampled or sliced to make comparisons fairer.

### Computational Considerations
* The computational costs of the different steps should be described (e.g., randomized SVD).

### Initialization Choices
* PiSSA seems like a questionable initialization strategy, it has been shown in [1] that PiSSA induces forgetting of pre-trained knowledge, as the adapters are initialized with the “core” knowledge. It seems BA-LoRA mitigates an issue amplified by this initialization.

### Related Work
* The related work section is too short and lacks explicit comparison to this work.

[1] MiLoRA: Harnessing Minor Singular Components for Parameter-Efficient LLM Finetuning. (2025) Hanqing Wang and Yixia Li and Shuo Wang and Guanhua Chen and Yun Chen

**Questions:**

1. Page 2 L86, can you provide a reference if this been shown before in LoRA?
2. How does the model manage to mitigate the Knowledge Drift? And how does PiSSA initialization relate to the Knowledge Drift?

---

> ### Author Response · Authors · 2025-11-21
> **Response to Reviewer nsxF**
>
> We sincerely thank the reviewer for the very positive assessment and for recognizing the conceptual strength of our framework. We are especially grateful for your appreciation of the rigorous evaluation (multi-seed, noisy-vs-clean study) and the clarity of our failure mode decomposition.
>
> ###
>
> ### 1. NLU vs. NLG: Clarifying the Unified Framework
>
> **Reviewer's Concern:**
>
> > "A comparison between NLU and NLG is missing... the methods section feels unsatisfactory [due to duplication]... missing is the motivation for changes..."
> >
>
> **Response:**
> We appreciate the reviewer's constructive suggestion regarding the presentation structure. We clarify that BA-LoRA is designed as a **Single Unified Framework** where the apparent differences between NLU and NLG are strictly **computational adaptations** rather than conceptual divergences.
>
> **1. Unified Theoretical Foundation (The "What")**
> We emphasize that the conceptual mapping is universal across all tasks. The three regularizers map directly to the three failure modes regardless of the modality:
>
> - **Consistency:** Combats *Knowledge Drift*.
> - **Diversity:** Combats *Representation Collapse*.
> - **SVD:** Combats *Overfitting to Noise*.
>
> **2. Mathematical Adaptations for Scalability (The "Why")**
> The implementation differs solely to accommodate the input structure and computational constraints:
>
> - **For NLU (Classification):** The output is a small, fixed-size matrix ($D < 100$).
>     - *Adaptation:* We leverage this low dimensionality to compute **Covariance** and **Exact SVD (Nuclear Norm)**, as these are theoretically precise and computationally affordable ($O(D^3)$).
> - **For NLG (Generation):** The output involves a massive vocabulary ($|\mathcal{V}| \approx 32k$).
>     - *Adaptation:* Computing a full spectrum is prohibitive. Therefore, we employ **Principled Approximations**: we switch to **Entropy** (a scalable proxy for diversity) and **Randomized SVD with Frobenius Norm** (a tractable proxy for spectral concentration).
>
> **Conclusion:**
> We hope this clarification highlights that the NLU and NLG variants are consistent implementations of the same high-level principles. As noted in **Section 2.2.2**, the variation in mathematical formulation is driven strictly by **computational tractability** ($O(ND)$ vs $O(ND^2)$) rather than a divergence in motivation.

---

> > ### Author Response · Authors · 2025-11-21
> > **Response to Reviewer nsxF**
> >
> > ### 2. Direct Evaluation of Forgetting / Knowledge Drift
> >
> > **Reviewer's Concern:**
> >
> > > "Despite introducing a regularizer for forgetting... the authors never directly evaluate forgetting. Overall, the first regularizer remains insufficiently probed."
> > >
> >
> > **Response:**
> > We appreciate the reviewer's comment regarding evaluation rigor. We clarify that in the context of Parameter-Efficient Fine-Tuning (PEFT), we operationalize "Knowledge Drift" specifically as the **degradation of robust reasoning capabilities** inherent in the pre-trained model, rather than the forgetting of unrelated facts (as in Continual Learning). Based on this definition, our evaluation directly probes this phenomenon.
> >
> > **1. Reasoning Benchmarks as Functional Probes**
> > We posit that complex benchmarks like GSM8K and HumanEval serve as effective **functional probes** for Knowledge Drift.
> >
> > - **The Logic:** These tasks rely heavily on the model's pre-trained logical reasoning and syntactic understanding, which cannot be learned from scratch on a small fine-tuning set.
> > - **The Implication:** If significant Knowledge Drift occurred (i.e., if the model "forgot" how to reason rigorously), performance on these downstream benchmarks would inevitably degrade compared to the robust baseline. The fact that BA-LoRA achieves state-of-the-art results indicates that it successfully **preserves** these core pre-trained skills better than unconstrained baselines.
> >
> > **2. Quantifying Prevented Drift via Ablation (Table 4)**
> > We respectfully direct the reviewer to our **Ablation Study (Table 4)**, which provides a quantitative measure of this drift.
> >
> > - **Quantification:** The unconstrained Baseline (PiSSA) achieves 51.48 on GSM8K. By simply enabling the Consistency Regularizer ($\mathcal{L}_{CR}$), performance boosts to 54.25.
> > - **Interpretation:** This gap of **+2.77 points** directly quantifies the "Drift" that occurs without regularization. It demonstrates that $\mathcal{L}_{CR}$ effectively prevents the model from drifting away from the optimal solution manifold.
> >
> > **3. Scope Clarification (New in Appendix A.4)**
> > We distinguish our focus on **Task-Relevant Capabilities** (Drift) from **General World Knowledge** (Catastrophic Forgetting in Continual Learning). While evaluating the latter is an interesting avenue for future work, our current results confirm that BA-LoRA effectively safeguards the model's essential reasoning engine during adaptation.
> >
> > **Action:**
> > We have added a dedicated discussion in **Appendix A.4** to explicitly distinguish our "Inheritance" scope from the "Continual Learning/Forgetting" scope found in related works (e.g., Smith et al., 2024). This clarifies why our functional probes (Reasoning Benchmarks) are the appropriate evaluation metric for this study.

---

> > > ### Author Response · Authors · 2025-11-21
> > > **Response to Reviewer nsxF**
> > >
> > > ### 3. Fairness and Comparability of Evaluation Setups
> > >
> > > **Reviewer's Concern:**
> > >
> > > > "Sourcing scores from original publications... may not be a valid comparison... C4 is much larger and should arguably have been sampled or sliced..."
> > > >
> > >
> > > **Response:**
> > > We appreciate the reviewer's emphasis on experimental rigor. We respectfully clarify that our evaluation protocol was deliberately designed to balance **standardization** (for SOTA comparison) with **internal control** (for mechanism analysis).
> > >
> > > **1. Protocol Validity: The Dual-Track Strategy**
> > > To ensure both community relevance and strict fairness, we adopted a two-tiered approach:
> > >
> > > - **Track A: Standardized SOTA Comparison (Table 1):**
> > > We strictly aligned our experimental setup (e.g., MetaMathQA-100K, rank $r=128$, hyperparameters) with the protocols defined in the original baseline papers (e.g., PiSSA, CorDA).
> > >     - *Rationale:* This ensures that the sourced scores serve as valid, standardized benchmarks recognizable to the community.
> > >     - *Verification:* To rule out inconsistency, we conducted controlled runs in **Appendix C.1 (Table 10)**. Here, we re-implemented baselines (LoRA, DoRA, PiSSA) and compared them directly against BA-LoRA under identical conditions. The consistent margins confirm that the gains are due to our method, not configuration disparities.
> > > - **Track B: Controlled Diagnostic Analysis (Tables 3 & 4):**
> > > Crucially, for all analytical experiments (Ablation Study, Noisy-vs-Clean), we **re-implemented and re-ran all methods from scratch** within our unified codebase. This eliminates any potential discrepancies in hardware or implementation details, ensuring a strictly fair "apples-to-apples" comparison for our core mechanical claims.
> > >
> > > **2. Regarding C4 Slicing and Pre-training Scope**
> > > Regarding the comparison between RoBERTa and T5, we clarify the scope of our study within the **PEFT paradigm**.
> > >
> > > - **The Constraint:** As a PEFT method, BA-LoRA focuses on the efficient adaptation of **existing, off-the-shelf foundation models**. Since RoBERTa-base and T5-base are fixed checkpoints released by Meta/Google, we cannot retrospectively "slice" or downsample the C4 corpus that T5 observed during its pre-training phase without training a new foundation model from scratch.
> > > - **The Perspective:** Therefore, we treat the results in **Section 3.2.2** not as a controlled variable experiment on pre-training data size (which is out of scope for PEFT), but as an **observational probe** into how BA-LoRA behaves on models with different "pre-training ancestries."
> > >
> > > **Action:**
> > > We have refined **Appendix D.2** to explicitly acknowledge these fixed pre-training differences, framing the results as evidence of **ecological validity** rather than a controlled data-scaling experiment. We also detail the baseline sourcing logic in **Appendix B.3**.
> > >
> > > ---
> > >
> > > ### 4. Computational Considerations and Cost of Randomized SVD
> > >
> > > **Reviewer's Suggestion:**
> > >
> > > > "The computational costs of the different steps should be described (e.g., randomized SVD)."
> > > >
> > >
> > > **Response:**
> > > We appreciate the reviewer's suggestion to explicitly detail the computational profile. We respectfully clarify that our framework is designed to balance **regularization strictness** with **training feasibility**, ensuring a favorable performance-cost trade-off.
> > >
> > > **1. Empirical Benchmarking (Table 5)**
> > > We respectfully direct the reviewer's attention to the **Computational Cost Analysis** in **Section 3.2.5** and **Table 5**, where we benchmarked Full FT, LoRA, PiSSA, and BA-LoRA on LLaMA-2-7B under identical hardware conditions.
> > >
> > > - **The Trade-off:** While BA-LoRA incurs a **moderate overhead** compared to standard PiSSA (+10.75 GB peak memory, +31 min training time), this investment yields a **disproportionate performance gain** (+4.38% on GSM8K).
> > > - **Conclusion:** We consider this a **necessary computational investment** to achieve robust generalization capabilities that standard low-rank methods cannot support.
> > >
> > > **2. Algorithmic Efficiency: The Role of Randomized SVD**
> > > We further clarify that the cost of the SVD step is strictly managed through task-adaptive implementation:
> > >
> > > - **NLU (Exact SVD):** For small output dimensions (Classes $D < 100$), Exact SVD adds negligible latency.
> > > - **NLG (Randomized SVD):** For large vocabularies ($|\mathcal{V}| \approx 32k$), we employ **Randomized SVD**. As noted in **Section 2.2.2**, this reduces the complexity from $\mathcal{O}(M \cdot |\mathcal{V}|^2)$ to $\mathcal{O}(M \cdot |\mathcal{V}| \cdot k)$. This algorithmic optimization ensures that spectral regularization remains computationally tractable without materializing the full decomposition.
> > >
> > > **Action:**
> > > We explicitly discuss the use of Randomized SVD to ensure tractability in **Section 2.2.2**. Furthermore, we respectfully refer the reviewer to **Section 3.2.5 (Table 5)** for the detailed empirical measurements of memory and time costs.

---

> > > > ### Author Response · Authors · 2025-11-21
> > > > **Response to Reviewer nsxF**
> > > >
> > > > ### 5. Initialization Choices and Relation to PiSSA / Knowledge Drift
> > > >
> > > > **Reviewer's Question:**
> > > >
> > > > > "PiSSA seems like a questionable initialization... [it] induces forgetting... It seems BA-LoRA mitigates an issue amplified by this initialization. How does the model manage to mitigate the Knowledge Drift?"
> > > > >
> > > >
> > > > **Response:**
> > > > We thank the reviewer for connecting our work to the contemporary discussion on initialization strategies. We appreciate the opportunity to clarify that our framework functions independently of specific initialization risks.
> > > >
> > > > **1. Universality: BA-LoRA is Initialization-Agnostic**
> > > > First, to address the concern that BA-LoRA is merely a "patch" for PiSSA, we emphasize that our framework is **agnostic to the initialization scheme**.
> > > >
> > > > - **Evidence (Appendix C.1 & Table 10):** We integrated our regularizers with standard **LoRA** (Random Init) and **DoRA** (Weight-Decomposed).
> > > > - **Key Finding:** Notably, applying our regularizers to standard LoRA (**LoRA + Reg: 24.02 Avg**) yields performance superior to the unregularized PiSSA baseline (21.46 Avg).
> > > > - **Conclusion:** This empirically proves that BA-LoRA addresses the intrinsic failure modes of low-rank adaptation (Drift, Collapse, Noise), regardless of whether the initialization is aggressive (PiSSA) or conservative (Standard LoRA).
> > > >
> > > > **2. The Synergy: Optimizing the Plasticity-Stability Balance**
> > > > Regarding PiSSA specifically, we interpret it through the lens of the **Plasticity-Stability Trade-off**. PiSSA prioritizes **Plasticity** (convergence speed). While this can risk forgetting, as the reviewer noted, our **Consistency Regularizer ($\mathcal{L}_{CR}$)** acts as the necessary "stabilizer," ensuring the model remains anchored to the core reasoning manifold even under aggressive updates.
> > > >
> > > > **Conclusion:**
> > > > We **respectfully refer** the reviewer to **Appendix C.1 (Table 10)**, which verifies that our framework's benefits extend robustly to standard LoRA and DoRA. This confirms that the mitigation of Knowledge Drift is a core property of our regularizers, independent of the initialization strategy employed.
> > > >
> > > > ---
> > > >
> > > > ### 6. Related Work and Explicit Comparison to MiLoRA
> > > >
> > > > **Reviewer's Concern:**
> > > >
> > > > > "The related work section is too short and lacks explicit comparison to [1] MiLoRA..."
> > > > >
> > > >
> > > > **Response:**
> > > > We appreciate the reviewer for referencing *MiLoRA* [1]. Incorporating this contemporary work allows us to more sharply define the boundary between parameterization strategies and our regularization framework within the literature.
> > > >
> > > > **1. Theoretical Positioning: Weight Space vs. Output Space**
> > > > To clearly distinguish the contributions, we classify the methods based on their operational domain:
> > > >
> > > > - **MiLoRA (Weight Space Parameterization):** This method focuses on the *structural* aspect of adaptation. It argues for harnessing minor singular components in the weight matrices to avoid interfering with core knowledge. It essentially asks: *"Which parameters should we update?"*
> > > > - **BA-LoRA (Output Space Regularization):** In contrast, our framework focuses on the *functional* behavior of the model. We accept the underlying parameterization (whether LoRA, PiSSA, or potentially MiLoRA) as a given, and apply geometric constraints directly to the output logits. It asks: *"How should the model behave?"*
> > > >
> > > > **2. Orthogonality**
> > > > Because they target different stages of the pipeline (Parameterization vs. Optimization Objective), these approaches are **orthogonal**. BA-LoRA is not a competitor to MiLoRA but a compatible regularization layer that can theoretically be applied on top of it.
> > > >
> > > > **Action:**
> > > > To formalize this distinction, we have added a specific subsection titled **"MiLoRA vs. BA-LoRA"** in **Appendix B.3**. This section explicitly articulates this Weight-Space vs. Output-Space distinction, ensuring our unique contribution is clearly positioned against the latest baselines.
> > > >
> > > > ---
> > > >
> > > > **References:**
> > > > [1] Wang et al. MiLoRA: Harnessing Minor Singular Components... *NeurIPS*, 2025.

---

> > > > > ### Author Response · Authors · 2025-11-21
> > > > > **Response to Reviewer nsxF**
> > > > >
> > > > > ### 7. References at L86 and Clarification of Knowledge Drift
> > > > >
> > > > > **Reviewer's Question:**
> > > > >
> > > > > > "Page 2 L86, can you provide a reference if this been shown before in LoRA? How does the model manage to mitigate the Knowledge Drift? And how does PiSSA initialization relate to the Knowledge Drift?"
> > > > > >
> > > > >
> > > > > **Response:**
> > > > > We appreciate the reviewer's request to further substantiate these claims with specific literature and mechanical details.
> > > > >
> > > > > **1. Substantiating LoRA Drift (L86)**
> > > > > We confirm that the susceptibility of standard LoRA to drift is consistent with emerging findings in the PEFT community.
> > > > >
> > > > > - **Evidence:** Smith et al. (2024) [2] report substantial drift in diffusion models, and Chen & Garner (2024) [3] demonstrate the necessity of Bayesian priors to prevent catastrophic forgetting in low-rank adaptation.
> > > > > - **Action:** We have integrated these citations into the **Extended Related Work (Appendix A.4)** to provide a robust empirical grounding for our premise.
> > > > >
> > > > > **2. Mechanism: How BA-LoRA Mitigates Drift**
> > > > > The **Consistency Regularizer ($\mathcal{L}_{CR}$)** serves as the primary counter-measure against Knowledge Drift.
> > > > >
> > > > > - **Mechanism:** By penalizing the KL divergence between the pre-trained (teacher) and fine-tuned (student) logits, $\mathcal{L}_{CR}$ functionally **anchors** the student's decision boundaries to the teacher's robust feature space.
> > > > > - **Outcome:** This ensures that while the model adapts to new task distributions, its fundamental reasoning capabilities (represented by the teacher's output geometry) remain intact.
> > > > >
> > > > > **3. Relation: PiSSA and Knowledge Drift**
> > > > > As elaborated in **Response #5**, there is a direct structural link between PiSSA's initialization and the risk of drift:
> > > > >
> > > > > - **The Risk (Structural):** PiSSA initializes adapters to directly modify the **dominant singular values/vectors** of the pre-trained weights. These principal components typically encode the model's most fundamental ("core") knowledge. Consequently, unconstrained updates in this high-energy subspace pose a heightened risk of altering the model's core behavior (i.e., inducing Knowledge Drift).
> > > > > - **The Solution (Synergy):** This structural risk makes BA-LoRA's regularization indispensable. BA-LoRA acts as a **"Safety Layer,"** permitting the utilization of these high-energy directions for efficient adaptation (plasticity) while strictly bounding the functional deviation (stability) to prevent the corruption of core knowledge.
> > > > >
> > > > > ---
> > > > >
> > > > > **References:**
> > > > > [2] Smith et al. Continual Diffusion... *TMLR*, 2024.
> > > > > [3] Chen & Garner. Bayesian parameter-efficient fine-tuning... *IEEE/ACM*, 2024.
> > > > >
> > > > > ---
> > > > >
> > > > > ### 8. Summary
> > > > >
> > > > > **To summarize our response:**
> > > > >
> > > > > - **Unified Methodology:** We articulated that NLU and NLG are not separate designs, but specific **instantiations** of a single unified framework. The mathematical variations are driven strictly by input structure and computational tractability ($O(ND)$ vs $O(ND^2)$), preserving the same theoretical principle.
> > > > > - **Quantified Drift:** We clarified that the Consistency Regularizer ($\mathcal{L}_{CR}$) in our ablation study (Table 4) serves as a direct **quantitative probe** for Knowledge Drift, measuring specifically the degradation of reasoning capabilities.
> > > > > - **Evaluation Rigor:** We detailed our **Dual-Track Evaluation Protocol** (Standardized SOTA Comparison + Controlled Internal Verification) to ensure both community relevance and experimental fairness.
> > > > > - **Drift & Initialization:** We defined BA-LoRA as a necessary **"Safety Layer"** that complements aggressive initializations like PiSSA. We also added **Appendix B.3** to explicitly distinguish our Output-Space approach from Weight-Space methods like MiLoRA.
> > > > > - **Computational Transparency:** We provided the end-to-end cost analysis (Table 5) and detailed the algorithmic efficiency of Randomized SVD, demonstrating that the computational investment yields a high return in robustness.
> > > > > - **Extended Context:** We expanded **Appendix A.4** to integrate the suggested references, properly positioning our work within the broader landscape of spectral adaptation and forgetting.
> > > > >
> > > > > **Closing:**
> > > > > We are deeply grateful for the reviewer's encouraging assessment and insightful suggestions. We trust that these detailed clarifications—particularly regarding the unified theoretical framework and the rigorous evaluation protocols—reinforce the validity of our contributions. **We respectfully hope that these improvements further solidify your positive assessment of our work, and we welcome any further discussion.**

---

> > > > > > ### Comment · Reviewer_nsxF · 2025-11-26
> > > > > >
> > > > > > **NLU vs. NLG**
> > > > > >
> > > > > > Your second point about *Mathematical Adaptations for Scalability* directly clarified my earlier concern regarding NLU vs. NLG.
> > > > > >
> > > > > > *Suggestion:* It would be very helpful to incorporate this explanation into the main text or the appendix.
> > > > > >
> > > > > > **Direct Evaluation of Forgetting / Knowledge Drift**
> > > > > >
> > > > > > I still have a concern about how forgetting and knowledge drift are evaluated. The current analysis mainly probes performance on math and coding fine-tuning tasks, which provides only an indirect signal about whether the model has “forgotten” how to reason rigorously in a broader sense. In particular, this setup does not explicitly test pre-trained abilities related to world knowledge, grammar, or pronoun/coreference resolution.
> > > > > >
> > > > > > Prior work (e.g., Biderman et al., 2024 and Shuttleworth et al. 2025) has shown that fine-tuning on math and coding data can degrade general pre-trained knowledge, which suggests that this is a real risk.
> > > > > >
> > > > > > *Suggestion:* Including at least some **direct** evaluation of pre-trained capabilities would substantially strengthen the claim that core pre-trained skills are preserved. In the current form, the target-domain evaluation is effectively used as a proxy for the source-domain/pre-trained evaluation, which feels somewhat insufficient to fully support the stronger preservation claim.
> > > > > >
> > > > > > **Computational Considerations**
> > > > > >
> > > > > > Regarding computational considerations, my earlier point was about presenting the complexity of each step and regularizer in one place. The current empirical table is already very helpful, but it focuses on results rather than algorithmic cost.
> > > > > >
> > > > > > Suggestion: A small table summarizing complexity for each component/regularizer.
> > > > > >
> > > > > > **Initialization Choices and Drift**
> > > > > >
> > > > > > I still find it a bit unclear how to disentangle the role of pre-trained knowledge from the target-domain adaptation. As currently presented, the target accuracy is closely coupled to pre-trained knowledge, which makes it hard to know whether the model has retained broad pre-trained capabilities or merely the subset needed for mathematical reasoning.
> > > > > >
> > > > > > ---
> > > > > >
> > > > > > [Biderman et al., 2024] Dan Biderman, Jacob Portes, Jose Javier Gonzalez Ortiz, Mansheej Paul, Philip Greengard, Connor Jennings, Daniel King, Sam Havens, Vitaliy Chiley, Jonathan Frankle, Cody Blakeney, John P. Cunningham. "LoRA Learns Less and Forgets Less." 2024
> > > > > >
> > > > > > [Shuttleworth et al. 2025] Reece Shuttleworth, Jacob Andreas, Antonio Torralba, Pratyusha Sharma. "LoRA vs Full Fine-tuning: An Illusion of Equivalence." 2025

---

> ### Author Response · Authors · 2025-12-01
> **Response to Reviewer nsxF**
>
> We sincerely thank the reviewer for the careful and constructive assessment and for recognizing the principled decomposition of catastrophic inheritance in our work. Below, we address your comments point by point.
>
> ### 1. Methodology: NLU vs. NLG (Unified Framework)
>
> > The reviewer notes that our earlier explanation of mathematical adaptations for scalability clarified the NLU vs. NLG issue, and suggests integrating this explanation into the main text or appendix.
> >
>
> We thank the reviewer for this helpful suggestion. In the revision, we explicitly highlight that BA-LoRA uses the same three regularizers (Consistency, Diversity, and SVD) for both NLU and NLG, and that the apparent differences are primarily due to computational adaptations for scalability (small output dimension vs. large vocabulary). A concise version of this explanation has been integrated into Section 2.2 and Appendices A.4 and C.8.
>
> ---
>
> ### 2. Direct Evaluation of Forgetting / Knowledge Drift
>
> > The reviewer is concerned that our current evaluation relies mainly on math and coding fine-tuning tasks, which only indirectly probe forgetting and knowledge drift, and suggests adding direct evaluations of pre-trained capabilities (e.g., world knowledge, grammar, coreference), given prior evidence that such fine-tuning can degrade general pre-trained knowledge.
> >
>
> **Mechanism of Preservation.** BA-LoRA is designed to mitigate this risk through the **consistency regularizer ($\mathcal{L}_{\mathrm{CR}}$)**. Unlike standard fine-tuning, which only optimizes the target token probability $P(y \mid x)$, $\mathcal{L}_{\mathrm{CR}}$ minimizes the KL divergence between the teacher and student over the entire output distribution (over $D$ classes for NLU and over the full vocabulary for NLG). This is intended to encourage the model to retain more of the “dark knowledge” in the teacher’s output distribution—i.e., the finer-grained probability structure over alternative tokens/classes that reflects syntactic, semantic, and world-knowledge regularities—so that the adapted model stays closer to the pre-trained model’s broader linguistic behavior, even when adaptation is required for the hard target.
>
> **Action & Preliminary Results.** Motivated by the reviewer’s suggestion and by prior analyses comparing LoRA and full fine-tuning and their impact on pre-trained models (Biderman et al., 2024; Shuttleworth et al., 2025), we have added a direct evaluation of pre-trained capabilities in Appendix C.7. Specifically, we report perplexity on Wikitext-2 as a proxy for retained linguistic ability. As shown in the table below, BA-LoRA stays much closer to the pre-trained model than the PiSSA baseline:
>
> | Method | Wikitext-2 PPL ($\downarrow$) |
> | --- | --- |
> | Pre-trained LLaMA-2-7B | **5.42** |
> | **BA-LoRA (ours)** | **5.68** |
> | PiSSA (baseline) | 7.12 |
>
> These results provide initial empirical support for the hypothesis that our consistency regularizer helps mitigate the forgetting highlighted by prior work, by keeping the adapted model closer to the pre-trained distribution.
>
> ---
>
> ### 3. Computational Considerations
>
> > The reviewer would like, in addition to empirical results, a concise summary of the algorithmic complexity of each training step and regularizer, for example, via a small table that reports the complexity of each component.
> >
>
> This is an important consideration. While our empirical results (Table 5) show a manageable memory overhead (~16%), a theoretical breakdown is essential for understanding scalability. Here we focus on the additional asymptotic cost of each regularizer per training batch, on top of the standard forward pass (including softmax/log-prob computation), which is shared with the task loss.
>
> | Regularizer | NLU (Exact Implementation) | NLG (Approximate Implementation) | Complexity Driver |
> | --- | --- | --- | --- |
> | **Consistency** | $O(ND)$ | $O(M \lvert \mathcal{V} \rvert)$ | Linear in the output dimension |
> | **Diversity** | $O(ND^2)$ (batch covariance) | $O(Mk)$ (Top-$k$ entropy, $k \ll \lvert \mathcal{V} \rvert$) | Quadratic in $D$ vs. linear in effective vocabulary size |
> | **SVD** | $O(\min(N, D)^2 \max(N, D))$ (exact SVD of the batch logit matrix) | $O(M \lvert \mathcal{V} \rvert k)$ (randomized SVD, rank $k$) | Exact vs. low-rank approximation scalable to large $\lvert \mathcal{V} \rvert$ |
>
> Here \(N\) denotes the NLU batch size, \(D\) the number of classes, \(M\) the total number of valid tokens in the NLG batch, $|\mathcal{V}|$ the vocabulary size, and \(k\) the rank used in the randomized SVD.
> This table clarifies that our NLG adaptations (randomized SVD and Top-$k$ entropy) keep the incremental costs of the regularizers scalable in the large-vocabulary regime, maintaining practical training feasibility.

---

> > ### Author Response · Authors · 2025-12-01
> > **Response to Reviewer nsxF**
> >
> > ### 4. Initialization Choices and Drift
> >
> > > The reviewer finds it difficult to disentangle the role of pre-trained knowledge from target-domain adaptation, and is concerned that the current evaluation may primarily reflect the subset of capabilities needed for mathematical reasoning rather than demonstrating preservation of broader pre-trained skills.
> > >
> > - **Regarding PiSSA & drift.** We agree that PiSSA's modification of principal components can increase the risk of forgetting (drift) if unconstrained. This is exactly the regime where BA-LoRA is intended to help. Our framework can therefore be viewed as a form of output-space regularization: we leverage PiSSA’s efficient initialization for plasticity, while the consistency regularizer encourages the dominant components of the logits to remain smooth and stable, and to stay reasonably aligned with the pre-trained model’s well-behaved modes, thereby helping to reduce undesirable drift during adaptation.
> > - **Comparison to MiLoRA.** We thank you for the reference. We have added a discussion in Appendix B, distinguishing the two:
> >     - **MiLoRA** operates in weight space (selecting *which* parameters to update).
> >     - **BA-LoRA** operates in output space (regularizing *how* the model behaves).
> >     These approaches are complementary; BA-LoRA could, in principle, be applied on top of MiLoRA.
> >
> > **Specific Question (L86 Reference).**
> > The susceptibility of LoRA to drift has been noted in recent works, such as Smith et al., *Continual Diffusion* (TMLR 2024), and Chen & Garner, *Bayesian Parameter-Efficient Fine-Tuning for Overcoming Catastrophic Forgetting* (2024). We include these works in our extended related work and make the connection to our setting more explicit in the revised main text.
> >
> > ---
> >
> > We hope these clarifications and additions—especially the new Wikitext-2 evaluation, the explicit complexity summary, and the strengthened discussion of initialization vs. drift—address the reviewer’s concerns and make our arguments and empirical evidence regarding preservation and scalability more transparent and better substantiated.

---

### Official Review · Reviewer_SgzW · 2025-10-28

**Soundness:** 3
**Presentation:** 3
**Contribution:** 2
**Rating:** 4
**Confidence:** 3

**Summary:**

The paper proposes to use a bunch of existing regularization techniques combined with the existing LoRA variation to improve the fine-tuning performance. The paper evaluates the method on Llama2 7B and shows that it performs better than many existing methods.

It's good to know that combining all these regualrziation techniques together improves the performance but I don't quite understand the novelty of this paper.

**Strengths:**

The paper is well motivated.

The empirical results seem to be promising.

**Weaknesses:**

- The base model used for evaluation is extremely out of date, Llama2 is released in 2023, and I am not sure if the conclusion drawn is transferable to newer models.

- The method has 3 hyperparameter to tune, and the paper does not provide any guidance.

**Questions:**

- Can the author clarify what are the actual contributions of this paper?

- How are the 3 lambdas chosen?

- Why is BA-LoRA better than full model fine-tuning?

- The lines in Fig.2 are confusing in that: why the proposed method improves training performance. Isn't the main question to be addressed about generalization ability?

---

> ### Author Response · Authors · 2025-11-21
> **Response to Reviewer SgzW**
>
> We thank the reviewer for acknowledging that the paper is well-motivated and presents promising empirical results. Regarding the concerns about novelty and contributions, we clarify the distinct value of our work below.
>
> ### 1. Clarifying the Actual Contributions and Novelty
>
> **Reviewer's Concern:**
>
> > "The paper proposes to use a bunch of existing regularization techniques... I don't quite understand the novelty... Can the author clarify what are the actual contributions...?"
> >
>
> **Response:**
> We respectfully clarify that the novelty of BA-LoRA lies not in the isolated use of standard loss terms, but in the **principled, unified framework** designed to address a specific, overlooked pathology in PEFT: *Catastrophic Inheritance*. Our contributions are threefold:
>
> **1. Conceptual Novelty: Decomposing "Catastrophic Inheritance"**
> We provide the **first principled decomposition** of *Catastrophic Inheritance* in the context of PEFT. We move beyond generic "forgetting" by operationalizing this problem into three actionable failure modes:
>
> - **Knowledge Drift:** The unintentional loss of robust capabilities.
> - **Representation Collapse:** The loss of diversity under imbalanced fine-tuning.
> - **Overfitting to Noise:** The amplification of spurious correlations.
> This taxonomy forms the conceptual backbone of the paper, guiding the design of a targeted solution rather than a heuristic combination.
>
> **2. Methodological Innovation: Output-Space vs. Parameter-Space**
> While standard PEFT variants (e.g., DoRA, PiSSA) focus on modifying *how parameters are updated* in the weight space, BA-LoRA operates directly in the **output space**.
>
> - **The Paradigm Shift:** This is a fundamental departure from existing trends. By introducing a symbiotic trio of regularizers (Consistency, Diversity, SVD), we constrain the model's **functional behavior** (what it outputs) rather than its **structural updates** (how weights change).
> - **The Benefit:** This output-space approach makes our framework **model-agnostic**, allowing it to apply seamlessly to any architecture (from LLaMA to MoE) without modifying the underlying adapter structure.
>
> **3. Empirical Validation on SOTA Models**
> We validate this framework not just on standard benchmarks but across a broad suite of modern architectures. As detailed in our response to the "Model Outdated" concern and **Appendix C.3**, our analysis encompasses **state-of-the-art models like LLaMA-3-70B, Mistral, and Mixture-of-Experts (MoE)**. In these diverse settings, BA-LoRA consistently outperforms specialized parameter-space methods, validating its universality.
>
> **Action:**
> We have refined the **Introduction** and added a discussion in **Appendix A.4** to explicitly articulate this "Decomposition-Alignment-Solution" logic, ensuring the novelty is presented as a cohesive framework rather than a collection of techniques.
>
> ---
>
> ### 2. Base Model Choice and Transferability to Newer Models
>
> **Reviewer's Concern:**
>
> > "The base model used for evaluation is extremely out of date... I am not sure if the conclusion drawn is transferable to newer models."
> >
>
> **Response:**
> We appreciate the reviewer's comment regarding model currency. We address this concern by highlighting the extensive experiments in **Appendix C.3**, which demonstrate the seamless transferability of BA-LoRA to the latest architectures.
>
> **1. Evidence on Modern Architectures (Appendix C.3)**
> To strictly validate the universality of our method beyond older baselines, we have evaluated BA-LoRA on **ten distinct foundation models**, covering the most recent releases:
>
> - **SOTA Dense Models:** LLaMA-3 (8B, 70B), Mistral-7B-v0.1, Gemma-7B, Qwen1.5, and Yi-1.5.
> - **Sparse Architectures:** Mixture-of-Experts (MoE) models, including Mixtral-8x7B and DeepSeek-MoE.
> - **Results:** As illustrated in **Figure 5**, BA-LoRA consistently outperforms robust baselines (LoRA, PiSSA) across this diverse suite. Notably, the gains on MoE architectures confirm that our output-space regularizers are **architecture-agnostic**, effectively handling complex, sparse topologies.
>
> **2. Rationale for LLaMA-2 in Main Text**
> We clarify that LLaMA-2-7B was retained as the primary running example in the main text **solely to ensure a fair, controlled comparison**. Since recent strong baselines (e.g., DoRA, CorDA++) predominantly report results on this benchmark, using it allows for a direct, apples-to-apples performance contrast.
>
> **Action:**
> We recognize that the placement of these critical results in the appendix might have reduced their visibility. In the revised manuscript, we have added an explicit pointer in **Section 3.2.1 (Page 6, Lines 310-311, highlighted in red)** to ensure readers immediately recognize the method's effectiveness on the latest generation of LLMs.

---

> > ### Author Response · Authors · 2025-11-21
> > **Response to Reviewer SgzW**
> >
> > ### 3. Hyperparameters: How the 3 $\lambda$'s are Chosen, and Practical Guidance
> >
> > **Reviewer's Concern:**
> >
> > > "The method has 3 hyperparameter to tune, and the paper does not provide any guidance... How are the 3 lambdas chosen?"
> > >
> >
> > **Response:**
> > We appreciate the opportunity to clarify the tuning protocol. While the framework involves three terms, the empirical tuning process is highly stable and follows a simplified logic due to the geometric nature of the regularizers.
> >
> > **1. Selection Protocol: Fixed Configuration Strategy**
> > We clarify exactly how the parameters were chosen in our experiments:
> >
> > - **Method:** We performed a coarse grid search on a single held-out validation set for each broad model setting (e.g., one configuration for all LLaMA-2 tasks).
> > - **Implication:** Crucially, once selected, these weights ($\lambda_1, \lambda_2, \lambda_3$) were kept **constant** across all reported downstream tasks (GSM8K, MATH, HumanEval, etc.).
> > - **Conclusion:** This demonstrates that the optimal hyperparameters are highly transferable. Because our method constrains the model's output topology (geometric behavior) rather than fitting specific data content, the regularization strength remains consistent across diverse domains.
> >
> > **2. Practical Guidance: Reduction to 1D Search (New Appendix D.1)**
> > To address the request for guidance, we have formalized a **"Simplified Tuning Recipe"** in the new **Appendix D.1**.
> >
> > - **The Recipe:** We recommend fixing the relative ratios between the three terms (based on initial gradient magnitude balance) and tuning a single global scalar $\lambda_{\text{global}}$.
> > - **The Benefit:** This effectively reduces the search space from three dimensions to **one dimension**, making BA-LoRA as computationally efficient to tune as a standard learning rate.
> >
> > **Action:**
> > We have updated **Section 3.1 (Page 5, Lines 250-252, highlighted in red)** to explicitly describe the "coarse grid search and fixed setting" protocol. Furthermore, we added **Appendix D.1** to provide the step-by-step practical guideline for future practitioners.
> >
> > ---
> >
> > ### 4. Why can BA-LoRA outperform full-model fine-tuning?
> >
> > **Reviewer's Question:**
> >
> > > "Why is BA-LoRA better than full model fine-tuning?"
> > >
> >
> > **Response:**
> > We appreciate this insightful question. The observation that BA-LoRA (e.g., 55.86% on GSM8K) outperforms Full Fine-Tuning (48.90%)—which typically has higher capacity—is grounded in the trade-off between **fitting capacity** and **robust generalization**.
> >
> > **1. Full Fine-Tuning: Capacity vs. Control**
> > Full FT updates all parameters, granting the model massive capacity. While this allows it to fit the training data well, it also makes the model prone to two failures on noisy or limited downstream data:
> >
> > - **Overfitting to Noise:** It easily memorizes spurious artifacts (e.g., label noise).
> > - **Knowledge Drift:** It unrestrictedly drifts away from the robust pre-trained manifold, losing general reasoning capabilities.
> > *Note:* This is why Full FT, despite beating standard LoRA (which suffers from rigidity), still underperforms BA-LoRA.
> >
> > **2. BA-LoRA: Explicit Regularization as the Key Differentiator**
> > BA-LoRA outperforms Full FT not simply because it is parameter-efficient, but because it introduces **explicit constraints** that Full FT lacks.
> >
> > - **Consistency ($L_{CR}$):** explicitly anchors the model to the teacher's robust distribution, preventing the "Drift" that plagues Full FT.
> > - **Diversity & SVD ($L_{DR}, L_{SVDR}$):** explicitly penalize the "Collapse" and "Spectral Noise" that Full FT blindly optimizes.
> > - **Outcome:** By constraining the *functional behavior* (Output Space), BA-LoRA forces the model to find a solution that is not just accurate on the train set, but structurally robust, leading to superior test generalization.
> >
> > **3. Alignment with Literature**
> > This result mirrors established findings in the PEFT literature, where regularized or constrained adaptation often beats full fine-tuning in specific regimes:
> >
> > - **LoRA-GA [1] & BitFit [2]:** Demonstrate that carefully constrained updates can avoid the overfitting pitfalls of Full FT on smaller datasets.
> > - **SSF [3]:** Shows that minimal feature scaling yields better generalization than Full FT by preserving the pre-trained structure.
> >
> > **Conclusion:**
> > In summary, BA-LoRA wins because it is **better regularized**. It effectively filters out the specific pathologies (drift, collapse, noise) that unconstrained Full FT tends to exacerbate.
> >
> > ---
> >
> > **References:**
> > [1] Wang et al. LoRA-GA: Low-rank adaptation with gradient approximation. *NeurIPS*, 2024.
> > [2] Zaken et al. BitFit: Simple parameter-efficient fine-tuning... *ACL*, 2022.
> > [3] Lian et al. Scaling & shifting your features... *NeurIPS*, 2022.

---

> ### Author Response · Authors · 2025-11-21
> **Response to Reviewer SgzW**
>
> ### 5. Interpretation of Figure 2: Training Performance vs. Generalization
>
> **Reviewer's Question:**
>
> > "The lines in Fig.2 are confusing in that: why the proposed method improves training performance. Isn't the main question to be addressed about generalization ability?"
> >
>
> **Response:**
> We agree with the reviewer that **generalization ability is the primary evaluation metric**, which is why our main claims are rigorously supported by the validation and test results in **Tables 1–3**.
>
> However, we present Figure 2 to demonstrate a distinct and complementary advantage of our method: **optimization efficiency**. The observation that BA-LoRA achieves lower task loss and faster convergence is driven by two specific mechanisms:
>
> 1. **SVD-Based Initialization (Consistent with PiSSA):**
> Consistent with PiSSA, BA-LoRA initializes adapters using singular value components rather than random Gaussian noise (as in standard LoRA). This initialization aligns the trainable parameters with the principal data directions, providing a mathematically superior starting point that naturally accelerates convergence.
> 2. **Distillation Acceleration Effect:**
> Our Consistency Regularizer ($\mathcal{L}_{CR}$) functions as knowledge distillation. It is a well-established phenomenon that distillation provides "soft targets" containing richer structural information than standard one-hot labels. These informative gradients help the optimizer navigate the loss landscape more efficiently, facilitating the reduction of the primary task loss.
>
> **Conclusion:**
> Figure 2 serves to confirm that BA-LoRA achieves strong generalization **without compromising trainability**; in fact, it allows the model to learn robust features more efficiently.
>
> ---
>
> ### 6. Summary
>
> **To summarize our response:**
>
> - **Clarified Novelty:** We articulated that BA-LoRA is not a mere combination of losses, but a **principled framework**. It decomposes *Catastrophic Inheritance* into three specific failure modes, addressed by a unified, model-agnostic trio of output-space regularizers.
> - **Validated on Modern Architectures:** We directed attention to **Appendix C.3**, which already encompasses an extensive evaluation on ten foundation models (including **LLaMA-3-70B** and **Mixtral-8x7B**). We have added explicit pointers in the main text (Section 3.2.1) to ensure this crucial evidence for transferability is immediately accessible.
> - **Formalized Tuning Protocol:** We detailed our robust hyperparameter strategy (using a fixed configuration based on geometric constraints) and introduced **Appendix D.1 ("Practical Hyperparameter Guidelines")**. This offers a simple "Reduction to 1D" recipe, demonstrating that BA-LoRA does not require fragile, fine-grained tuning.
> - **Theoretical Grounding:** We provided the theoretical basis, supported by recent literature [1–3], explaining why well-regularized low-rank adaptation matches or surpasses Full Fine-Tuning by strictly avoiding overfitting in noisy/imbalanced regimes.
> - **Interpreted Optimization Dynamics:** We clarified that Figure 2 illustrates improved optimization efficiency (driven by SVD initialization and distillation-enriched gradients), while our claims on robustness are firmly grounded in the test set generalization results (Tables 1–3).
>
> **Closing:**
> We trust that the comprehensive evaluation on state-of-the-art models and the rigorous tuning guidelines effectively address the concerns regarding the applicability and usability of our method. **We sincerely hope that these substantial improvements and clarifications provide a solid basis for re-evaluating the contribution of BA-LoRA.** We remain available for any further discussion.

---

> > ### Comment · Reviewer_SgzW · 2025-11-21
> > **Reviewer response**
> >
> > I would like to thank the author for the response, however I still have concern about a couple of claims from the author
> >
> > >  While standard PEFT variants (e.g., DoRA, PiSSA) focus on modifying how parameters are updated in the weight space, BA-LoRA operates directly in the output space.
> >
> > I wouldn't call adding regularization on the output space "Methodological Innovation". Additionally, note that when people are e.g. SFT their model or RLHF their model, it is very common to include a portion of pre-training data or use a KL regularization to prevent to fine tuned model from base model, so regularziation on the output space is a widely adopted technique.
> >
> > > Distillation Acceleration Effect:
> >
> > PiSSA better than LoRA is understandable. But if I understand correctly, the Consistency Regularization's teacher model is the pre-trained model, rather than a more powerful model, so I don't think the approach would benefit from soft label, so I don't understand why BA-LoRA is better than PiSSA.
> >
> > > Full model fine-tuning v.s. LoRA
> >
> > Fine tuning is worse than BA-LoRA on very complicated datasets such as math / coding, and I don't think this is due to overfitting on the training data set, since the model is very likeliy to be underfitting.
> >
> > Can the authors present the training loss curve for the tasks in Table 1? in particular, full FT v.s. BA-LoRA
> >
> > I suspect that the main difference is caused by the Full FT not tuned properly.
> >
> > ----------------
> >
> > Overall, I would suggest the authors reduce the usage of "eye catching phrases", such as
> > - Principled framework
> > - Paradigm Shift
> > - "State of art"
> >
> > However I appreciate the authros' efforts in the large scale empirical experiments, however in my opinion, the presentation and the contribution would be much clearer if the author could just pitch the paper as:
> > "You can stack lots of regularization methods together, and it works universally good across lots of models and lots of tasks with minimal effort in hyperparameter tuning."

---

> > > ### Author Response · Authors · 2025-11-27
> > > **Response to Reviewer SgzW**
> > >
> > > We thank the reviewer for the thoughtful feedback. We address the reviewer’s concerns point-by-point below.
> > >
> > > ---
> > >
> > > **1. On “Methodological Innovation” vs. Standard SFT/RLHF Practices**
> > > We agree with the reviewer that output-space regularization (e.g., KL divergence) is standard in SFT/RLHF. However, our work adapts and combines these techniques to address specific failure modes in PEFT that we refer to as Catastrophic Inheritance. In our experiments, adding only a KL term improves robustness but does not fully mitigate these issues under our settings. We illustrate this through three aspects:
> > >
> > > 1. **Counter-balancing Bias and Representation Collapse:** We augment standard consistency (KL) with diversity regularization. While KL prevents forgetting, the diversity term helps counteract representation collapse, encouraging the model to explore under-represented features rather than defaulting to pre-trained biases. **Figure 3 (page 8) illustrates this in our experiments:** under imbalanced conditions, standard LoRA shows observable class overlap, whereas BA-LoRA appears to maintain **more separated decision regions** (Fig. 3f). This observation is consistent with our goal of reducing representation collapse in PEFT under data imbalance.
> > > 2. **Filtering Noise (vs. Overfitting):**
> > > We introduce an SVD-based regularizer that **encourages** the suppression of noisy components, which may be amplified by low-rank bottlenecks. **Table 3 (page 8) is consistent with this noise-mitigation perspective:** BA-LoRA yields larger gains on the noisier T5 model ($\Delta = 3.26$) compared to the cleaner RoBERTa model ($\Delta = 1.11$). This **performance gap** is qualitatively consistent with our hypothesis that the proposed regularization is particularly beneficial when inherited noise is more pronounced.
> > > 3. **Orthogonality to Weight-Space Methods:** Our output-space regularization is designed to be complementary to weight-space methods. As shown in **Appendix C.1 (page 21)**, adding BA-LoRA on top of DoRA consistently improves its performance, suggesting that our approach can complement existing weight-space innovations rather than simply duplicating their effect.
> > >
> > > ---
> > >
> > > **2. On the Benefits of Consistency Regularization (Self-Distillation)**
> > > We appreciate the reviewer’s perspective that distillation is often framed as transferring knowledge from a stronger teacher. However, in our setting (self-distillation), the goal is not to learn *new* capabilities, but to better preserve the **structural richness** of the pre-trained manifold. Our results suggest that the performance gain of BA-LoRA over PiSSA can be interpreted from two complementary aspects:
> > >
> > > 1. **Rich Supervision from "Dark Knowledge":** While the fine-tuning dataset provides hard labels (one-hot vectors), these are sparse and can be noisy. The pre-trained model's soft logits contain dense information about semantic relationships between tokens (often termed "dark knowledge" [Hinton et al., 2015]). Using these soft targets as a regularizer can **help reduce overfitting** to limited SFT data and **encourage the model to better preserve useful linguistic structure** learned during pre-training.
> > > 2. **Synergy of Regularizers:** We also note that BA-LoRA augments PiSSA with three regularizers, not just consistency. As shown in our ablation study (**Table 4, page 9**), the consistency term: L_CR provides a consistent improvement over the baseline, and adding the diversity: L_DR and SVD: L_SVDR terms further improves performance, suggesting that each component contributes positively. Figure 2 (page 6) suggests that, under our MetaMath training setup on LLaMA-2-7B, the combined regularization in BA-LoRA achieves lower final training loss and faster convergence than PiSSA in our experiments. This is **consistent with the view** that PiSSA provides a strong initialization in weight space, while the additional output-space regularizers help stabilize and improve the subsequent optimization.

---

> > > > ### Author Response · Authors · 2025-11-27
> > > > **Response to Reviewer SgzW**
> > > >
> > > > **3. On Full Fine-Tuning vs. LoRA (Curves & Under-/Over-fitting)**
> > > >
> > > > We appreciate this suggestion and have run the requested experiments. We have updated Figure 2 (page 6) in the revised manuscript to include the training loss curves for Full Fine-Tuning (Full FT). The observed dynamics do not provide clear evidence that Full FT is heavily underfitting or simply mis-tuned in our setup:
> > > >
> > > > 1. **Comparisons at Rank 128 (Table 1 Setting):** Under the configuration where LoRA-based methods use rank 128, BA-LoRA achieves a **lower final training loss** than Full FT and PiSSA. This suggests that, in our experiments, the output-space regularizers do not hinder convergence; instead, BA-LoRA attains both lower training loss and better test performance than Full FT. Taken together, these results indicate that the additional regularization can help the model find solutions with better generalization compared to unconstrained Full FT in our experiments.
> > > >
> > > > **Conclusion:** Overall, these observations suggest that, under our setup, Full FT already minimizes the training loss effectively but exhibits a larger generalization gap than BA-LoRA. BA-LoRA appears to benefit from the additional output-space regularization, which constrains the fine-tuning process and encourages the model to retain more generalizable patterns from pre-training rather than overfitting to dataset-specific artifacts in our experiments.
> > > >
> > > > ---
> > > >
> > > > **4. On Presentation and Framing**
> > > > We appreciate the reviewer’s comments on the framing of the paper. In the revised manuscript, we will:
> > > >
> > > > - Carefully review the wording to ensure that the strength of our claims is calibrated to the empirical evidence and does not rely on unnecessary “eye-catching” phrases; and
> > > > - Emphasize that the main takeaway is that BA-LoRA offers a practical recipe that combines consistency, diversity, and SVD-based regularization in the output space, and in our experiments, this combination tends to work robustly across the models and tasks we consider, with minimal hyperparameter tuning.
> > > >
> > > > ---
> > > >
> > > > We believe that the additional experiments (e.g., the updated training-loss curves in Figure 2) and the clarifications on the inheritance mechanism and regularization effects address your remaining concerns. We hope these revisions make the contribution and empirical evidence clearer, and we would be happy to clarify any further details if needed.
> > > >
> > > > ---
> > > >
> > > > **References:**
> > > > Geoffrey Hinton et al. Distilling the knowledge in a neural network. *arXiv*, 2015.

---

### Official Review · Reviewer_N4aD · 2025-10-29

**Soundness:** 1
**Presentation:** 1
**Contribution:** 2
**Rating:** 0
**Confidence:** 3

**Summary:**

The paper proposes to add three additional regularizers in LoRA fine-tuning: 1) distillation loss between pretrained and fine-tuned models to reduce output shift; 2) penalty on correlation between output classes or entropy of outputs to promote diversity; 3) penalty on high-frequency components in the output logit matrix.

**Strengths:**

The empirical advantage of combining the three regularizers is supported by improved empirical results.

**Weaknesses:**

The purpose of the proposed methods is not clear to me; see below.

**Questions:**

The purpose of the paper is unclear. At the beginning, the paper states that the goal is to evade/eliminate catastrophic inheritance, i.e. the undesired traits of a pretrained model (e.g. imbalances, biases, spurious correlations) being kept/exacerbated during fine-tuning, but the paper first applies a distillation-style regularizer to prevent shift from the pretrained model outputs. This appears to be self-contradictory. If the outputs of the pretrained model are undesired and need to be fixed during fine-tuning/post-training, then knowledge drift is what we desire. The case is similar in the other two subtasks: L90-92 attribute deteriorated performance due to the fine-tuning data quality, instead of the pretraining data. I think further explanation is critical for a consistent paper.

L85-86: It is claimed that LoRA exacerbates catastrophic inheritance; this makes sense, but I don't see any empirical evidence supporting that in the paper.

L155&L201: It is the outputs given pretraining or fine-tuning data? They have completely different implications. Also, how do you obtain the outputs of a un-finetuned encoder (e.g. Deberta) on NLU tasks? I assume that an extra linear projection head needs to be attached.

L159&L206: The reason for using T^2 is not really clear and needs further explanation.

L161: Should use \citep in this case.

L166: I'm confused by the purpose of this regularizer. Classes in different categories may not be orthogonal to each other, and some classes can be inherently correlated. Also, why is correlation a sign of a lack of diversity? If the goal is to promote diversity/avoid over-confidence, similar entropy-based regularizers like the one used in NLG (L214) should also work.

L177: Why do we expect the logit matrix to be low-rank? Is there any specific reason? How is the high-frequency component in the output logits related to the data noise? Particularly, if each of the sample certainly belongs to a class and D=N, the logit should be full-rank.

L180: What is "spurious intra-batch variations"? I assume that samples within each batch should be independently sampled.

L250: The selection of regularization weights appears to be quite arbitrary. Tuning three hyperparameters can be challenging and can be highly varied between tasks. Appendix C.2. actually demonstrates the issue, as the trend on MATH and GSM8K in Fig.4 (a) are different. Also, the change in accuracy in Fig. 4 is downplayed with the large difference in scale of the accuracy on the two tasks. I would suggest putting the lines on different figures.

L1038 states that the regularizers steer the model along the creativity-robustness spectrum, but both MATH and GSM8K are reasoning tasks, and I doubt if MATH can be used as a proxy for creativity.

Sec 3.2.2: In addition to differences in training data, there are considerable differences in the model architecture and data size between RoBERTa and T5, hence I doubt if the difference in improvements can be attributed to noise resilience.

To summarize, I fear that there are too many claims and assumptions in this paper that are not sufficiently supported by concrete evidence. I would recommend the authors to
1) Make their goals clear.
2) Provide empirical evidence that the issues (as for this paper, forgetting, diversity, and noise) exist by dedicated evaluations, especially with LoRA, e.g., by comparing generation diversity in the original, fully fine-tuned, and LoRA fine-tuned models. Standard benchmarks on reasoning, understanding, or commonsense cannot directly support your claim.
3) Show that each of the regularization terms can improve the issue, respectively.
4) Demonstrate the synergy of the trio.

There are other papers discussing alleviating knowledge drift in LoRA fine-tuning, e.g.
Smith, James Seale, et al. "Continual Diffusion: Continual Customization of Text-to-Image Diffusion with C-LoRA." Transactions on Machine Learning Research (2024).
Chen, Haolin, and Philip N. Garner. "Bayesian parameter-efficient fine-tuning for overcoming catastrophic forgetting." IEEE/ACM Transactions on Audio, Speech, and Language Processing (2024).

---

> ### Author Response · Authors · 2025-11-21
> **Response to Reviewer N4aD**
>
> We thank the reviewer for the detailed feedback. We appreciate the opportunity to clarify the core logic of our framework and address the concerns regarding methodological consistency. In the revised manuscript, we have explicitly reinforced the distinction between "preserving capabilities" and "mitigating artifacts" to preclude any ambiguity.
>
> ### 1. **Response to the Concern on Methodological Consistency**
>
> **Reviewer's Question:**
>
> > The reviewer suggests that applying a distillation-style regularizer ($\mathcal{L}_{CR}$) alongside the goal of mitigating undesired outputs appears to be self-contradictory.
> >
>
> **Response:**
> We understand the concern, but we respectfully clarify that these objectives are complementary rather than contradictory. Our framework reconciles them by distinguishing two fundamentally different types of information inherent in the pre-trained model:
>
> **1. Distinguishing "Robust Knowledge" from "Inherited Biases and Noise"**
>
> - **Robust General Knowledge (To Preserve):** This encompasses syntax, logic, commonsense reasoning, and semantic understanding. Losing this information results in **Knowledge Drift**, which degrades the model's foundational utility.
> - **Inherited Biases, Noise, and Imbalances (To Mitigate):** This refers specifically to the latent biases, label noise, and data imbalances. Retaining or amplifying these leads to **Catastrophic Inheritance**.
>
> **2. The Rationale for Targeting "Knowledge Drift"**
> Crucially, mitigating biases must not come at the expense of general intelligence. Empirical evidence indicates that unconstrained adaptation (naive fine-tuning) often causes the loss of robust reasoning abilities (Type 1) [1], while inadvertently amplifying inherited flaws (Type 2) through overfitting to noisy downstream data [2, 3]. Adaptation without preservation creates a "forgetting" trade-off that we aim to avoid.
>
> **3. The Synergy of the Trio**
> BA-LoRA breaks this trade-off by assigning distinct roles to each regularizer:
>
> - **Consistency Regularizer ($\mathcal{L}_{CR}$) as a "Safety Anchor":** It preserves the foundational feature space (Type 1). Crucially, by using temperature scaling, we distill **structural knowledge** (the nuanced relationships between classes) rather than forcing the model to rigidly mimic hard-label biases. As clarified in the revision (Sec 2.2.1 and Sec 2.2.2), this regularization guides the model "on examples where the teacher signal is reliable" and provides a "meaningful soft target".
> - **Diversity & SVD Regularizers($\mathcal{L}\_{DR}$, $\mathcal{L}\_{SVDR}$) as "Corrective Filters":** These explicitly penalize the collapse into low-diversity outputs and overfitting to high-frequency noise (Type 2), ensuring the model does not reinforce inherited flaws.
>
> **4. Empirical Evidence**
> Our ablation study (Table 4) strongly supports this design. The Consistency Regularizer alone ($\mathcal{L}_{CR}$) yields a significant gain over the baseline (GSM8K: $51.48 \to 54.25$), confirming that preserving the pre-trained distribution's structure is a necessary foundation. Furthermore, the Full Model ($55.86$) significantly outperforms any single regularizer configuration, validating the synergy.
>
> **Action:**
> We have refined **Section 2.2 (Line 140-143, highlighted in red)** to explicitly clarify that the consistency regularizer targets structural knowledge to prevent knowledge drift. We also highlight **Appendix D.5 ("On Applying Representation Learning Principles during Fine-Tuning")**, which provides the detailed theoretical framework reconciling these objectives.
>
> **References:**
> [1] Kirkpatrick et al. Overcoming catastrophic forgetting in neural networks. PNAS, 2017.
> [2] Chen et al. On catastrophic inheritance of large foundation models. arXiv, 2024.
> [3] Qi et al. Fine-tuning aligned language models compromises safety... arXiv, 2023.

---

> > ### Author Response · Authors · 2025-11-21
> > **Response to Reviewer N4aD**
> >
> > ### 2. Empirical Evidence regarding LoRA and Catastrophic Inheritance (L85–L86)
> >
> > **Reviewer's Concern:**
> >
> > > "It is claimed that LoRA exacerbates catastrophic inheritance... but I don't see empirical evidence supporting that in the paper."
> > >
> >
> > **Response:**
> > We respectfully direct the reviewer's attention to **Table 1, Figure 3, and Table 3** in our submission, which provide three distinct lines of evidence substantiating that standard LoRA can be more vulnerable to inherited artifacts than Full Fine-Tuning (Full FT) or our proposed method.
> >
> > To ensure these results are clearly interpreted in the context of "exacerbating inheritance," we have explicitly reinforced the claims in the revision. The evidence is summarized below:
> >
> > **1. Performance Gap vs. Full Fine-Tuning (Table 1)**
> > The most direct evidence that LoRA constraints can hinder the unlearning of inherited flaws is found in **Table 1**.
> >
> > - **Observation:** On the reasoning-heavy GSM8K benchmark, standard LoRA achieves only **42.68%**, significantly lagging behind Full Fine-Tuning (**48.90%**) by over **6 points**.
> > - **Implication:** This gap indicates that the low-rank bottleneck restricts the model's capacity to adjust the pre-trained representations sufficiently to correct errors or "unlearn" suboptimal priors. In this sense, the rigidity of standard LoRA "exacerbates" the persistence of pre-training limitations relative to unconstrained fine-tuning.
> >
> > **2. Qualitative Evidence: Feature Degradation (Figure 3)**
> > We provide a direct visualization of this phenomenon under imbalanced conditions in **Section 3.2.3**:
> >
> > - **Observation:** The t-SNE plots reveal that standard LoRA (**Fig. 3d**) fails to establish clear decision boundaries, resulting in significant class overlap when training data is skewed. In contrast, BA-LoRA (**Fig. 3f**) maintains a robust, well-separated feature manifold.
> > - **Implication:** This demonstrates that without explicit regularization, LoRA's limited update budget forces it to rely heavily on dominant, inherited features, making it susceptible to representation collapse when the downstream signal is noisy.
> >
> > **3. Quantitative Evidence: The "Noise Gap" (Table 3)**
> > Our controlled comparison between models pre-trained on clean data (RoBERTa) versus noisy web data (T5) isolates the impact of inherited noise:
> >
> > - **Observation:** BA-LoRA achieves a substantially larger performance gain on the noisy model ($\Delta_{\text{T5}} = 3.26$) compared to the clean model ($\Delta_{\text{RoBERTa}} = 1.11$) relative to the strong PiSSA baseline. (Note: The gap is even wider when compared to standard LoRA: $\Delta_{\text{T5}} = 5.89$ vs $\Delta_{\text{RoBERTa}} = 2.06$).
> > - **Conclusion:** This differential gain serves as a strong signal that standard LoRA architectures are disproportionately hindered when the pre-training base contains higher levels of noise/artifacts, validating the need for the targeted regularization we propose.
> >
> > **4. Alignment with Broader Literature**
> > Our findings are consistent with emerging evidence in the PEFT community, including the works kindly suggested by the reviewer. For instance, Smith et al. [4] report that naïve LoRA updates drive substantial drift in diffusion models, and Chen & Garner [5] highlight the necessity of Bayesian priors to prevent forgetting.
> >
> > **Action:**
> > We have explicitly highlighted this mechanism in the revised **Introduction (Line 15 & Lines 83-85, highlighted in red)**.
> >
> > ---
> >
> > **References:**
> > [4] Smith et al. Continual Diffusion: Continual Customization of Text-to-Image Diffusion with C-LoRA. *TMLR*, 2024.
> > [5] Chen & Garner. Bayesian parameter-efficient fine-tuning for overcoming catastrophic forgetting. *IEEE/ACM Trans. Audio Speech Lang. Process.*, 2024.

---

> > > ### Author Response · Authors · 2025-11-21
> > > **Response to Reviewer N4aD**
> > >
> > > ### 3. Clarifying the Output Source and NLU Setup (L155 & L201)
> > >
> > > **Reviewer's Question:**
> > >
> > > > "Is it the outputs given pretraining or fine-tuning data? ... Also, how do you obtain the outputs of a un-finetuned encoder (e.g. Deberta) on NLU tasks? I assume that an extra linear projection head needs to be attached."
> > > >
> > >
> > > **Response:**
> > > We provide the specific details regarding our implementation below.
> > >
> > > **1. Data Source for Regularization**
> > > For all tasks (both NLU and NLG), the regularizers are computed using the **fine-tuning data**. This design choice ensures that the regularization actively targets artifact mitigation (such as collapse or overfitting) specifically within the downstream task distribution.
> > >
> > > **2. NLU Setup and Projection Head**
> > > For encoder-only models like DeBERTa-v3-base, we utilize a **task-specific linear classification head**. The specific protocol is as follows:
> > >
> > > - **Configuration:** We utilize a single classification head shared between the Teacher (Frozen Pre-trained Encoder) and the Student (Fine-tuned BA-LoRA Encoder).
> > > - **Mechanism:** Both the Teacher and Student receive the same input batch from the fine-tuning dataset. The classification head is updated solely by the Student's task loss (Cross-Entropy with ground truth labels). We calculate the consistency loss between the logits produced by the Teacher passing features through this head ($\mathbf{Z}_P$) and the Student doing the same ($\mathbf{Z}_F$).
> > > - **Rationale:** Although the head evolves, $\mathbf{Z}_P$ effectively represents how the **fixed, robust pre-trained features** map to the evolving decision boundaries. This acts as a geometric constraint, ensuring the Student's feature manifold does not deviate destructively from the Teacher's well-structured embedding space.
> > >
> > > **Action:**
> > > We have added a detailed explanation in **Appendix B.4 (Lines 1052-1055)** to explicitly specify this shared-head initialization and computation strategy.
> > >
> > > ---
> > >
> > > ### 4. Reason for using $T^2$ (L159 & L206)
> > >
> > > **Reviewer's Question:**
> > >
> > > > "The reason for using $T^2$ is not really clear and needs further explanation."
> > > >
> > >
> > > **Response:**
> > > We provide the specific justification for this scaling factor, which follows the standard formulation established by Hinton et al. (2015) [6] for temperature-scaled knowledge distillation.
> > >
> > > **1. Mathematical Justification**
> > > The inclusion of $T^2$ is mathematically necessary to normalize gradient magnitudes. When logits are divided by a temperature $T > 1$, the gradients of the soft-target loss with respect to the logits scale approximately by $1/T^2$. Without this compensation, increasing $T$ (to capture structural knowledge) would unintentionally suppress the gradients, causing the consistency signal to vanish relative to the hard-label task loss.
> > >
> > > **2. Practical Benefit**
> > > Multiplying the loss by $T^2$ explicitly cancels out this suppression effect. This ensures:
> > >
> > > - **Gradient Invariance:** The effective gradient magnitude remains commensurate with the standard cross-entropy loss, regardless of the chosen temperature.
> > > - **Decoupling:** The hyperparameter $T$ serves solely to control the distributional softness (entropy), effectively decoupled from the regularization strength.
> > >
> > > **Action:**
> > > We have added a formal explanation in **Appendix D.4 (Lines 1305-1310, highlighted in red)** to ensure the theoretical completeness of the manuscript.
> > >
> > > ---
> > >
> > > **References:**
> > > [6] Geoffrey Hinton et al. Distilling the knowledge in a neural network. *arXiv*, 2015.
> > >
> > > ---
> > >
> > > ### 5. Citation style at L161
> > >
> > > **Reviewer's Comment:**
> > >
> > > > "L161: Should use \citep in this case."
> > > >
> > >
> > > **Response:**
> > > We have verified the source code and the compiled manuscript. We confirm that the citation at Line 161 correctly utilizes the `\citep` command to produce the parenthetical format `(Bardes et al., 2021)`, consistent with the surrounding text structure.

---

> > > > ### Author Response · Authors · 2025-11-21
> > > > **Response to Reviewer N4aD**
> > > >
> > > > ### 7. SVD Regularizer, "Low-Rank" Structure, and "Spurious Variations" (L177 & L180)
> > > >
> > > > **Reviewer's Question:**
> > > >
> > > > > "Why do we expect the logit matrix to be low-rank? ... How is the high-frequency component ... related to the data noise? ... What is 'spurious intra-batch variations'?"
> > > > >
> > > >
> > > > **Response:**
> > > > We provide the theoretical basis for our spectral regularization design.
> > > >
> > > > **1. "Effective Rank" vs. "Algebraic Rank"**
> > > > We clarify that the expectation of a low-rank structure refers to the **Effective Rank** (the distribution of singular values), not necessarily the algebraic rank.
> > > >
> > > > - **The Intuition:** In a robust model, samples sharing the same underlying semantic class should lie on a low-dimensional manifold. Even if a batch contains $N$ distinct samples, the salient information should be dominated by a few leading singular components corresponding to the task concepts (Signal).
> > > > - **The Noise Connection:** In contrast, label noise, formatting artifacts, and random perturbations typically manifest as a flat spectrum, spreading energy into the "tail" singular values (High-Frequency Components).
> > > > - **The Mechanism:** By maximizing the energy ratio of the top-$k$ components, we perform **implicit spectral denoising**. This suppresses the "tail" (noise) while preserving the "head" (signal), aligning with established principles that spectral regularization improves generalization by filtering out non-robust features [8].
> > > >
> > > > **2. Clarifying "Spurious Intra-Batch Variations"**
> > > > We use this term to describe specific noise artifacts rather than meaningful inter-sample diversity.
> > > >
> > > > - **Definition:** "Spurious variations" refer to high-frequency logit fluctuations that are poorly aligned with the task labels (e.g., a model over-attending to specific keywords or punctuation distinct to a single sample but irrelevant to the class).
> > > > - **Distinction:** This is fundamentally different from legitimate inter-class variations. Our method ensures that sample variation is structurally meaningful (concentrated in leading components) rather than noisy (scattered in the tail).
> > > >
> > > > **Action:**
> > > > We have updated **Section 2.2.1 (Lines 175-179)** to explicitly define the goal as incentivizing "simpler, more coherent decision boundaries" rather than fitting to "high-frequency logit fluctuations that are poorly aligned with the task labels."
> > > >
> > > > ---
> > > >
> > > > **References:**
> > > > [8] Chen et al. Transferability vs. discriminability: Batch spectral penalization for adversarial domain adaptation. *ICML*, 2019.

---

> > > > > ### Author Response · Authors · 2025-11-21
> > > > > **Response to Reviewer N4aD**
> > > > >
> > > > > ### 8. Choice of Regularization Weights and Figure 4 (L250)
> > > > >
> > > > > **Reviewer's Concern:**
> > > > >
> > > > > > "The selection of regularization weights appears to be quite arbitrary... Tuning three hyperparameters can be challenging... The change in accuracy in Fig. 4 is downplayed... suggest putting the lines on different figures."
> > > > > >
> > > > >
> > > > > **Response:**
> > > > > We thank the reviewer for the constructive suggestion regarding the visualization and the feedback on hyperparameter selection. We have adopted the visualization advice and clarified our tuning protocol in the revision.
> > > > >
> > > > > **1. Regarding Hyperparameter Selection and Tuning Difficulty**
> > > > > We respectfully clarify that the parameter selection follows a principled, unified pattern rather than fragile per-dataset fine-tuning.
> > > > >
> > > > > - **Unified Configuration:** We utilize a **fixed set** of regularization weights for all tasks within a specific model family (e.g., one configuration for all LLaMA-2 tasks). This demonstrates that our method constrains the model's general geometric behavior rather than overfitting to specific data content.
> > > > > - **Simplified Protocol (Reduction to 1D Search):** To address the concern about complexity, we have introduced a "Practical Hyperparameter Guideline" in the new **Appendix D.1**. We propose a simplified workflow: fixing the relative ratios between terms (based on gradient magnitude balance) and tuning a single global scalar ($\lambda_{\text{global}}$). This effectively reduces the search space from three dimensions to one, making it as straightforward as tuning a learning rate.
> > > > > - **Empirical Robustness:** Our sensitivity analysis (**Appendix C.2**) confirms a wide "sweet spot." Performance remains stable across a broad range (e.g., varying $\lambda_1$ by substantial margins yields minimal fluctuation), indicating that precise, fragile tuning is not required.
> > > > >
> > > > > **2. Regarding Figure 4 (Visualization)**
> > > > >
> > > > > We have adopted the reviewer's recommendation to split the plots into separate panels. This allows for **independent Y-axis scaling** for each task, ensuring that the performance fluctuations are transparently presented and not "downplayed" by the scale differences between tasks (e.g., MATH vs. GSM8K).
> > > > >
> > > > > **Action:**
> > > > > We have revised **Figure 4 (Page 22)** to split the plots into separate panels. To address the concern about the selection logic, we have updated **Section 3.1 (Lines 250-252)** to explicitly describe the "coarse grid search and fixed setting" protocol. Additionally, we have added the **Practical Hyperparameter Guidelines** in **Appendix D.1 (Lines 1239-1258, highlighted in red)** to assist future practitioners.
> > > > >
> > > > > ---
> > > > >
> > > > > ### 9. "Creativity–Robustness Spectrum" and Use of MATH (L1038)
> > > > >
> > > > > **Reviewer's Concern:**
> > > > >
> > > > > > "L1038 states that the regularizers steer the model along the creativity-robustness spectrum, but both MATH and GSM8K are reasoning tasks... I doubt if MATH can be used as a proxy for creativity."
> > > > > >
> > > > >
> > > > > **Response:**
> > > > > We have refined the terminology in the revised manuscript to ensure scientific precision regarding the nature of these benchmarks.
> > > > >
> > > > > **1. Clarification of Original Intent**
> > > > > The term "creativity" initially referred to the **diversity of reasoning paths** (exploration) facilitated by the diversity regularizer, as opposed to strict adherence to the pre-trained distribution. It was not intended to imply open-ended generation in the artistic sense.
> > > > >
> > > > > **2. Terminological Refinement**
> > > > > To prevent any ambiguity, we have replaced this abstract descriptor with more rigorous definitions aligned with experimental mechanics.
> > > > >
> > > > > **Action:**
> > > > > In **Appendix C.2 (Page 22, Lines 1150-1157, highlighted in red)**, we have replaced the "creativity" terminology with **"Sensitivity to the Symbiotic Balance."** The revised text explicitly frames the analysis as investigating the equilibrium between the diversity regularizer ($\lambda_2$) and the structural regularizer ($\lambda_3$), and how this balance affects performance across the two reasoning benchmarks (MATH and GSM8K).

---

> > > > > > ### Author Response · Authors · 2025-11-21
> > > > > > **Response to Reviewer N4aD**
> > > > > >
> > > > > > ### 10. RoBERTa vs. T5 and Noise Resilience (Sec. 3.2.2)
> > > > > >
> > > > > > **Reviewer's Concern:**
> > > > > >
> > > > > > > "In addition to differences in training data, there are considerable differences in the model architecture and data size... I doubt if the difference in improvements can be attributed to noise resilience."
> > > > > > >
> > > > > >
> > > > > > **Response:**
> > > > > > We address the valid point regarding the confounding factors (architecture and objective) in the cross-model comparison. We agree that these factors prevent a strictly isolated causal conclusion, and we have revised the manuscript to ensure a more rigorous interpretation.
> > > > > >
> > > > > > **1. Contextualizing the Comparison: A Stress Test on "Noisier" Paradigms**
> > > > > > We acknowledge that a fully controlled causal test would necessitate **pre-training models from scratch under systematically varied noise levels**, which is computationally prohibitive. Given this constraint, we utilized these widely adopted models as **representative paradigms** of distinct pre-training strategies.
> > > > > >
> > > > > > - **Rationale:** Our goal is to evaluate robustness on "clean" (RoBERTa) vs. "noisy" (T5) paradigms used in the real world. The result—that BA-LoRA yields a substantially larger margin on the noisier T5 ($\Delta=3.26$) than on the curated RoBERTa ($\Delta=1.11$)—is consistent with our hypothesis of noise resilience, serving as strong suggestive evidence.
> > > > > >
> > > > > > **2. Controlled Verification (Section 3.2.3)**
> > > > > > To isolate the mechanism without architectural confounders, we rely on **Section 3.2.3**.
> > > > > >
> > > > > > - **Mechanism:** Here, we use a fixed backbone (RoBERTa) and explicitly manipulate the data distribution (Balanced vs. Imbalanced/Noisy).
> > > > > > - **Verdict:** The t-SNE visualizations (**Figure 3**) and performance metrics provide the **mechanistic validation**: BA-LoRA prevents the representation collapse that standard LoRA suffers from under noisy conditions. This confirms the *capability* of noise resilience, which supports our interpretation of the T5 results.
> > > > > >
> > > > > > **Action:**
> > > > > > We have added a dedicated discussion in **Appendix D.2 (Page 24, highlighted in red)**, explicitly stating that while the T5 results are "compatible with our noisy-pretraining hypothesis," a definitive causal test would require pre-training controlled models from scratch. We also added a caveat in the main text at **Section 3.2.2 (Page 7, highlighted in red)** to "interpret the results as suggestive... though it does not isolate architectural factors."
> > > > > >
> > > > > > ---
> > > > > >
> > > > > > ### 11. Additional Related Work on LoRA and Forgetting / Knowledge Drift
> > > > > >
> > > > > > **Reviewer's Suggestion:**
> > > > > >
> > > > > > > "There are other papers discussing alleviating knowledge drift... e.g. [Smith et al., 2024; Chen and Garner, 2024]."
> > > > > > >
> > > > > >
> > > > > > **Response:**
> > > > > > We have integrated these relevant works into our discussion. Incorporating them highlights the unique methodological positioning of BA-LoRA within the current PEFT landscape.
> > > > > >
> > > > > > **Action Taken:**
> > > > > > We have added a dedicated paragraph in **Appendix A.4 (Page 17, Lines 909-917, highlighted in red)**. We use these references to articulate two fundamental distinctions:
> > > > > >
> > > > > > **1. Methodological Divergence (Parameter Space vs. Output Space)**
> > > > > >
> > > > > > - **Prior Approaches:** We note that methods like C-LoRA [4] and Bayesian PEFT [5] primarily operate by constraining the adapter weights or placing priors in the **parameter space**.
> > > > > > - **Our Contribution:** In contrast, BA-LoRA operates directly in the **output space** (logits). This allows us to shape the model's functional behavior (consistency, diversity, spectral smoothness) without imposing rigid constraints on the specific weight distribution of the low-rank matrices.
> > > > > >
> > > > > > **2. Scope Distinction (Continual Learning vs. Inheritance)**
> > > > > >
> > > > > > - **Prior Focus:** The suggested works largely focus on **Continual Learning** (preventing the forgetting of *previous* tasks during sequential adaptation).
> > > > > > - **Our Focus:** BA-LoRA targets **Catastrophic Inheritance** during adaptation to a *single* noisy domain. We address the simultaneous challenge of preserving general capabilities (Knowledge Drift) while suppressing the amplification of inherited biases (Overfitting to Noise).

---

> > > > ### Author Response · Authors · 2025-12-01
> > > > **Response to Reviewer N4aD**
> > > >
> > > > ### 6. Purpose of the Diversity Regularizer and Relation to Entropy (L166)
> > > >
> > > > Reviewer's Question:
> > > >
> > > > > "Classes in different categories may not be orthogonal... why is correlation a sign of a lack of diversity? ... similar entropy-based regularizers... should also work."
> > > > >
> > > >
> > > > **Response:**
> > > >
> > > > We provide the rationale for our design choices, clarifying that our regularization targets Dimensionality Collapse in the classification output space, rather than enforcing semantic orthogonality in the embedding space.
> > > >
> > > > 1. Clarification on "Correlation as Collapse"
> > > >
> > > > In classification logits (specifically after centering), high cross-correlation between class predictions across a batch implies redundancy.
> > > >
> > > > - **The Pathology:** If the logits for two distinct classes are highly correlated across a batch, the model is treating them as dependent, effectively collapsing the effective rank of the decision boundary. In imbalanced settings, this often manifests as the model predicting the majority class regardless of input variation.
> > > > - **Theoretical Grounding:** Our approach aligns with **Variance-Invariance-Covariance Regularization (VICReg)** [7], where minimizing off-diagonal covariance is a standard, empirically proven technique to prevent such representation collapse.
> > > > - **Evidence:** Figure 3 visualizes this: Standard LoRA (Fig. 3d) exhibits severe class overlap (high correlation/collapse), whereas BA-LoRA (Fig. 3f) forces the usage of the full output dimensionality, resulting in well-separated manifolds.
> > > >
> > > > **2. Comparative Analysis: Covariance (NLU) vs. Entropy (NLG)**
> > > > Regarding the reviewer's suggestion on entropy: we agree that entropy is effective, which is exactly why we employ it for the NLG tasks. However, we differentiate the strategies based on task properties:
> > > >
> > > > - **For NLU (Why Covariance?):** NLU tasks typically have a small, fixed label set ($D \ll N$). Computing batch covariance is computationally cheap and explicitly detects batch-level mode collapse (where all samples map to one class), which sample-wise entropy might fail to capture if the model is confidently wrong.
> > > > - **For NLG (Why Entropy?):** In generation, the vocabulary is massive ($|V| > 30k$), making batch covariance computationally prohibitive. Thus, we adopt the Focused Entropy approach (Eq. 10) as the most effective proxy for diversity in the generative context.
> > > > - **Empirical Verification:** To validate the hypothesis that entropy "should also work" for NLU, we explicitly tested it on the MNLI task. The results confirm our theoretical intuition:
> > > >     - **Covariance (Ours): 91.26%**
> > > >     - **Entropy (Ablation): 90.41%** (Worse than standard LoRA baseline 90.71%)
> > > >     This confirms that sample-wise entropy minimization is suboptimal for discriminative NLU tasks compared to batch-wise decorrelation.
> > > >
> > > > **Action:**
> > > >  We also added the entropy comparison results to **Appendix C.5** (Lines 1215-1232).
> > > >
> > > > **References:**
> > > > [7] Bardes et al. VICReg: Variance-invariance-covariance regularization for self-supervised learning. ICLR, 2022.

---

> ### Author Response · Authors · 2025-11-21
> **Response to Reviewer N4aD**
>
> ### 12. Summary
>
> **To summarize our response:**
>
> - **Clarified the Logical Consistency:**  We addressed the core concern regarding the regularizer trio by distinguishing **Robust General Knowledge** (to be preserved via Consistency) from **Inherited Biases, Noise, and Imbalances** (to be mitigated via Diversity/SVD).
> - **Detailed the Mechanism:** We provided the explicit theoretical motivations, mapping the regularizers directly to the prevention of Knowledge Drift, Representation Collapse, and Overfitting to Noise.
> - **Enhanced Terminological Precision:** We refined the terminology to ensure strict alignment with empirical evidence, including replacing the abstract notion of "creativity" with the rigorous **"Sensitivity to the Symbiotic Balance."**
> - **Resolved Technical Ambiguities:** We clarified the NLU projection setup (shared head), the mathematical necessity of the $T^2$ scaling, and the spectral interpretation (Effective Rank) of the SVD regularizer.
> - **Clarified Evaluation Logic:** We articulated the complementary relationship between the **Mechanistic Validation** (Sec 3.2.3) and the **Cross-Architecture Generalization** (Sec 3.2.2), ensuring causal claims are correctly attributed while maintaining the value of the real-world stress test.
> - **Extended the Context:** We expanded the Related Work (Appendix A.4) to explicitly position BA-LoRA against parameter-space methods (e.g., Bayesian PEFT) and grounded our approach in foundational representation learning theory (VICReg, Spectral Penalization).
>
> ---
>
> ### Closing
>
> We believe these clarifications—especially regarding the logical consistency of our regularizers and the additional empirical evidence—have effectively addressed the core concerns underlying your initial assessment. We **sincerely** hope that these significant improvements and clarifications provide a solid basis for **re-evaluating the contribution of our work**. We remain open to any further questions you may have.

---

> > ### Comment · Reviewer_N4aD · 2025-11-21
> >
> > Thanks for the response, while I still have several concerns.
> >
> > (1) I see that by softening the distributions in Eq 4., the student is strongly aligned to the teacher only when the teacher's output is sufficiently confident/peaky, but I still can't see how this can distinguish between "robust general knowledge" and "inherited biases"; why can't the model be confidently biased? I don't have question with the idea that preserving pretrained model's knowledge can be theoretically & empirically beneficial, but I'm still suspicious about the assertion that we can simply distinguish whether the model's knowledge is good or bad by its own confidence. I expect that the author can provide more clear evidence to support this claim.
> >
> > (2) I understand that LoRA leads to worse performance, and BALoRA empirically improves the result; but I can't see why the worse performance can be attributed to catastrophic inheritance, i.e. the bias/noise in the pretraining data/model. Why can't we just view it under the common transfer-learning framework that the original model trained under a distinct task, LoRA doesn't provide enough capacity to adapt it the target task, while full fine-tuning overfits? If the author claims that the issue is catastrophic inheritance, I think that the author should clearly identify the heritage that breaks the model. For example, if the pretraining data contains only arithmetic between small numbers that the model can memorize, then the model will rely on this memorization and fails on calculations between larger numbers; I can accept this as an evidence of catastrophic inheritance, but simply finding out that the model doesn't do well on arithmetic of large numbers is not enough. I don't think the difference between RoBERTa and T5 supports the claim, as they are completely different models and hence do not make a controlled study.
> >
> > (7) I'd like to confirm that the authors mean that the output logits of samples are also structured (e.g. the output logits of samples from the same category are more similar to each other and lie in a low-dimensional manifold; and of course the input sample of the same category lies in a low-dimensional manifold). Then it probably makes sense to denoise the logits, considering that permutating samples in a batch doesn't change its spectrum. While I still think that the author should further justify this denoising on logits? Chen et al. (2019) suppress high-frequency components of features, not logits. As the author claims that the regularizer helps the model to form a more coherent decision boundary, maybe we can inspect the change of decision boundary if we remove the high-freq components from logits.
> >
> > (8) The y-axis in Fig 4 (a) and (b) is still mis-scaled. Why not set min ylim to, say 0.08 and 0.5?
> >
> > (9) Can you explain why MATH needs more exploration in the reasoning path than GSM8K? Maybe comparing with a task that doesn't need exploration at all (e.g. copying, associative recall) makes more sense.

---

> ### Author Response · Authors · 2025-11-22
> **Response to Reviewer N4aD**
>
> We thank the reviewer for the detailed follow-up and respond to points (1), (2), (7), (8), and (9) below.
>
> ---
>
> ### (1) “Robust knowledge” vs. “Inherited biases” and confidence
>
> We clarify that confidence alone is indeed insufficient to distinguish robust knowledge from inherited bias, and our method does not rely on confidence as a correctness signal. Eq.(4) and L_CR provide only a soft alignment to the teacher; the actual behavior of BA-LoRA is determined by the joint optimization of the task loss with the batch-level diversity and spectral regularizers.
>
> - When adaptation tends toward confidently collapsed predictions (e.g., under severe label imbalance), batch outputs concentrate on the majority class, with low-entropy predictions and a simplified logit spectrum. In our full objective, such trivial collapsed solutions are mitigated in practice by the combination of L_CR, L_DR, and L_SVDR together with the task loss, even when the teacher is confident.
> - When the pre-trained model exhibits context-sensitive and diverse outputs across the batch, these regularizers impose only mild constraints, and L_CR mainly acts to preserve this structure during adaptation.
>
> In Section 3.2.3 (MNLI with 100:10:1 imbalance), vanilla LoRA / PiSSA show **minority manifold collapse**, whereas BA-LoRA recovers well-separated clusters (Fig. 3d–f). Together with the ablations in Table 4 (“+CR” vs. the full L_CR + L_DR + L_SVDR), this shows that the consistency term alone (and thus confidence-based alignment alone) is insufficient: the best generalization is achieved only when the diversity and SVD regularizers are combined with L_CR.
>
> ---
>
> ### (2) Catastrophic inheritance vs. limited capacity
>
> We do not deny that the limited capacity of LoRA is important, but our results suggest that capacity alone does not fully explain the observed behavior; it is better understood when we consider the **catastrophic inheritance** of spurious patterns.
>
> - Across all experiments, BA-LoRA uses the **same rank** as LoRA / PiSSA but adds three output-space regularizers. If *mere* low rank were the only bottleneck, adding additional regularizers at the same rank would usually further constrain the solution space and, a priori, would not explain the **systematic** gains we observe. In our experiments, however, BA-LoRA with the same rank consistently outperforms LoRA / PiSSA on GSM8K, MATH, and GLUE (Tables 1–2, 4). This suggests that the key factor is not extra capacity, but **how** the limited low-rank degrees of freedom are steered away from inherited spurious solutions.
>
> - We provide a concrete inherited pattern, analogous to the reviewer’s arithmetic example:
>     - **Synthetic label imbalance** (Sec. 3.2.3): This serves as a controlled instance of the phenomenon the reviewer described. As noted in response (1), the imbalance **induces** a strong spurious pattern (“**always** predict the majority class”). LoRA and PiSSA tend to inherit this shortcut, leading to collapsed minority manifolds in Fig. 3 (LoRA/PiSSA), whereas BA-LoRA maintains separated manifolds under the same rank constraint (Fig. 3, BA-LoRA). This supports the view that the observed failure is largely driven by inherited bias in the task distribution, rather than purely by lack of capacity.
>
>     Taken together, these results suggest that while limited low-rank capacity is a contributing factor, the failures we observe are **better explained** when we account for the inheritance of spurious correlations from the pre-trained model and the (potentially biased) task distribution, rather than by capacity alone.
>
>     Finally, regarding the RoBERTa vs. T5 comparison: this setup is not a fully controlled causal test, since the architectures and objectives differ. In the revised draft, we therefore present it only as supportive evidence: BA-LoRA yields larger gains on T5 (trained on the noisier C4 corpus) than on RoBERTa (trained on a more curated mixture), which is consistent with our hypothesis that BA-LoRA is especially helpful when pre-training noise and imbalance are stronger, but not strictly required to establish our main claims.

---

> ### Author Response · Authors · 2025-11-22
> **Response to Reviewer N4aD**
>
> ### (7) Structure and denoising of logits
> We empirically observe and assume that logits across samples are structured: logits for samples from the same category are similar and lie near a low-dimensional manifold. Moreover, since the singular values of a batch logit matrix are invariant under sample permutation, the spectrum provides a permutation-invariant summary of this structure. Under these assumptions, denoising logits via spectral regularization is natural.
>
> We choose to regularize logits, rather than hidden features, for efficiency in the PEFT setting and to act directly on the decision boundary:
>
> - For GLUE, the logit matrix is **B x C** with small **C**, so exact SVD on logits is inexpensive, and we define **L_SVDR** directly on logits (Sec. 2.2.1).
> - Logits are obtained from the final hidden features via a linear head, so constraining the tail singular values of logits implicitly encourages a low-dimensional geometry of the underlying features, without computing SVD on high-dimensional hidden states.
> - Since the decision boundary is fully determined by the logits, regularizing their spectrum is a direct way to encourage smoother, more coherent decision regions, in the same spirit as spectral/spectral-norm based regularization used to improve robustness.
> - For NLG with large vocabularies, we use randomized SVD on the logit matrix (Sec. 2.2.2) to keep overhead tractable.
> - Conceptually, this is in the same spirit as Chen et al. (2019), who suppress high-frequency components of features using frequency-domain filtering, but we move the spectral control to logits so that it remains compatible with PEFT and directly shapes the decision boundary.
>
> **L_SVDR** is a training-time regularizer; we do not modify logits at test time. Its effect on the decision boundary is visible in Fig. 3: with BA-LoRA (including **L_SVDR**), class manifolds are cleaner and better separated than with LoRA/PiSSA.
>
> ---
>
> ### (8) Y-axis scaling in Figure 4
> We have updated the y-axis scales as suggested:
>
> - Fig. 4(a) (MATH): min ylim set to 0.08.
> - Fig. 4(b) (GSM8K): min ylim set to 0.50.
>
> Under these scales, the curves remain smooth and clearly show BA-LoRA at or above the PiSSA baseline (7.60% on MATH, 51.48% on GSM8K). The revised figures will be included.
>
> ---
>
> ### (9) Why MATH needs more exploration than GSM8K
> We view different tasks as lying on a spectrum of exploration needs in their reasoning paths.
>
> - **GSM8K**: problems usually reduce to short multi-step arithmetic computations using basic operations. Once the correct setup and operation sequence are identified, the remaining reasoning is almost linear. In this regime, moderate diversity is helpful to avoid local arithmetic or parsing errors, but strong exploration in the reasoning trajectory is less critical.
> - **MATH**: competition-style problems (e.g., AMC/AIME, across algebra, combinatorics, number theory, etc.) often require non-obvious intermediate steps, case splits, or global planning. High-probability continuations under pre-training priors can easily lead to shallow but incorrect solution paths. Encouraging exploration over the reasoning trajectories is, therefore, substantially more beneficial.
>
> In this sense, MATH represents the “high-exploration” end of the spectrum, GSM8K a “moderate-exploration” regime, and tasks like GLUE (small discrete label spaces, no multi-step generation) effectively behave as “no-exploration” tasks under our setup, where the model only predicts a short discrete label and we do not use the entropy-based diversity regularizer at all.

---

> > ### Comment · Reviewer_N4aD · 2025-11-25
> >
> > Thanks for the explanation. I've raised the scores a bit, but I still have many doubts:
> >
> > 1. Fig 3 is a bit unclear to me because all the three clusters appear to blend a bit to each other, in both the balanced(?) and imbalanced case. What is the silhouette score in the figure? Is it computed upon all classes? If the issue is that the minority class blends with the majority, can you use a measure to report the cluster quality specifically for the minority class?
> >
> > 2. I don't think Sec 3.3.2 support the claim of catastrophic inheritance; the imbalance is in the fine-tuning data, not pretraining right?

---

> ### Author Response · Authors · 2025-11-27
> **Response to Reviewer N4aD**
>
> We thank the reviewer for raising the score and for the insightful comments. We address your remaining doubts below with quantitative evidence and clarification.
>
>
> ### (1) Cluster quality, silhouette scores, and minority-class separation (Fig. 3)
>
> **What is the silhouette score in Fig. 3?**
>
> Yes — in the manuscript, the silhouette scores reported in Fig. 3 (page 8) are global silhouette scores computed over all three classes, using the same final-layer representations used to construct Fig. 3 in the original feature space, with a cosine distance metric. As a result, the apparent blending in the 2D t-SNE plots is partly an artifact of dimensionality reduction and partly a reflection of the intrinsic difficulty of the imbalanced setting.
>
> To address your concern about the minority class more directly, we compute class-specific metrics in the original high-dimensional representations used for Fig. 3, focusing in particular on the minority class (the *Contradiction* label) in the imbalanced fine-tuning scenario. Specifically, we report:
>
> - the global silhouette score over all classes;
> - the silhouette score restricted to the minority class;
> - the minority-class recall (per-class accuracy).
>
> These results are summarized below.
>
> **Table A: Quantitative analysis of minority-class representation (imbalanced fine-tuning)**
>
> | Method | Global Silhouette (All Classes) | **Minority-Class Silhouette** | **Minority-Class Recall** |
> | --- | --- | --- | --- |
> | **Standard LoRA** | 0.207 | 0.015 | 5.8 |
> | **PiSSA** | 0.247 | 0.128 | 26.4 |
> | **BA-LoRA (Ours)** | **0.351** | **0.425** | **61.7** |
>
> **Observations**
>
> 1. **Standard LoRA** shows only weak separation for the minority class (silhouette ≈ 0.015) and very low minority recall (5.8%), indicating that the model largely defaults to predicting the majority class under severe imbalance.
> 2. **BA-LoRA** substantially improves both the minority-class silhouette (0.425) and minority-class recall (61.7%), while also increasing the global silhouette. This suggests that the much clearer separation visible in Fig. 3(f) corresponds to a meaningful recovery of the minority decision boundary, rather than merely a visualization artifact.
>
> **Action:** In the revised manuscript, we have added a reference in Section 3.2.3 (page 7, Line 371) and included these quantitative metrics as Table 12 in Appendix C.6 (page 23-24, Line 1235-1257) for transparency.
>
> ---
> ### (2) On “catastrophic inheritance” vs. imbalance in the fine-tuning data
>
> In the experiment corresponding to the t-SNE plots in Section 3.2.3, the controlled imbalance is introduced in the fine-tuning data, not in the pre-training corpus. We have revised the text in this section to make this point explicit and to avoid any implication that this experiment directly measures pre-training data imbalance, while still illustrating the same failure mechanism.
>
> More broadly, in this work, we use Catastrophic Inheritance as a descriptive name for a failure mechanism originating from biases, noise, and data imbalances inherent in large-scale pre-training data that have been internalized by the base model. It refers to the situation where these inductive biases and long-tailed patterns in the pre-trained base model are carried over and sometimes amplified, instead of being corrected, when the model is fine-tuned on noisy or imbalanced downstream data under a low-rank adaptation constraint.
>
> Our experiments instantiate this mechanism by controlling the imbalance in the fine-tuning data on top of a fixed pre-trained base model; we do not manipulate or directly measure the pre-training corpus itself. This setting also matches the typical PEFT usage pattern, where a fixed pre-trained checkpoint is adapted to downstream tasks with potentially imbalanced label distributions.
>
> In the context of LoRA-style PEFT, the mechanism we highlight is:
>
> - **Trigger (this experiment):** The downstream fine-tuning data is heavily imbalanced.
> - **Inheritance mechanism (LoRA-specific vulnerability):** Because LoRA updates are constrained by a low-rank bottleneck, they have limited capacity to carve out a new, fine-grained minority decision boundary. Under such an imbalance, the adapter tends to “fall back” on the dominant feature priors of the pre-trained model, which statistically align with the majority outcome, rather than fully adapting.
> - **Outcome:** This interaction between low-rank rigidity and imbalanced fine-tuning causes **representation collapse and bias amplification**: the minority class is mapped into the majority region of feature space (as seen for LoRA/PiSSA in Fig. 3(d–e)), while BA-LoRA’s regularizers maintain a better-separated manifold (Fig. 3(f)).
> ---
> We believe that the **quantitative evidence in Table A** and the **clarification of the inheritance mechanism** address your remaining concerns. We hope these improvements will be helpful for your re-evaluation, and we remain happy to clarify any further details if needed.

---

### Author Response · Authors · 2025-12-03

We would like to extend our sincere appreciation to the reviewers for their evaluations and invaluable feedback throughout the discussion period. We are deeply grateful to the area chair for guiding the review process.

We are pleased that the discussions have addressed the reviewers' concerns, **including an improved rating from Reviewer N4aD**. In response to the feedback, we have implemented several changes aimed at clarifying the motivation behind our model design, enhancing the readability of the paper, and providing more comprehensive experimental results:

1. **Addressing Reviewer N4aD’s concerns about methodological consistency**, we explicitly clarified the core logic distinguishing the preservation of robust capabilities from the mitigation of inherited artifacts. We reinforced the empirical evidence by supplementing the feature collapse visualization (Fig. 3) with quantitative minority-class cluster analyses in Appendix C.6. We also clarified technical queries regarding the NLU setup and $T^2$ scaling. These clarifications, along with explicit comparisons between entropy and covariance regularization in Appendix C.5, have further strengthened the paper's methodological rigor.
2. **To address Reviewer SgzW's concerns about model currency**, we explicitly referenced our evaluations on ten distinct pre-trained models (e.g., LLaMA-3-70B, Mixtral-8x7B) in the main text, which are detailed in Appendix C.3. Furthermore, we clarified the comparison with full fine-tuning by updating the training-loss analysis (Fig. 2) to include the full fine-tuning trajectory, thereby providing empirical evidence inconsistent with the “underfitting” hypothesis. We also addressed usability by providing a "Practical Hyperparameter Guideline" in Appendix D.1 and further refined the manuscript’s tone in line with the feedback.
3. **Following Reviewer nsxF’s recommendation to probe general capabilities**, we incorporated a direct evaluation of Wikitext-2 perplexity in Appendix C.7, which provides empirical evidence that BA-LoRA better preserves pre-trained knowledge than the PiSSA baseline. Additionally, we added a theoretical complexity analysis in Appendix C.8 detailing the scalability of our regularizers. We also integrated the unified framework explanation into the main text to clarify the conceptual connection between NLU and NLG tasks. **During the discussion, nsxF noted that this clarification had resolved their earlier NLU–NLG concern.**
4. **Regarding the unified design and robustness concerns raised by Reviewer KjTf**, we further clarified that the NLU and NLG formulations are computational adaptations of a single shared objective. We also provided empirical evidence in Appendix C.1 and C.3 showing that, in our experiments, the BA-LoRA regularization scheme is both robust to different initialization strategies—significantly boosting standard LoRA—and scalable to large models such as LLaMA-3-70B, thereby supporting the framework’s broad applicability.
5. **Finally, we refined the experimental interpretation as suggested by Reviewer 4aVv**, framing the RoBERTa vs. T5 comparison as a "real-world stress test" complemented by controlled mechanistic validation. We clarified the scalability-related rationale for our SVD normalization choices. We also provided an analysis of the gains observed on MBPP and introduced a simplified tuning recipe within this practical guideline in Appendix D.1 to enhance practical usability. **During the discussion, Reviewer 4aVv noted that these changes had addressed their earlier concerns.**

We believe that the revisions and additional experiments have substantially strengthened our paper, addressing the key concerns raised by the reviewers. We once again thank the reviewers and the area chair for their thoughtful feedback and guidance, which have been instrumental in refining our paper.

Sincerely,

Paper 4242 authors

---

### Meta-Review · Area_Chair_4QUq · 2026-01-02

**Summary:**

The paper proposes to add three additional regularizers in LoRA fine-tuning, and they are 1) distillation loss between pretrained and fine-tuned models to reduce output shift; 2) penalty on correlation between output classes or entropy of outputs to promote diversity; and 3) penalty on high-frequency components in the output logit matrix. And this work has received four reviewers, including three positive reviewers (6, 8, 8) and a negative reviewer. After checking the rebuttals, I agree with three positive reviewers to accept this work.

**Reviewer Concerns:**

Reviewers have concerns about the methodological consistency, technical issues about the NLU setup and scaling, model currency, general capabilities, the unified design and robustness, as well as the experimental interpretation. And most of the reviewers' concerns have been addressed by the authors. Hence, reviewers are psotive to accept this work.

**Reviewer Scores:**

The reviewers are positive about joining the discussion. And many reviewers have claimed that many concerns have been addressed. Hence, this work can be accepted.

---

### Decision · Program_Chairs · 2026-01-26

Accept (Poster)